# Randomized Feasibility Methods for Constrained Optimization with Adaptive Step Sizes

**Abhishek Chakraborty** [1]  **Angelia Nedić** [1]

## Abstract

We consider minimizing an objective function subject to constraints defined by the intersection of lower-level sets of convex functions. We study two cases: (i) strongly convex and Lipschitz-smooth objective function and (ii) convex but possibly nonsmooth objective function. To deal with the constraints that are not easy to project on, we use a randomized feasibility algorithm with Polyak steps and a random number of sampled constraints per iteration, while taking (sub)gradient steps to minimize the objective function. For case (i), we prove linear convergence in expectation of the objective function values to any prescribed tolerance using an adaptive stepsize. For case (ii), we develop a fully problem parameter-free and adaptive stepsize scheme that yields an $O(1/\sqrt{T})$ worst-case rate in expectation. The infeasibility of the iterates decreases geometrically with the number of feasibility updates almost surely, while for the averaged iterates, we establish an expected lower bound on the function values relative to the optimal value that depends on the distribution for the random number of sampled constraints. For certain choices of sample-size growth, optimal rates are achieved. Finally, simulations on a Quadratically Constrained Quadratic Programming (QCQP) problem, Support Vector Machines (SVM), and logistic regression with group fairness constraints demonstrate the computational efficiency of our algorithm compared to other state-of-the-art methods.

[1]School of Electrical, Computer and Energy Engineering, Arizona State University, Tempe, USA. Correspondence to: Abhishek Chakraborty <achakr61@asu.edu>, Angelia Nedić <Angelia.Nedich@asu.edu>.

*Proceedings of the 43rd International Conference on Machine Learning*, Seoul, South Korea. PMLR 306, 2026. Copyright 2026 by the author(s).

## 1. Introduction

We consider the constrained optimization problem

$$\min f(x), \qquad \text{subject to } x \in X \cap Y$$
$$\text{with } X := \cap_{i=1}^{m} \{x \in \mathbb{R}^n \mid g_i(x) \le 0\}, \qquad \text{(P)}$$

where the functions $f : \mathbb{R}^n \to \mathbb{R}$ and $g_i : \mathbb{R}^n \to \mathbb{R}$ for all $i = 1, \dots, m$ are convex, and $m$ can be a large number. We assume that the set $Y \subseteq \mathbb{R}^n$ is closed, convex, and easy to project on (e.g., ball or box constraint), while $X$ is not.

Optimization problems with inequality constraints arise throughout machine learning, operations research, economics, and control. In machine learning, such constraints encode fairness (Zafar et al., 2017), robustness, or structural restrictions on models and predictions (Cotter et al., 2019). In finance and engineering, constraints capture regulatory, budgetary, or physical limits on feasible decisions (Rockafellar et al., 2000; Krokhmal et al., 2002). In energy systems and networked infrastructure, they model capacity, reliability, and safety requirements that must hold under nominal operating conditions (Zampara et al., 2025; Da Costa et al., 2020). Constrained reinforcement learning similarly relies on state-action constraints to ensure safe policy behavior (Chow et al., 2018; Bai et al., 2023), while trust-region and stability constraints underpin many policy optimization methods (Schulman et al., 2015; Achiam et al., 2017). Such constraints also frequently appear as tractable surrogates or relaxations of more complex chance-constrained formulations (Ahmed, 2014; Nemirovski & Shapiro, 2007; Zhang et al., 2023). These broad applications motivate the study of efficient and theoretically grounded methods for solving problem (P) when the number of constraints is large.

**Related Works:** The difficulty in solving problem (P) depends on the structure of the objective function $f$ and the form of the constraint set $X$. For example, if the constraint $X$ consists of intersection of many sets each of which is easy to project on, then one can use random projections (Wang & Bertsekas, 2015; 2016). When projection onto $X$ is difficult, alternative methods are required to minimize the objective while reducing infeasibility. We discuss such methods next.

In machine learning, constraints such as sparsity are often incorporated into the objective via regularization, but

choosing an appropriate regularization parameter is challenging and can yield suboptimal solutions to the original problem. Other existing techniques to solve constrained problems is by introducing penalty terms for constraint violations, for example Huber loss (Tatarenko & Nedić, 2021; Nedić & Tatarenko, 2023) or other barrier functions for deterministic constraints (Freund, 2004; O'Neill & Wright, 2021), and stochastic constraints (Usmanova et al., 2024). Barrier penalty methods are a type of interior-point algorithm and have been studied in the context of variational inequalities (also applicable here) using the Alternating Direction Method of Multipliers (ADMM) with log-barrier penalty functions (Yang et al., 2022). Although ADMM-based methods can yield good solutions, they require careful hyperparameter tuning and solving a subproblem at each iteration, which can be computationally demanding. Primal-dual methods (Zhang et al., 2022b; Li et al., 2024) and Arrow-Hurwicz-Uzawa schemes (He et al., 2022; Zhang et al., 2022a) offer computationally cheaper alternatives to ADMM for solving problem (P), though they may yield inferior performance in some cases. A key drawback of these methods is that all constraints must be handled jointly in the penalty or Lagrangian formulation, which becomes computationally expensive when the number of constraints is large. The Frank-Wolfe method (Beznosikov et al., 2024; Nazykov et al., 2024; Freund & Grigas, 2016; Garber & Hazan, 2015) or the conditional gradient method (Vladarean et al., 2020) for constrained problems require solving a Linear Minimization Oracle which involves finding a minimum within the constraint set, which is not computationally efficient when the projection on the constraint set is complicated.

Randomized feasibility updates (Polyak, 2001) offer a computationally efficient way to handle hard-to-project constraints by subsampling constraints and performing feasibility updates toward their intersection. This approach achieves decay in the infeasibility gap at a geometric rate (Nedić, 2010; Renegar & Zhou, 2021; Chakraborty & Nedić, 2024b) while avoiding the cost of evaluating all constraints at each iteration. Feasibility algorithms have been studied for problems with finitely many constraints (Polyak, 2001; Nedić, 2011; Nedić & Necoara, 2019) and infinitely many constraints (Chakraborty & Nedić, 2024b; Necoara & Singh, 2022). Optimization problems with infinitely many constraints have also been addressed using duality and smoothing techniques (Fercoq et al., 2019), which avoid projections but incur an additional factor of $\ln(k)$ in the convergence rate compared to our random feasibility algorithm studied in the strongly convex setting.

In addition to feasibility enforcement, we address parameter-free step-size selection for solving convex optimization problems. When problem parameters such as strong convexity and Lipschitz smoothness constants are known, step sizes can be chosen accordingly (Nedić & Necoara, 2019;

Chakraborty & Nedić, 2024b). When these parameters are unknown, adaptive step-size schemes are required, including Normalized Gradient Descent (Shor, 2012; Nesterov et al., 2018; Levy, 2017; Grimmer, 2019), line-search methods (Nesterov, 2015), AdGD (Malitsky & Mishchenko, 2020; 2024), local-smoothness-based variants (Latafat et al., 2024), AdaGrad-Norm (Levy et al., 2018; Ene et al., 2021; Ward et al., 2020; Traoré & Pauwels, 2021), D-Adaptation (Defazio & Mishchenko, 2023), Distance over Gradients (DoG) (Ivgi et al., 2023), UniXGrad (Kavis et al., 2019), U-DoG (Kreisler et al., 2024), and Distance over Weighted Gradients (DoWG) (Khaled et al., 2023; Khaled & Jin, 2024), etc. Existing adaptive methods typically require a gradient step followed by projection onto the feasible set. To the best of our knowledge, no fully parameter-free method exists for solving the constrained problem (P) without projections while using computationally inexpensive randomized feasibility updates. For the first time, we study the combination of DoWG or its tamed variant (T-DoWG) with randomized feasibility updates, motivated by the strong empirical performance and parameter-free nature of the method. The Polyak step-size scheme (Polyak, 1969) is also a universal method but requires knowledge of the optimal objective value, which is generally unavailable in constrained optimization; hence, it is not considered here.

**Contributions:** We connect the results from prior works (Lewis & Pang, 1998; Bertsekas, 1999; Mangasarian, 1998), and show that a *uniform* upper bound on dual multipliers exists under Strong Slater condition, without assuming compactness of $Y$. Building on these results, we study the minimization of a strongly convex and Lipschitz-smooth objective function by combining gradient methods with randomized feasibility updates, establishing for the first time linear convergence up to an $\epsilon$ tolerance with adaptive step sizes. In contrast to prior work, we do not assume bounded subgradients of the constraint functions and instead prove this property holds surely. Furthermore, we introduce and analyze a novel strategy involving a random number of feasibility updates per iteration and examine the impact of different sampling distributions on convergence. Moreover, we propose the first fully parameter-free method for solving problem (P) when $f$ is convex but possibly nonsmooth, by combining Distance over Weighted Subgradients (DoWS) with randomized feasibility updates, and provide convergence guarantees under a compactness assumption on $Y$. We then relax this assumption by studying Tamed-DoWS (T-DoWS) with randomized feasibility updates, proving boundedness of the iterates and establishing optimal convergence rates in expectation, along with guidance on choosing the feasibility sample size to achieve these rates.

**Flow of the paper:** Section 2 introduces the assumptions, error bounds, and their connection to Lagrange dual multipliers. Section 3 presents the randomized feasibility algorithm

and its analysis. Section 4 studies gradient method and convergence for strongly convex and Lipschitz-smooth objective function, while Section 5 develops a fully parameter-free adaptive method for convex problems. Section 6 reports simulation results for the problems on QCQP, SVM, and Logistic Regression with Group Fairness Constraints, while Section 7 concludes the paper.[1]

**Notation:** We consider the Euclidean space $\mathbb{R}^n$ with the standard inner product $\langle x, y \rangle$ and the Euclidean norm $\|x\| := \sqrt{\langle x, x \rangle}$. For a closed convex set $X \subseteq \mathbb{R}^n$ and a point $x \in \mathbb{R}^n$, the Euclidean distance from $x$ to the set $X$ is denoted by $\mathrm{dist}(x, X) := \min_{z \in X} \|z - x\|$, while the projection of a point $x$ on the set $X$ is $\Pi_X[x] := \mathrm{argmin}_{x \in X} \|x - \bar{x}\|^2$. For a scalar $a \in \mathbb{R}$, we write $a^+ := \max\{a, 0\}$. The expectation of a random variable is denoted by $\mathbb{E}[\cdot]$. If a random quantity depends on multiple random variables, we will use the subscripts to clarify what the expectation refers to. The probability of a discrete random variable $\omega$ taking value $i$ is denoted by $\mathbb{P}[\omega = i]$. The abbreviation *a.s.* stands for almost surely.

## 2. Assumptions and Error Bounds

We impose the following assumption on the constraint set.

**Assumption 2.1.** The function $g_i : \mathbb{R}^n \to \mathbb{R}$ is convex on $\mathbb{R}^n$ for all $i = 1, \ldots, m$. The constraint set $Y$ is closed and convex, and $X \cap Y$ is nonempty.

Since each $g_i$ is convex, it is continuous on $\mathbb{R}^n$ (Bertsekas et al., 2003) [Proposition 1.4.6], which implies that the set $X$ is convex and closed. Moreover, for each $g_i$, the subdifferential set $\partial g_i(x)$ is non-empty for all $x \in \mathbb{R}^n$ and all $i = 1, \ldots, m$ (Bertsekas et al., 2003) [Proposition 4.2.1]. Consequently, the subdifferential set $\partial g_i^+(x)$ of $g_i^+(x) = \max\{0, g_i(x)\}$ is also non-empty for all $i$.

We also impose the following assumption related to $\mathrm{dist}(x, X \cap Y)$ and violation of the constraint $X$.

**Assumption 2.2.** Let $\omega$ be a random variable taking values in the set $\{1, \ldots, m\}$ with positive probabilities. Assume there exists a constant $c > 0$ such that for all $x \in Y$,

$$\mathrm{dist}^2(x, X \cap Y) \le c\, \mathbb{E}_\omega[(g_\omega^+(x))^2].$$

The constant $c$ may depend on the distribution of $\omega$. Assumption 2.2 is closely related to the notion of global error bound (Lewis & Pang, 1998), stating that: for a proper, closed, and convex function $h : \mathbb{R}^n \to \mathbb{R} \cup \{+\infty\}$, there exists a constant $\gamma > 0$ such that for all $x \in \mathbb{R}^n$,

$$\mathrm{dist}\,(x, \{u \in \mathbb{R}^n \mid h(u) \le 0\}) \le \gamma\, h^+(x).$$

The global error bound is a generalization of Hoffman's error bound (Hoffman, 1952; 2003), which corresponds to the case when the function $h$ is piece-wise affine. Works (Hoffman, 2003; Li, 2013) show that when the constraint functions are affine and the set $Y$ is polyhedral, the global error bound exists. Additionally, existence of global error bounds are shown for general convex functions (Li, 2013; Luo & Luo, 1994; Lewis & Pang, 1998; Hu & Wang, 1989). To connect the global error bound with Assumption 2.2, we let $G(x) = \max_{1 \le i \le m} g_i(x)$ and note that the set $X = \cap_{i=1}^m \{x \in \mathbb{R}^n \mid g_i(x) \le 0\}$ coincides with the set $\{x \in \mathbb{R}^n \mid G(x) \le 0\}$. Define the function $h(x) = G(x) + \delta_Y(x)$ for all $x \in \mathbb{R}^n$, where $\delta_Y(\cdot)$ is the characteristic function of the set $Y$, i.e., $\delta_Y(x) = 0$ if $x \in Y$, and $\delta_Y(x) = +\infty$ otherwise. Thus, $X \cap Y = \{x \in \mathbb{R}^n \mid h(x) \le 0\}$. Under Assumption 2.1, $h(x)$ is proper, closed, and convex, and if it admits a global error bound with constant $\gamma > 0$, then

$$\mathrm{dist}(x, X \cap Y) \le \gamma h^+(x) \quad \text{for all } x \in \mathbb{R}^n. \tag{1}$$

Restricting $x$ to lie in the set $Y$ results in $h(x) = G(x)$, with $G(x) = \max_{1 \le i \le m} g_i(x)$, yielding

$$\mathrm{dist}(x, X \cap Y) \le \gamma \max_{1 \le i \le m} g_i^+(x) \text{ for all } x \in Y. \tag{2}$$

Work (Nedić, 2011) [Section 2.2] shows that under equation (2), Assumption 2.2 is satisfied with $c = \frac{\gamma^2}{\min_{1 \le i \le m} \mathbb{P}[\omega = i]}$. When the distribution of $\omega$ is uniform, the denominator is maximized ($\mathbb{P}[\omega = i] = 1/m$ for all $i$), yielding $c = m\gamma^2$; thus $c$ scales linearly with the number of constraints in this case.

The global error bound holds when the Strong Slater condition $0 \notin \mathrm{cl}(\partial h(h^{-1}(0)))$ is satisfied (Lewis & Pang, 1998) [Corollary 2(b)]. A connection exists between the error bound constant $\gamma$ in relation (1) (Bertsekas, 1999; Mangasarian, 1998) and the optimal dual multipliers (for a related projection problem). Work (Mangasarian, 1998) [Definition 3.2] defines the Strong Slater condition without exploring when the condition is satisfied, while the analysis in (Bertsekas, 1999) [Proposition 3] yields the direct connection between the constant $\gamma$ and the multipliers, as discussed in Appendix A.

## 3. Randomized Feasibility Update

This section focuses on the randomized feasibility update (cf. Algorithm 1) to bypass the projection on the set $X$. Note that Algorithm 1 is called inside a main algorithm for minimizing the objective function $f$ (cf. Algorithms 2 and 3). The subscript $k$ used for the iterates of Algorithm 1 represents the outer iteration index corresponding to gradient steps for the objective function (Algorithms 2 and 3). The superscript $i$ is the iteration index for Algorithm 1. The

---

[1]Code available at `https://github.com/AbhishekChak/Ada-method-random-feas.git`.

---

**Algorithm 1** Randomized Feasibility

---

**Require:** $v_k$, $N_k$, deterministic step size $0 < \beta < 2$
1: **Initialization:** $z_k^0 = v_k$
2: **for** $i = 1, 2, \ldots, N_k$ **do**
3:    **Sample:** Index $\omega_k^i \in \{1, \ldots, m\}$ uniformly
4:    **Compute:** Subgradient $d_k^i \in \partial g_{\omega_k^i}^+(z_k^{i-1})$
5:    **Update:** $z_k^i = \Pi_Y\left[z_k^{i-1} - \beta \frac{g_{\omega_k^i}^+(z_k^{i-1})}{\|d_k^i\|^2} d_k^i\right]$
6: **end for**
7: **Output** $x_k = z_k^{N_k}$ .

---

algorithm requires an input point $v_k$ and the number of samples $N_k$. Unlike prior work (Nedić, 2010; Nedić & Necoara, 2019; Necoara & Singh, 2022; Singh & Necoara, 2024a; Chakraborty & Nedić, 2024b; 2025), we allow $N_k$ to be either deterministic or random, drawn from a discrete set $\mathcal{I}_k \subseteq \mathbb{N}$ with distribution $\mathcal{D}_k$ (cf. Appendix D.5).

The algorithm generates a sequence of iterates $\{z_k^i\}$ via $N_k$ subgradient steps. At each iteration $i$, it uniformly samples a constraint index $\omega_k^i \in \{1, \ldots, m\}$, computes a subgradient $d_k^i \equiv d_k^i(z_k^{i-1}) \in \partial g_{\omega_k^i}^+(z_k^{i-1})$ of the corresponding sampled constraint function at the current iterate, takes a subgradient step with the Polyak step size (Polyak, 1969; 2001) scaled by a parameter $\beta \in (0, 2)$, and projects onto the set $Y$. After $N_k$ iterations, the final iteration $z_k^{N_k}$ is the output of the method, denoted by $x_k$. This method provides an efficient and scalable approach for handling feasibility in the presence of uncertainty and data-driven constraints.

Let the initial iterate $x_0 \in Y$ be random with $\mathbb{E}[\|x_0\|] < \infty$. Define the sigma algebra $\mathcal{F}_0 = \{x_0\}$ and $\mathcal{F}_k, \forall k \geq 1$, as

$$\mathcal{F}_k = \{x_0\} \cup \{w_t^i \cup N_t \mid 1 \leq i \leq N_t, 1 \leq t \leq k\}. \quad (3)$$

The main optimization algorithm for the outer loop will be discussed in the subsequent sections. Next, we present a lemma that relates the distances of the output $x_k$ of Algorithm 1 and the input $v_k$ from an arbitrary point $x \in X \cap Y$.

**Lemma 3.1.** *Let Assumptions 2.1 and 2.2 hold. Then, the iterates $v_k, x_k \in Y$ satisfy for all $x \in X \cap Y$:*

*(a)* $\|x_k - x\|^2 \leq \|v_k - x\|^2$ *surely for all $k \geq 1$.*

*(b)* *If the subgradients $\{d_k^i\}$ are bounded, i.e., $\|d_k^i\| \leq M_g$ for all $i$ and $k$, then we have a.s. for all $k \geq 1$,*

$$\mathbb{E}\left[\|x_k - x\|^2 \mid \widetilde{\mathcal{F}}_{k-1}\right] \leq \|v_k - x\|^2$$
$$- ((1-q)^{-N_k} - 1)\mathbb{E}\left[\text{dist}^2(x_k, X \cap Y) \mid \widetilde{\mathcal{F}}_{k-1}\right],$$
$$\mathbb{E}\left[\text{dist}(x_k, X \cap Y) \mid \widetilde{\mathcal{F}}_{k-1}\right] \leq (1-q)^{\frac{N_k}{2}} \|v_k - x\|,$$

*where $\widetilde{\mathcal{F}}_{k-1} = \mathcal{F}_{k-1} \cup \{N_k\}$ and $q = \frac{\beta(2-\beta)}{cM_g^2}$ with $\beta \in (0, 2)$ being the step size of Algorithm 1 and $c$ is the constant in Assumption 2.2.*

The proof follows arguments in (Polyak, 1969; 2001; Nedić & Necoara, 2019; Chakraborty & Nedić, 2025), with minor modifications to the sigma field to accommodate random sample sizes $N_k$ (cf. Appendix C.1). The proof of Lemma 3.1(a) relies on upper estimates for the distance between the iterate $z_k^i$ and any point $x \in X \cap Y$, using the properties of subgradient step and the convexity of the constraint functions along with the use of Polyak step sizes. The relations in Lemma 3.1(b) are obtained by utilizing Polyak step sizes, Jensen's inequality, and Assumption 2.2. The last relation of the lemma shows a geometric decay in the distance from $x_k$ to the constraint set with respect to the number of samples $N_k$. Lemma 3.1(b) assumes bounded subgradients $\{d_k^i\}$, this assumption is unnecessary for certain step-size choices in the main optimization algorithm (cf. Corollary D.2 and Lemma F.1).

## 4. Strongly Convex and Lipschitz Smooth Objective Function

In this section, we consider a gradient-based algorithm for minimizing the objective function $f$ to solve problem (P). We impose the following assumptions on $f$.

**Assumption 4.1.** The function $f : \mathbb{R}^n \to \mathbb{R}$ has Lipschitz continuous gradients over $Y$ with a constant $L > 0$, i.e.,

$$f(y) - f(x) \leq \langle \nabla f(x), y - x \rangle + \frac{L}{2}\|y - x\|^2 \quad \forall x, y \in Y.$$

**Assumption 4.2.** The function $f : \mathbb{R}^n \to \mathbb{R}$ is strongly convex over $Y$ with the constant $\mu > 0$, i.e.,

$$f(y) - f(x) \geq \langle \nabla f(x), y - x \rangle + \frac{\mu}{2}\|y - x\|^2 \quad \forall x, y \in Y.$$

Assumption 4.2 ensures the existence of a unique minimizer $x^*$ to the problem (P). Next, we present our method.

Algorithm 2 updates using the gradient of the objective function and its outputs are fed to Algorithm 1 to reduce the feasibility violation with respect to the constraint set $X$. Under Assumptions 4.1 and 4.2, prior work (e.g., (Nedić &

---

**Algorithm 2** Gradient Method with Randomized Feasibility

---

**Require:** Random initialization $x_0 \in Y$, step size $\alpha_k > 0$
1: **for** $k = 0, 1, 2, \ldots, T$ **do**
2:    $v_{k+1} = \Pi_Y[x_k - \alpha_k \nabla f(x_k)]$
3:    $x_{k+1} = \text{Algorithm } 1(v_{k+1}, N_{k+1})$
4: **end for**

---

Necoara, 2019)) established an $O(1/T)$ convergence rate

of Algorithm 2 with diminishing step sizes. In contrast, we prove linear convergence up to a prescribed tolerance using adaptive step sizes. We next present a generic lemma.

**Lemma 4.3.** *Let Assumptions 2.1, 4.1, and 4.2 hold. Then, the iterates $v_{k+1}$ and $x_k$ of Algorithm 2 satisfy the following relation surely for all $y \in Y$ and $k \geq 0$,*

$$\|v_{k+1} - y\|^2 \leq (1 - \alpha_k \mu)\|x_k - y\|^2$$
$$- (1 - \alpha_k L)\|v_{k+1} - x_k\|^2 - 2\alpha_k(f(v_{k+1}) - f(y)).$$

The analysis follows standard arguments and the proof is given in Appendix D.1. Since Algorithm 1 generates points in $Y$ that approach $X$ but may remain infeasible, both $v_{k+1}$ and $x_{k+1}$ can violate the constraints. Consequently, setting $y = x^*$ in Lemma 4.3 may yield $f(v_{k+1}) - f(x^*) < 0$, preventing a direct application of standard gradient descent analysis and introducing additional analytical challenges.

It can be proved that under Assumptions 2.1, 4.1, and 4.2 and $\alpha_k \in (0, \frac{1}{L}]$, the sequence of iterates $\{x_k\}$, $\{v_k\}$ and $\{z_k^i, i = 1, \ldots, N_k\}$ are surely bounded, i.e., $\forall k \geq 1$,

$$\|x_k - x^*\| \leq B_0 + \|x^*\|, \qquad \|v_k - x^*\| \leq B_0 + \|x^*\|,$$

$$\|z_k^i - x^*\| \leq B_0 + \|x^*\| \quad \text{for all } i = 1, \ldots, N_k - 1,$$

where $B_0$ is a constant defined formally in Lemma D.1. As a consequence (Bertsekas et al., 2003)[Proposition 4.2.3], there exist (deterministic) scalars $M_g > 0$ and $M_f > 0$ such that the gradients $\|\nabla f(x_k)\| \leq M_f$ and the subgradients $\|d_k^i\| \leq M_g$ surely for all $i = 1, \ldots, N_k$ and $k \geq 1$ (cf. Corollary D.2). The prior works using randomized feasibility updates (Nedić & Necoara, 2019; Singh & Necoara, 2024a; Singh et al., 2024; Singh & Necoara, 2024b; Necoara & Singh, 2022) have assumed boundedness of $\{d_k^i\}$.

As shown in Appendix D.3, with exponentially weighted averages $\bar{v}_k$ and $\bar{x}_k$ of $\{v_t\}$ and $\{x_t\}$, and a constant stepsize $\alpha_k = \alpha \in (0, \frac{1}{L}]$, $f(\bar{v}_k) - f(x^*)$ admits a geometrically decaying upper bound surely, whereas $\mathbb{E}[f(\bar{x}_k) - f(x^*)]$ admits a geometrically decaying lower bound almost surely (depending on the number of samples $N_k$) for all $k \geq 1$. Moreover, Lemma 4.3 with $y = x_k$ yields $f(v_{k+1}) \leq f(x_k)$ surely for any $\alpha \in (0, \frac{2}{L}]$. If $\nabla f(x^*) = 0$, the iterate $v_k$ converge geometrically fast surely; otherwise, each gradient step still decreases $f(v_{k+1}) < f(x_k)$, while the randomized feasibility step moves $f(x_{k+1})$ toward $f(x^*)$ in the expected sense. Using diminishing stepsizes $\alpha_k = \frac{4}{\mu(k+1)}$ yields $O(1/\sqrt{k})$ rate for $\mathbb{E}[|f(\tilde{x}_k) - f(x^*)|]$ (Nedić & Necoara, 2019)[Theorem 3], where $\tilde{x}_k$ is $\alpha_k$-weighted average of $x_k$. In contrast, with adaptive stepsizes, intelligent averaging, and sufficient number of feasibility samples $N_k$, $\mathbb{E}[|f(\bar{x}_k) - f(x^*)|]$ decays geometrically fast up to an error tolerance, as shown in the following theorem.

**Theorem 4.4.** *Let Assumptions 2.1, 2.2, 4.1, and 4.2 hold. Let $\epsilon > 0$ be arbitrary and let the step size be given by $\alpha_k = \min\left\{\frac{1}{2(L-\mu)}, \frac{1}{L}, \frac{\epsilon}{2\|\nabla f(x_k)\|^2}\right\}$. Also, assume that the sample size selection is such that $\mathbb{E}\left[(1-q)^{\frac{N_t}{2}}\right] \leq \frac{\epsilon}{(B_0 + \|x^*\|)M_f}$ for all $1 \leq t \leq k$. Then, Algorithm 2 reaches an $\epsilon$-accuracy in expectation, i.e., $\mathbb{E}[|f(\bar{x}_k) - f(x^*)|] \leq \epsilon$, in the number of iterations $k \geq O\left(\ln\left(\frac{1}{\epsilon}\right)\right)$, where $\bar{x}_k = \frac{\sum_{t=1}^{k}(1-\bar{\alpha}\mu)^{k-t}\alpha_t x_t}{\sum_{s=1}^{k}\alpha_s(1-\bar{\alpha}\mu)^{k-s}}$ and $\bar{\alpha} = \min\left\{\frac{1}{2(L-\mu)}, \frac{1}{L}, \frac{\epsilon}{2M_f^2}\right\}$. The constants $L$, $\mu$, $q$, and $M_f$ are defined in Assumption 4.1, Assumption 4.2, Lemma 3.1, and Corollary D.2, respectively, while the bound $B_0 + \|x^*\|$ comes from Lemma D.1.*

For all $k \geq 1$, the sample size $N_k$ depends on the support $\mathcal{I}_k \subset \mathbb{N}$ and the distribution $\mathcal{D}_k$, which may vary with $k$. Additional discussion and the proof of Theorem 4.4 are provided in Appendix D.4, while Appendix D.5 details the computation of $\mathbb{E}[(1-q)^{N_k/2}]$ for various distributions (e.g., Poisson, Uniform, and Binomial). Note that the constant $\bar{\alpha}$ in Theorem 4.4 depends on $M_f$ (see Corollary D.2).

# 5. Parameter-free Adaptive Step Size for Convex Nonsmooth Objective Function

In the section, we present parameter-free approaches for solving problem (P) without requiring the knowledge of any problem specific parameters. Here, we consider the following assumption on the objective function $f$.

**Assumption 5.1.** The function $f : \mathbb{R}^n \to \mathbb{R}$ is convex and potentially nonsmooth on $\mathbb{R}^n$, i.e.,

$$f(y) - f(x) \geq \langle s_f(x), y - x \rangle \qquad \text{for all } x, y \in \mathbb{R}^n,$$

where $s_f(x) \in \partial f(x)$ is a subgradient of the function $f$ at $x$. We also assume that optimal function value $f^*$ of problem (P) is finite.

## 5.1. Distance over Weighted Subgradients (DoWS) with Randomized Feasibility

For this subsection, we consider the following assumption on the constraint set $Y$.

**Assumption 5.2.** The constraint set $Y$ is bounded, i.e., $\max_{x,y \in Y} \|x - y\| \leq D$.

Assumption 5.2 ensures bounded subgradients of the objective function and the constraints (Bertsekas et al., 2003)[Proposition 4.2.3], i.e., $\|s_f(x)\| \leq M_f$ for all $x \in Y$, and $\|d_k^i\| \leq M_g$ for all $i = 1, \ldots, N_k$ and all $k \geq 1$.

We study the parameter-free adaptive stepsize scheme Distance over Weighted Gradients (DoWG) (Khaled et al., 2023; Khaled & Jin, 2024), which estimates the set diameter and combines weighted subgradient norms to set the stepsize. With per-iteration projection on $X$, DoWG is universal,

achieving $O(1/T)$ rates for Lipschitz-smooth objectives and $O(1/\sqrt{T})$ rates for Lipschitz (possibly nonsmooth) objectives. Without projection, the iterates may be infeasible, invalidating the descent lemma and the $O(1/T)$ rate. Thus, we consider a weaker assumption that $f$ is possibly nonsmooth, and show that DoWG attains the optimal $O(1/\sqrt{T})$ rate. In this setting, we use subgradients and combine them with Algorithm 1 to control infeasibility, referring to the resulting method as Distance over Weighted Subgradients (DoWS) with randomized feasibility – see Algorithm 3.

---

**Algorithm 3** DoWS with Randomized Feasibility

---

1: **Input:** $v_0 = v_1 \in Y$, estimates $p_0 = 0$, $\bar{r}_0 = r > 0$
2: **Pass:** $v_1$ and $N_1 \in \mathcal{I}_1$ to Algorithm 1 to obtain $x_1 \in Y$
3: **Equate:** $x_0 = x_1 \in Y$
4: **for** $k = 1, 2, \ldots, T$ **do**
5: $\quad \bar{r}_k = \max\{\|x_k - x_0\|, \bar{r}_{k-1}\}$
6: $\quad p_k = p_{k-1} + \bar{r}_k^2 \|s_f(x_k)\|^2$
7: $\quad v_{k+1} = \Pi_Y[x_k - \alpha_k s_f(x_k)]$ with $\alpha_k = \dfrac{\bar{r}_k^2}{\sqrt{p_k}}$
8: $\quad x_{k+1} = \text{Algorithm 1}(v_{k+1}, N_{k+1})$
9: **end for**

---

The algorithm initializes with a small estimate $r$ of the set diameter $D$ ($0 < r < D$) and tracks the maximum distance $\bar{r}_k$ of $x_k$ from a random initial point $x_0 \in Y$ with $\mathbb{E}[\|x_0\|^2] < \infty$. We set $x_1 = x_0$. The variable $p_k$ accumulates weighted subgradient norm squares, yielding the adaptive stepsize $\alpha_k = \bar{r}_k^2/\sqrt{p_k}$. A subgradient step produces $v_{k+1}$, followed by a randomized feasibility update (Algorithm 1) with sample size $N_{k+1}$ to obtain $x_{k+1}$.

We now present the convergence rates for the method.

**Theorem 5.3.** *Let Assumptions 2.1, 2.2, 5.1, and 5.2 hold. Let $\{x_k\}$ be a sequence of iterates generated by Algorithm 3. For any $T \geq 1$ and $\tau = \text{argmin}_{1 \leq k \leq T} \frac{\bar{r}_{k+1}^2}{\sum_{i=1}^{k} \bar{r}_i^2}$, the averaged iterate $\bar{x}_\tau = \frac{\sum_{k=1}^{\tau} \bar{r}_k^2 x_k}{\sum_{i=1}^{\tau} \bar{r}_i^2}$ satisfies the following:*

$$\mathbb{E}[|f(\bar{x}_\tau) - f^*|] \leq \max\{A_1(T), \min\{A_2(T), A_3(T)\}\},$$

*where* $A_1(T) = \dfrac{3DM_f}{\sqrt{T}}\left(\dfrac{D}{r}\right)^{\frac{2}{T}}\ln\left(\dfrac{eD^2}{r^2}\right)$,

$A_2(T) = DM_f \max_{1 \leq k \leq T} \mathbb{E}\left[(1-q)^{\frac{N_k}{2}}\right]$,

$A_3(T) = \dfrac{DM_f}{T}\left(\dfrac{D}{r}\right)^{\frac{2}{T}}\ln\left(\dfrac{eD^2}{r^2}\right)\sum_{k=1}^{T}\mathbb{E}\left[(1-q)^{\frac{N_k}{2}}\right]$.

The constants in the original DoWG analysis (Khaled et al., 2023) are not optimal and using Lemma B.3 (see also (Liu & Zhou, 2023)[Lemma 30] and (Moshtaghifar et al., 2025)[Lemma A.1]), the constants in Theorem 5.3 has been tightened. The bound $A_1(T)$ arises from the gradient update on the objective function, whereas $A_2(T)$ and $A_3(T)$

come from the analysis of the feasibility violation. Both $A_1(T)$ and $A_3(T)$ involve the quantity $\left(\frac{D}{r}\right)^{\frac{2}{T}}$, which tends to 1 as $T$ increases and can be replaced by a modest constant for all $T \geq 1$. The proof and further discussion are provided in Appendix E.1. The infeasibility bound depends on $\min\{A_2(T), A_3(T)\}$ and decreases at rate $O(1/T)$ provided $\sum_{k=1}^{\tau} \mathbb{E}\left[(1-q)^{\frac{N_k}{2}}\right]$ is uniformly bounded by a constant independent of the total number of iterations $T$. Such bounds hold for both deterministic and stochastic sampling schemes using arguments similar to (Chakraborty & Nedić, 2025) (see Appendix E.2). For example, if $N_k = \lceil k^{\frac{1}{p}} \rceil$ for some $p > 0$ for all $k \geq 1$, then $\sum_{k=1}^{\tau}(1-q)^{\frac{N_k}{2}}$ is bounded by a constant. Consequently, Theorem 5.3 implies a worst-case rate of $O(1/\sqrt{T})$, up to logarithmic factors depending on the ratio between the set $Y$ diameter $D$ and the initial distance estimate $r$.

### 5.2. Tamed Distance over Weighted Subgradients (T-DoWS) with Randomized Feasibility

Subsection 5.1 assumes a bounded set $Y$ (Assumption 5.2) to guarantee bounded subgradients and enable the convergence analysis of Algorithm 3. However, if $Y = \mathbb{R}^n$, these properties may fail. Our goal is therefore to remove Assumption 5.2 by ensuring bounded iterates. In addition, the aggressive stepsize $\alpha_k$ in Algorithm 3 can accelerate early progress but may cause oscillations between feasibility and infeasibility, especially when the optimal solution to Problem (P) lies on the boundary while the optimal value of the problem $\min_{x \in Y} f(x)$ is significantly smaller.

We modify Algorithm 3 to obtain Algorithm 4 (see Appendix F.1), in which the stepsizes are further scaled to yield a less aggressive sequence $\{\alpha_k\}$, defined as

$$\alpha_k = \begin{cases} \dfrac{\bar{r}_k^2}{2\sqrt{p_k}\ln\left(\frac{ep_k}{p_1}\right)} & \text{if } p_0 = 0, \\[3mm] \dfrac{\bar{r}_k^2}{\sqrt{2p_k}\ln\left(\frac{ep_k}{p_0}\right)} & \text{if } p_0 > 0. \end{cases}$$

The method guarantees bounded iterates under the following sub-level set assumption on $f$, together with Assumption 5.1, where $f^*$ denotes the optimal value of problem (P).

**Assumption 5.4.** *The sub-level set $\{y \in Y \mid f(y) \leq f^*\}$ is bounded, i.e., there exists a positive constant $B$ such that $\|x\| \leq B$ for all $x$ in the sublevel set $\{y \in Y \mid f(y) \leq f^*\}$.*

Assumption 5.4 holds, for example, when $f$ is coercive; there are also other examples satisfying this assumption but are not discussed here. Using this assumption, it can be proved that if the initial estimate satisfies $r \leq 4\|x_0 - x^*\|$, where $x^*$ is a solution of problem (P), then the subgradients of the objective and constraint functions are surely bounded, and so are the sequences $\{\bar{r}_k\}$, $\{x_k\}$, $\{v_k\}$, and $\{z_k^i\}$ for $i = 1, \ldots, N_k$ and for all $k \geq 1$, i.e., for $i = 1, \ldots, N_k$ and

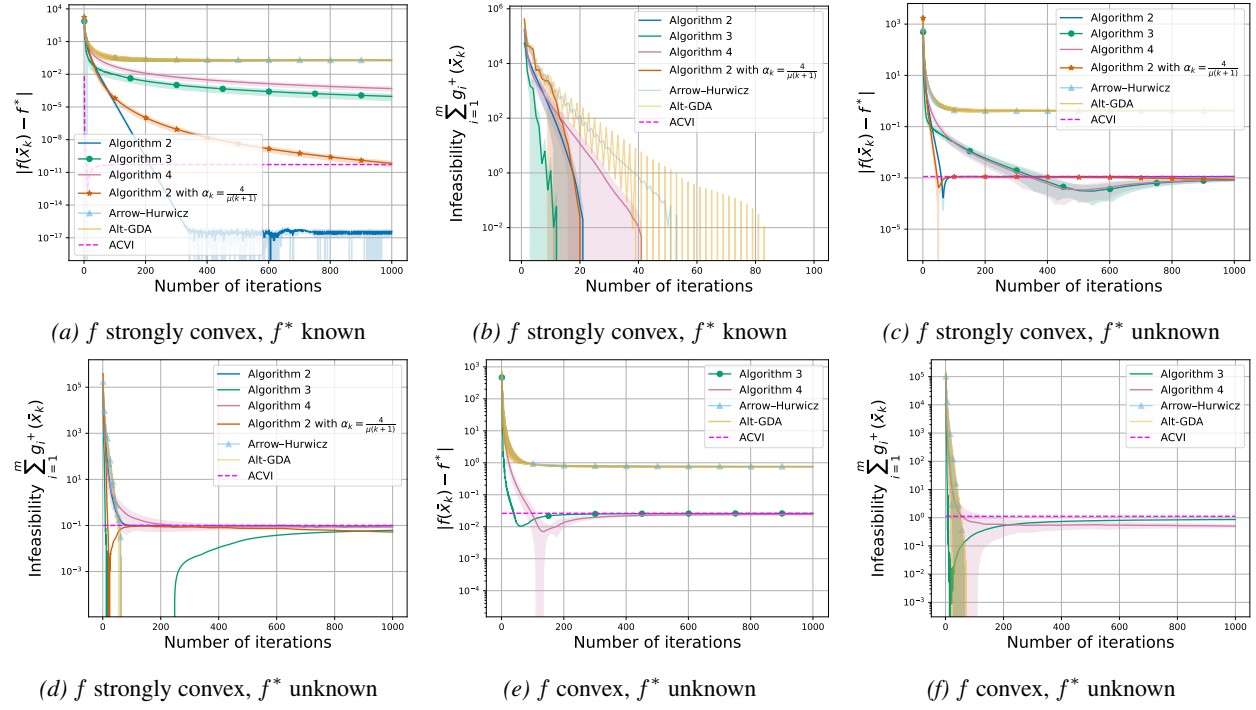

*Figure 1.* Simulation plots for the QCQP problem for the three cases considered. We show the decay in the function values as well as the infeasibility for all the algorithms. In the cases where the optimal function value $f^*$ is unknown, we use the value returned by the CVXPY solver as $f^*$. We choose $N_k = \lceil k^{1/2} \rceil$.

for all $k \geq 1$,

$$\|s_f(x_k)\| \leq M_f, \ \|d_k^i\| \leq M_g, \ \bar{r}_k \leq \widehat{B}, \ \|x_k - x^*\| \leq \widetilde{B},$$
$$\|z_{k+1}^i - x^*\| \leq \|v_{k+1} - x^*\| \leq \bar{D}(p_0),$$

where $M_f, M_g, \widehat{B}, \widetilde{B}$, and $\bar{D}(p_0)$ are some positive constants. This is shown in Lemma F.1 of Appendix F.2, where the constants $\widehat{B}, \widetilde{B}$, and $\bar{D}(p_0)$ are formally derived.

Next, we present the convergence rate of the method.

**Theorem 5.5.** *Let Assumptions 2.1, 2.2, 5.1, and 5.4 hold.*
*Define $\tau = \arg\min_{1 \leq k \leq T} \frac{\bar{r}_{k+1}^2}{\sum_{i=1}^k \bar{r}_i^2}$ and the averaged iterate*
*$\bar{x}_\tau = \frac{\sum_{k=1}^\tau \bar{r}_k^2 x_k}{\sum_{i=1}^\tau \bar{r}_i^2}$. Then, for any $T \geq 1$,*

$$\mathbb{E}[|f(\bar{x}_\tau) - f^*|] \leq \max\{\bar{A}_1(T), \min\{\bar{A}_2(T), \bar{A}_3(T)\}\},$$

*where* $\bar{A}_1(T) = \frac{M_f \widehat{D}(p_0,T)}{\sqrt{T}} \left(\frac{\widehat{B}}{r}\right)^{\frac{2}{T}} \ln\left(\frac{e\widehat{B}^2}{r^2}\right)$ *with*

$$\widehat{D}(p_0,T) = \begin{cases} 4\ln\left[\frac{e\widehat{B}^2 M_f^2 T}{r^2 \|s_f(x_0)\|^2}\right] \widetilde{B} + \frac{\widehat{B}}{2} & \text{if } p_0 = 0, \\ 2\sqrt{2}\ln\left[\frac{e\widehat{B}^2 M_f^2(T+1)}{p_0}\right] \widetilde{B} + \frac{\widehat{B}}{2\sqrt{2}} & \text{if } p_0 > 0, \end{cases}$$

$$\bar{A}_2(T) = M_f \bar{D}(p_0) \max_{1 \leq k \leq T} \mathbb{E}\left[(1-q)^{\frac{N_k}{2}}\right], \quad \text{and}$$

$$\bar{A}_3(T) = \frac{M_f \bar{D}(p_0)}{T} \left(\frac{\widehat{B}}{r}\right)^{\frac{2}{T}} \ln\left(\frac{e\widehat{B}^2}{r^2}\right) \sum_{k=1}^T \mathbb{E}\left[(1-q)^{\frac{N_k}{2}}\right].$$

*The constants $\widehat{B}, \widetilde{B}$, and $\bar{D}(p_0)$ are defined in Lemma F.1.*

The analysis follows similar steps to Theorem 5.3, with slight changes in the constants, and is provided in Appendix F.3. Compared to Theorem 5.3, no boundedness assumption on the constraint set is required, but the convergence rate incur an additional $\ln(T)$ factor (inside $\widehat{D}(p_0,T)$). The same sample-size selection arguments for the randomized feasibility algorithm (Algorithm 1) discussed after Theorem 5.3 also apply here.

# 6. Simulation

This section presents the QCQP, SVM, and Logistic Regression with Group Fairness Constraints, all run on a MacBook Pro 2021 (Apple M1 Pro chip, 16 GB RAM). An extended version of Section 6, including the problem setup, parameter selection, runtime analysis, dataset details, and discussion of plots, is provided in Appendix G.

## 6.1. Quadratically Constrained Quadratic Programming (QCQP)

We consider the following optimization problem

$$\min_{x \in [-10,10]^{10}} \langle x, Ax \rangle + \langle b, x \rangle$$
$$\text{s.t. } \langle x, C_i x \rangle + \langle u_i, x \rangle - e_i \leq 0 \quad \text{for } i = 1, \dots, m.$$

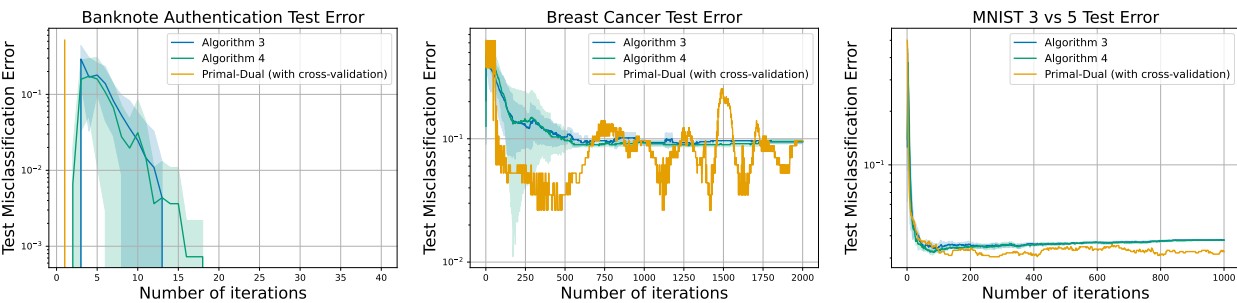

*Figure 2.* Test misclassification error for Algorithms 3 and 4, and the primal-dual method with step sizes chosen via 3-fold cross-validation.

We consider three problem setups: (i) strongly convex objective function with known $f^*$, (ii) strongly convex objective function with unknown $f^*$, and (iii) convex objective function with unknown $f^*$. We compare Algorithms 2, 3, and 4 with the method in (Nedić & Necoara, 2019) (i.e., Algorithm 2 with $\alpha_k = 4/\mu(k+1)$), the Arrow-Hurwicz method (He et al., 2022), Alt-GDA (Zhang et al., 2022a), ADMM with log-barrier penalty functions known as ACVI (Yang et al., 2022), and CVXPY (Diamond & Boyd, 2016; Agrawal et al., 2018). We observe that CVXPY and IPOPT (Wächter & Biegler, 2006) fails for high-dimensional problems; CVXPY also fails when the number of constraints is large. Hence, we set number of constraints $m = 1000$. For cases (ii) and (iii), where $f^*$ is unknown, the CVXPY solution is used as a reference for $f^*$. Across all cases, ACVI incurs the highest computational cost and requires careful hyperparameter tuning, as it may otherwise diverge; when properly tuned, however, it converges the fastest. The decay in the objective function and infeasibility for all the algorithms is shown in Figure 1. Algorithm 2 achieves the best objective convergence in case (i), showing a linear trend, while Algorithms 3 and 4 perform well in cases (ii) and (iii), being fully problem parameter-free and adaptive. ACVI attains the fastest infeasibility reduction, while our methods remain competitive and improve with larger $N_k$; Arrow-Hurwicz and Alt-GDA perform slightly worse with mild oscillations. The effect of increasing $N_k$ is illustrated for Algorithm 3 in the convex case in Figure 5 (cf. Appendix G.1).

### 6.2. Support Vector Machine (SVM) Classification

We consider a binary classification problem with a supervised dataset $\{z_i, y_i\}_{i=1}^m$, where $z_i$ denotes the feature vectors, $y_i \in \{-1, +1\}$ are the corresponding class labels, and $m$ is the number of training data points. We consider the soft-margin SVM classification problem

$$\min_{w,b,\xi} \quad \frac{1}{2}\|w\|^2 + C\sum_{i=1}^m \xi_i$$

subject to   $g_i(w, b, \xi_i) \leq 0$,   $\xi_i \geq 0$,   $i = 1, \ldots, m$,

where $g_i(w, b, \xi_i) = 1 - \xi_i - y_i(w^\top z_i + b)$ and $C > 0$ is the hyperparameter controlling the tradeoff between margin size and misclassification penalties. Here, we choose $C = 10^{-6}$. We consider 3 different datasets: (i) Banknote Authentication (Lohweg, 2012), (ii) Breast Cancer Wisconsin (Wolberg et al., 1993), and (iii) MNIST 3 vs 5 digit classification (LeCun et al., 2010). Since, the objective function is convex, we implement Algorithms 3 and 4 and compare it with Lagrangian primal-dual updates. The step-size rules in Arrow–Hurwicz (He et al., 2022) and Alt-GDA (Zhang et al., 2022a) fail to converge, whereas our methods use adaptive step sizes and converge without requiring problem parameters. Hence to obtain good primal and dual step sizes for the primal-dual method, we performed 3-fold cross validation on the train set. This cross-validation step adds computational overhead. The dimension of dual variable vector equals the number of training samples, making primal-dual methods more suitable for offline settings; accordingly, the full dataset is processed at each iteration. In contrast, our Algorithms 3 and 4 enforce constraints using randomly sampled data points and therefore do not require processing of the full dataset at each iteration. Hence, our algorithms are more suitable for online learning. Figure 2 reports the test misclassification error. The Banknote Authentication dataset exhibits better class separability than other datasets. All the algorithms achieve comparable performance. Additional plots and detailed discussion of objective and feasibility behavior are provided in Appendix G.

### 6.3. Constrained Logistic Regression with Group Fairness Constraints

Let us consider the optimization problem

$$\min_{x \in \mathbb{R}^n} \quad f(x) := \frac{1}{\widehat{N}}\sum_{j=1}^{\widehat{N}} \ln(1 + \exp(-y_j\langle a_j, x\rangle))$$

s.t.   $x \in Y \cap X$, where $Y := \{x \in \mathbb{R}^n \mid \ell \leq x \leq u\}$,

and $X := \bigcap_{i=1}^m \{x \in \mathbb{R}^n \mid |c_i^\top x| \leq \tau_i\}$,

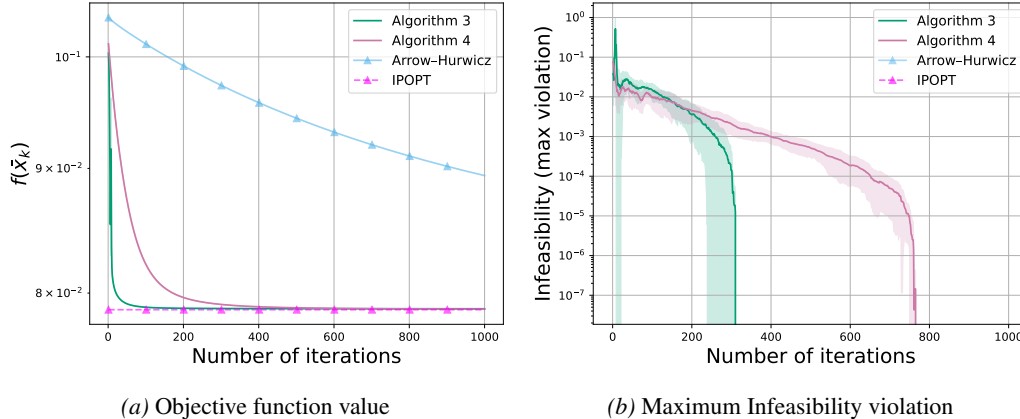

*(a)* Objective function value         *(b)* Maximum Infeasibility violation

*Figure 3.* Simulation plots for the constrained logistic regression with fairness constraints with $N_k = 100 + \lceil k^{1/1.1} \rceil$.

where $\{(a_j, y_j)\}_{j=1}^{\widehat{N}}$ denote the dataset with the feature vectors $a_j \in \mathbb{R}^n$ and the labels $y_j \in \{-1, +1\}$. The set $Y$ encodes simple box constraints and the set $X$ encodes group fairness constraints. To construct the fairness constraints, we partition the dataset into $G$ demographic groups indexed by $\mathcal{G}_1, \ldots, \mathcal{G}_G$. For each group, we define the empirical mean feature vector

$$\mu_r := \frac{1}{|\mathcal{G}_r|} \sum_{j \in \mathcal{G}_r} a_j \quad \text{for all } r = 1, \ldots, G.$$

We then generate $m$ fairness constraints by randomly selecting pairs of groups $(r_1, r_2)$ and defining $c_i := \mu_{r_1} - \mu_{r_2}$. This leads to constraints enforcing similarity of model responses across groups:

$$\left| (\mu_{r_1} - \mu_{r_2})^\top x \right| \leq \tau_i.$$

The constraint thresholds are chosen as

$$\tau_i := \left| c_i^\top \widehat{x} \right| + \eta,$$

where $\eta > 0$ is a small slack parameter ensuring feasibility of $\widehat{x}$. Each fairness constraint can equivalently be written as two convex inequalities:

$$c_i^\top x - \tau_i \leq 0, \qquad -c_i^\top x - \tau_i \leq 0.$$

For the synthetic data generation, we sample feature vectors $a_j \sim \mathcal{N}(0, I_n)$ and generate labels according to a logistic model

$$\mathbb{P}(y_j = 1 \mid a_j) = \sigma(a_j^\top \widehat{x}),$$

where $\sigma(t) = (1 + e^{-t})^{-1}$ and $\widehat{x} \in \mathbb{R}^n$ is a ground-truth vector sampled from a Gaussian distribution and projected onto the box $[\ell, u]$. The parameter selection of the problem and the CPU runtime is provided in Appendix G.3.

We ran our Algorithms 3 and 4, and compared them with the Arrow-Hurwicz method. To obtain a baseline estimate of the optimal function value, we additionally employed the state-of-the-art solver IPOPT.

We report the function values and the maximum infeasibility violation across iterations for all the algorithms in Figure 3. From Figure 3, it is apparent that Algorithms 3 and 4 converge to the baseline optimal function value obtained by IPOPT. In contrast, the convergence of the Arrow-Hurwicz method is relatively slow. Moreover, IPOPT and Arrow-Hurwicz exhibit the best feasibility performance, while our algorithms also achieve zero feasibility violation after a sufficient number of iterations. Note that we choose $N_k = 100 + \lceil k^{1/1.1} \rceil$. Increasing $N_k$ further accelerates the reduction in infeasibility. More generally, the growth schedule of $N_k$ in the randomized feasibility updates provides a trade-off between feasibility accuracy and computational cost (time and memory). Empirically, larger values of $N_k$ lead to faster reductions in the infeasibility gap.

## 7. Conclusion

In this work, we study a constrained optimization problem involving the minimization of a convex objective function subject to the intersection of many sublevel sets of convex functions. We employ a randomized feasibility algorithm to bypass exact projection steps and combine it with gradient-based updates to minimize the objective function. We establish adaptive stepsize schemes and obtain linear convergence up to a threshold for strongly convex functions. Moreover, we obtain an $O(1/\sqrt{T})$ convergence rate for convex, possibly nonsmooth objectives using fully line-search-free and parameter-free step sizes. We further study random sampling strategies for the number $N_k$ of constraint samples based on general distributions and provide corresponding theoretical guarantees. Finally, we demonstrate the effectiveness of the proposed algorithms through simulations on QCQP, SVM, and constrained logistic regression problems with group fairness constraints.

## Acknowledgements

We are grateful to the reviewers for invaluable comments and insights that helped us improve this manuscript. This work is partially supported by the NSF award CIF 2134256.

## Impact Statement

"This paper presents work whose goal is to advance the field of Machine Learning. There are many potential societal consequences of our work, none which we feel must be specifically highlighted here."

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

## A. Strong Slater Condition: Global Error Bound and Bounds on Lagrangian Dual Multipliers

We connect the global error bound $\gamma$ with the optimal dual multipliers, which is not readily available in the existing literature. However, we do not claim any novel contribution rather than just providing a simple direct connection. Here, we assuming that the functions $g_i$ are convex over $\mathbb{R}^n$, $Y$ is convex and closed, and $X \cap Y \neq \emptyset$ (Assumption 2.1 holds). Recall that $X = \cap_{i=1}^m \{x \in \mathbb{R}^n \mid g_i(x) \leq 0\}$, which is equivalently given by $X = \{x \in \mathbb{R}^n \mid G(x) \leq 0\}$ with $G(x) = \max_{1 \leq i \leq m} g_i(x)$.

Under the Strong Slater condition, requiring that $0 \notin \mathrm{cl}(\partial h(h^{-1}(0)))$ where $\mathrm{cl}(A)$ denotes the closure of a set $A$, the following global error bound is shown in (Lewis & Pang, 1998) [Corollary 2(a,b)] for a proper closed convex function $h$:

$$\mathrm{dist}(x, X \cap Y) \leq \gamma h^+(x) \quad \text{for all } x \in \mathbb{R}^n, \tag{4}$$

with the constant $\gamma$ given by

$$\gamma = \frac{1}{\mathrm{dist}(0, \mathrm{cl}(\partial h(h^{-1}(0))))}. \tag{5}$$

Note that $\gamma$ is finite since $0 \notin \mathrm{cl}(\partial h(h^{-1}(0)))$. Applying this result to the function $h(x) = G(x) + \delta_Y(x)$ for all $x$ with $G(x) = \max_{1 \leq i \leq m} g_i(x)$, and noting that $h(x) = +\infty$ for all $x \notin Y$, under the Strong Slater condition $0 \notin \mathrm{cl}(\partial h(h^{-1}(0)))$, we have

$$\mathrm{dist}(x, X \cap Y) \leq \gamma G^+(x) \quad \text{for all } x \in Y,$$

with the constant $\gamma$ given in (5).

A different approach that leads to a necessary and sufficient condition for the existence of an error bound has been used in (Bertsekas, 1999) [Proposition 3]. Proposition 3 of (Bertsekas, 1999) (applied to a single constraint inequality $G(x) = \max_{1 \leq i \leq m} g_i(x)$) states that: The optimal value $d(x)$ of the projection problem

$$\begin{aligned} \text{minimize} \quad & \|x - y\| \\ \text{subject to} \quad & y \in Y, \ G(y) \leq 0, \end{aligned} \tag{6}$$

where $Y$ is a convex subset of $\mathbb{R}^n$ and the function $G$ is convex, satisfies the Hoffman-like bound: for some constant $\bar{c} > 0$,

$$d(x) \leq \bar{c} G^+(x) \quad \text{for all } x \in Y$$

if and only if the projection problem (6) has a Lagrange multiplier $\mu^*(x)$ such that the set $\{\mu^*(x) \mid x \in Y\}$ is bounded by $\bar{c}$, i.e.,

$$\mu^*(x) \leq \bar{c} \quad \text{for all } x \in Y. \tag{7}$$

In view of the definition of $d(x)$, we have $d(x) = \mathrm{dist}(x, X \cap Y)$, and the preceding result gives a necessary and sufficient condition for the relation

$$\mathrm{dist}(x, X \cap Y) \leq \bar{c} G^+(x) \quad \text{for all } x \in Y. \tag{8}$$

to hold.

In order to explicitly relate $\gamma$ in relation (5) with $\bar{c}$ in relation (7), consider the projection problem of the following form:

$$\begin{aligned} \text{minimize} \quad & \|x - y\| \\ \text{subject to} \quad & h(y) \leq 0, \end{aligned} \tag{9}$$

where $h(x) = G(x) + \delta_Y(x)$ for all $x \in \mathbb{R}^n$. The projection problem (9) always has a unique solution - the projection $\Pi_{X \cap Y}[x]$ of $x$ on the convex closed set $X \cap Y = \{z \in \mathbb{R}^n \mid h(z) \leq 0\}$, which we denote by $y_x^*$.

Note that $h^+(x) = +\infty$ for $x \notin Y$, while $\mathrm{dist}(x, X \cap Y) = 0$ for $x \in X \cap Y$. In these two cases relation (4) would hold for any $\gamma > 0$. Thus, it is sufficient to consider the projection problem (9) for the case when $x \in Y$, $x \notin X \cap Y$. In this case, the projection problem (9) has a nonzero optimal value, i.e.,

$$\mathrm{dist}(x, X \cap Y) = \|x - y_x^*\| > 0 \quad \text{for all } x \in Y, \ x \notin X \cap Y.$$

Assume that the Lagrangian dual problem of the projection problem (9) has a nonempty set $\mathcal{M}_x$ of dual optimal multipliers for any $x \in Y$, $x \notin X \cap Y$. This will hold for example when the Slater condition holds i.e., $h(\bar{x}) < 0$ for some $\bar{x} \in \mathbb{R}^n$. The Lagrangian function associated with the projection problem (9) is given by

$$\mathcal{L}_x(y, \mu) = \|x - y\| + \mu h(y) \qquad y \in \mathbb{R}^n, \ \mu \geq 0.$$

By the KKT conditions, the optimal primal point and any dual optimal multiplier $\mu_x^* \in \mathcal{M}_x$ satisfy the following relations:

$$h(y_x^*) \leq 0, \qquad \mu_x^* \geq 0, \qquad \frac{x - y_x^*}{\|x - y_x^*\|} + \mu_x^* s_h(y_x^*) = 0, \qquad \mu_x^* h(y_x^*) = 0, \tag{10}$$

where $s_h(y_x^*)$ is a subgradient of $h$ at the point $y_x^*$. From the first equality in (10) we obtain

$$\mu_x^* s_h(y_x^*) = \frac{y_x^* - x}{\|x - y_x^*\|} \qquad \Longrightarrow \qquad \mu_x^* \|s_h(y_x^*)\| = 1. \tag{11}$$

Thus, $\mu_x^*$ must be positive, which by the last condition in (10) implies that $h(y_x^*) = 0$. Hence,

$$y_x^* \in h^{-1}(0) \qquad \text{for all } x \in Y, \ x \notin X \cap Y. \tag{12}$$

Relation (11) implies that

$$\mu_x^* = \frac{1}{\|s_h(y_x^*)\|} \qquad \text{for all } \mu_x^* \in \mathcal{M}_x \text{ and all } x \in Y, x \notin X \cap Y. \tag{13}$$

When the Strong Slater condition $0 \notin \mathrm{cl}(\partial h(h^{-1}(0)))$ holds, we have that $0 \notin (\partial h(h^{-1}(0)))$, which by convexity of $h$ implies that the Slater condition holds, i.e., $h(\bar{x}) < 0$ for some $\bar{x} \in \mathbb{R}^n$. Moreover, we have

$$\|s_h(z)\| \geq \mathrm{dist}(0, \mathrm{cl}(\partial h(h^{-1}(0)))) \qquad \text{for all } s_h(z) \in \partial h(z) \text{ and for all } z \text{ satisfying } h(z) = 0. \tag{14}$$

From relations (12)–(14) and the definition of $\gamma$ in (5) it follows that

$$\mu_x^* \leq \gamma \qquad \text{for all } \mu_x^* \in \mathcal{M}_x \text{ and all } x \in Y, x \notin X \cap Y. \tag{15}$$

Next, we consider the optimal Lagrangian dual multipliers for the projection problem (9) for the case when $x \in X \cap Y$. We note that in this case $y_x^* = x$, and the Lagrangian optimality condition (the first equality in (10)) takes the following form:

$$v + \mu_x^* s_h(y_x^*) = 0 \qquad \text{for some } v \in \mathbb{R}^n \text{ with } \|v\| \leq 1,$$

which comes from the fact that $\|x - y\|$ is not differentiable with respect to $y$ at the point $y = x$, but its subdifferential set at the point $y = x$ is given by the unit ball. The preceding condition can be satisfied with $v = 0$ and $\mu_x^* = 0$, and the other conditions in (10) are also satisfied with $\mu_x^* = 0$ and $y_x = x$. Therefore, when $x \in X \cap Y$, we can choose $\mu_x^* = 0$ as an optimal Lagrangian multiplier. With this choice of multipliers for the case when $x \in X \cap Y$ and relation (15), the upper bound for the multipliers in relation (7) holds with $\bar{c} = \gamma$.

## B. Some Auxiliary Results

### B.1. General Gradient Descent Lemma

**Lemma B.1.** *Let $Z$ be a closed convex set, and let $f(\cdot)$ be a continuously differentiable function. Consider an update of the following form:*

$$w = \Pi_Z[x - \alpha \nabla f(x)],$$

*where $\alpha > 0$ and $x \in \mathbb{R}^n$. If the function $f(\cdot)$ is strongly convex and has Lipschitz continuous gradients (Assumptions 4.1 and 4.2 hold), then we have for all $z \in Z$,*

$$\|w - z\|^2 \leq (1 - \alpha\mu)\|x - z\|^2 - (1 - \alpha L)\|w - x\|^2 - 2\alpha(f(w) - f(z)).$$

*Proof.* From the definition of $w$ and the non-expansiveness property of the projection, we have for any $z \in Z$,

$$\|w - z\|^2 = \|\Pi_Z[x - \alpha \nabla f(x)] - z\|^2$$
$$\leq \|x - \alpha \nabla f(x) - z\|^2 - \|w - (x - \alpha \nabla f(x))\|^2$$
$$= \|x - z\|^2 - \|w - x\|^2 - 2\alpha \langle \nabla f(x), x - z \rangle - 2\alpha \langle \nabla f(x), w - x \rangle. \tag{16}$$

We next bound the inner product terms on the right hand side of relation (16). By the strong convexity of $f(\cdot)$ (Assumption 4.2), we have

$$-2\alpha \langle \nabla f(x), x - z \rangle \leq -2\alpha(f(x) - f(z)) - \alpha \mu \|x - z\|^2. \tag{17}$$

Using the Lipschitz continuity of $\nabla f(\cdot)$ (Assumption 4.1), the last term on the right hand side of relation (16) can be bounded as follows:

$$-2\alpha \langle \nabla f(x), w - x \rangle \leq 2\alpha(f(x) - f(w)) + \alpha_k L \|w - x\|^2. \tag{18}$$

Combining the estimates in relations (17) and (18) with relation (16), we obtain for any $z \in Z$,

$$\|w - z\|^2 \leq (1 - \alpha \mu)\|x - z\|^2 - (1 - \alpha L)\|w - x\|^2 - 2\alpha(f(w) - f(z)),$$

thus showing the stated relation. **Q.E.D.**

## B.2. Some results on sequences

**Lemma B.2.** *Let $\{r_k\}$ be a non-negative scalar sequence and $\{s_k\}$ be a scalar sequence such that the following relation holds for some $p \geq 0$ and for all $k \geq 1$,*

$$r_{k+1} + s_{k+1} \leq pr_k.$$

*Then, we have for all $k \geq 1$,*

$$r_{k+1} + \sum_{t=2}^{k+1} p^{k+1-t} s_t \leq p^k r_1.$$

*Proof.* The proof is by the mathematical induction on $k$. For $k = 1$, the stated relation reduces to

$$r_2 + s_2 \leq pr_1,$$

which holds in view of the given recursive relation $r_{k+1} + s_{k+1} \leq pr_k$ for $k = 1$. Now, assume that for a given $k$, the relation

$$r_{k+1} + \sum_{t=2}^{k+1} p^{k+1-t} s_t \leq p^k r_1$$

is valid. Consider the case of $k + 1$. By the given recursive relation for $r_{k+2} + s_{k+2}$, we have

$$r_{k+2} + s_{k+2} \leq pr_{k+1}.$$

By adding $p \sum_{t=2}^{k+1} p^{k+1-t} s_t$ to both sides of the preceding inequality, we obtain

$$r_{k+2} + s_{k+2} + p \sum_{t=2}^{k+1} p^{k+1-t} s_t \leq p \left( r_{k+1} + \sum_{t=2}^{k+1} p^{k+1-t} s_t \right) \leq p \cdot p^k r_1 = p^{k+1} r_1.$$

Since $s_{k+2} + p \sum_{t=2}^{k+1} p^{k+1-t} s_t = \sum_{t=2}^{k+2} p^{k+2-t} s_t$, it follows that

$$r_{k+2} + \sum_{t=2}^{k+2} p^{k+2-t} s_t \leq p^{k+1} r_1,$$

where the last inequality follows by the inductive hypothesis. Thus, the stated relation also holds for $k + 1$. **Q.E.D.**

**Lemma B.3.** *Let $\{s_i\}_{i=0}^{T+1}$ be a sequence of positive non-decreasing quantities. Then, for any $T \geq 1$,*

$$\min_{1 \leq k \leq T} \frac{s_{k+1}}{\sum_{i=1}^{k} s_i} \leq \frac{1}{T} \left( \frac{s_{T+1}}{s_0} \right)^{\frac{1}{T}} \ln \left( \frac{e s_{T+1}}{s_0} \right).$$

*Proof.* The proof of this result is established in (Liu & Zhou, 2023) [Lemma 30] and (Moshtaghifar et al., 2025) [Lemma A.1]. In our case, some of the indices of the sequences we work with will be changed. Hence, accordingly the result will be modified. Let us consider

$$A_k := \frac{1}{s_{k+1}} \sum_{i=1}^{k} s_i \quad \text{for all } k \geq 1 \quad \text{and} \quad A_0 = 0.$$

Then, we can see that

$$A_{k+1} s_{k+2} - A_k s_{k+1} = \sum_{i=1}^{k+1} s_i - \sum_{i=1}^{k} s_i = s_{k+1}.$$

Diving both sides by $s_{k+2}$, and after some algebraic manipulations, we obtain

$$\frac{s_{k+1}}{s_{k+2}} = A_{k+1} - A_k \frac{s_{k+1}}{s_{k+2}} = A_{k+1} - A_k + A_k \left( 1 - \frac{s_{k+1}}{s_{k+2}} \right).$$

Summing the preceding relation from $k = 0$ to $T - 1$ and denoting $S_T = \sum_{k=0}^{T-1} \frac{s_{k+1}}{s_{k+2}}$, we obtain

$$S_T = A_T - A_0 + \sum_{k=0}^{T-1} A_k \left( 1 - \frac{s_{k+1}}{s_{k+2}} \right) = A_T - A_0 + A_0 \left( 1 - \frac{s_1}{s_2} \right) + \sum_{k=1}^{T-1} A_k \left( 1 - \frac{s_{k+1}}{s_{k+2}} \right).$$

Since, $A_0 = 0$, we can further simplify the preceding expression as

$$S_T = A_T + \sum_{k=1}^{T-1} A_k \left( 1 - \frac{s_{k+1}}{s_{k+2}} \right) \leq \bar{A}_T + \bar{A}_T \sum_{k=0}^{T-1} \left( 1 - \frac{s_{k+1}}{s_{k+2}} \right) = \bar{A}_T (1 + T - S_T),$$

where $\bar{A}_T = \max_{1 \leq k \leq T} A_k$. The preceding relation implies

$$\bar{A}_T \geq \frac{S_T}{1 + T - S_T}. \tag{19}$$

By the Arithmetic Mean – Geometric Mean (AM-GM) Inequality, we obtain

$$\frac{S_T}{T} = \frac{1}{T} \sum_{k=0}^{T-1} \frac{s_{k+1}}{s_{k+2}} \geq \left( \prod_{k=0}^{T-1} \frac{s_{k+1}}{s_{k+2}} \right)^{\frac{1}{T}} = \left( \frac{s_1}{s_{T+1}} \right)^{\frac{1}{T}}.$$

Since $s_0 \leq s_1$, hence we obtain

$$S_T \geq T \left( \frac{s_0}{s_{T+1}} \right)^{\frac{1}{T}}.$$

Let us denote $\zeta_T = \left( \frac{s_0}{s_{T+1}} \right)^{\frac{1}{T}}$. Then substituting the preceding relation back to relation (19) yields

$$\bar{A}_T \geq \frac{T \zeta_T}{1 + T(1 - \zeta_T)}.$$

We note that for $\zeta_T \in (0, 1]$, we obtain $1 - \zeta_T \leq -\ln(\zeta_T) = \ln\left(\frac{1}{\zeta_T}\right)$ for all $T \geq 1$. Substituting this relation in the preceding expression, we obtain

$$\bar{A}_T \geq \frac{T\zeta_T}{1 + \ln\left(\frac{1}{\zeta_T^T}\right)} = \frac{T\zeta_T}{\ln\left(\frac{e}{\zeta_T^T}\right)}.$$

We note that $1/\bar{A}_T = \min_{1 \leq k \leq T} \{1/A_k\} = \min_{1 \leq k \leq T} \frac{s_{k+1}}{\sum_{i=1}^k s_i}$. Hence, using this relation to the preceding one, we obtain the desired relation. **Q.E.D.**

**Lemma B.4.** *Let $\{p_k\}_{k \geq 0}$ be the sequences of non-decreasing numbers as in Algorithm 4. Then, the following statements are valid:*

*(a) For $p_0 = 0$ and $p_1 > 0$, we have*

$$\sum_{k=1}^T \frac{p_k - p_{k-1}}{p_k \left(\ln\left(\frac{ep_k}{p_1}\right)\right)^2} \leq 2.$$

*(b) For $p_0 > 0$, we have*

$$\sum_{k=1}^T \frac{p_k - p_{k-1}}{p_k \left(\ln\left(\frac{ep_k}{p_0}\right)\right)^2} \leq 1.$$

*Proof.* The proof for both the parts follows along the similar lines as (Ivgi et al., 2023) [Lemma 6] with some minor modifications.

*Part (a):* We have

$$\sum_{k=1}^T \frac{p_k - p_{k-1}}{p_k \left(\ln\left(\frac{ep_k}{p_1}\right)\right)^2} = \frac{p_1 - p_0}{p_1(\ln e)^2} + \sum_{k=2}^T \frac{\frac{p_k}{p_1} - \frac{p_{k-1}}{p_1}}{\frac{p_k}{p_1}\left(\ln\left(\frac{ep_k}{p_1}\right)\right)^2} \leq 1 + \sum_{k=2}^T \int_{\frac{p_{k-1}}{p_1}}^{\frac{p_k}{p_1}} \frac{dz}{z(1 + \ln z)^2} \leq 1 + \int_1^{\frac{p_T}{p_1}} \frac{dz}{z(1 + \ln z)^2}.$$

Upon integrating the last quantity on the right hand side, we can finally obtain

$$\sum_{k=1}^T \frac{p_k - p_{k-1}}{p_k \left(\ln\left(\frac{ep_k}{p_1}\right)\right)^2} \leq 1 + 1 - \frac{1}{1 + \ln\left(\frac{p_T}{p_1}\right)} \leq 2,$$

which concludes the proof for part (a).

*Part (b):* The proof for part (b) follows along the similar lines as above. We have

$$\sum_{k=1}^T \frac{p_k - p_{k-1}}{p_k \left(\ln\left(\frac{ep_k}{p_0}\right)\right)^2} = \sum_{k=1}^T \frac{\frac{p_k}{p_0} - \frac{p_{k-1}}{p_0}}{\frac{p_k}{p_0}\left(\ln\left(\frac{ep_k}{p_0}\right)\right)^2} \leq \sum_{k=1}^T \int_{\frac{p_{k-1}}{p_0}}^{\frac{p_k}{p_0}} \frac{dz}{z(1 + \ln z)^2} \leq \int_1^{\frac{p_T}{p_0}} \frac{dz}{z(1 + \ln z)^2}.$$

Integrating the preceding relation, we obtain

$$\sum_{k=1}^T \frac{p_k - p_{k-1}}{p_k \left(\ln\left(\frac{ep_k}{p_0}\right)\right)^2} \leq 1 - \frac{1}{1 + \ln\left(\frac{p_T}{p_0}\right)} \leq 1.$$

This concludes the proof. **Q.E.D.**

## C. Missing Proofs of Section 3

### C.1. Proof of Lemma 3.1

*Proof. Part (a):* We take any arbitrary point $x \in X \cap Y$. Then by the randomized feasibility update, we see

$$\|z_k^i - x\|^2 = \left\| \Pi_Y \left[ z_k^{i-1} - \beta \frac{g_{\omega_k^i}^+(z_k^{i-1})}{\|d_k^i\|^2} d_k^i \right] - x \right\|^2$$

$$\leq \|z_k^{i-1} - x\|^2 - 2\beta \frac{g_{\omega_k^i}^+(z_k^{i-1})}{\|d_k^i\|^2} \langle d_k^i, z_k^{i-1} - x \rangle + \beta^2 \frac{\left( g_{\omega_k^i}^+(z_k^{i-1}) \right)^2}{\|d_k^i\|^2}.$$

Note that $d_k^i \in \partial g_{\omega_k^i}^+(z_k^{i-1})$. Hence by the convexity of the function $g_i(\cdot)$ for all $i \in \{1, \ldots, m\}$ (Assumption 2.1), we obtain the relation

$$\langle d_k^i, z_k^{i-1} - x \rangle \geq g_{\omega_k^i}(z_k^{i-1}) - g_{\omega_k^i}(x) \geq g_{\omega_k^i}^+(z_k^{i-1}).$$

Substituting the preceding relation back to the main expression, we obtain

$$\|z_k^i - x\|^2 \leq \|z_k^{i-1} - x\|^2 - \beta(2 - \beta) \frac{\left( g_{\omega_k^i}^+(z_k^{i-1}) \right)^2}{\|d_k^i\|^2}. \tag{20}$$

Dropping the last non-positive term in relation (20) and summing it from $i = 1, \ldots, N_k$, and noting that $z_k^0 = v_k$ and $z_k^{N_k} = x_k$, we obtain the first relation of the lemma.

*Part (b):* Next, if the gradient $\|d_k^i\| \leq M_g$, then using this bound in relation (20), we obtain

$$\|z_k^i - x\|^2 \leq \|z_k^{i-1} - x\|^2 - \beta(2 - \beta) \frac{\left( g_{\omega_k^i}^+(z_k^{i-1}) \right)^2}{M_g^2}. \tag{21}$$

Taking minimum with respect to $x \in X \cap Y$ on both sides of the preceding relation, we obtain

$$\text{dist}^2(z_k^i, X \cap Y) \leq \text{dist}^2(z_k^{i-1}, X \cap Y) - \beta(2 - \beta) \frac{\left( g_{\omega_k^i}^+(z_k^{i-1}) \right)^2}{M_g^2}.$$

Denote the sigma algebra $\widetilde{\mathcal{F}}_{k-1} := \mathcal{F}_{k-1} \cup \{N_k\}$. Taking conditional expectation with respect to the sigma algebra $\widetilde{\mathcal{F}}_{k-1}$ on both sides of the preceding relation, we obtain almost surely for all $k \geq 1$, and all $i = 1, \ldots, N_k$

$$\mathbb{E}\left[ \text{dist}^2(z_k^i, X \cap Y) \mid \widetilde{\mathcal{F}}_{k-1} \right] \leq \mathbb{E}\left[ \text{dist}^2(z_k^{i-1}, X \cap Y) \mid \widetilde{\mathcal{F}}_{k-1} \right] - \frac{\beta(2 - \beta)}{M_g^2} \mathbb{E}\left[ \left( g_{\omega_k^i}^+(z_k^{i-1}) \right)^2 \mid \widetilde{\mathcal{F}}_{k-1} \right]. \tag{22}$$

Applying the law of iterated expectations and Assumption 2.2 to the last quantity on the right hand side of relation (22) yields almost surely for all $i = 1, \ldots, N_k$ and $k \geq 1$,

$$\mathbb{E}\left[ \left( g_{\omega_k^i}^+(z_k^{i-1}) \right)^2 \mid \widetilde{\mathcal{F}}_{k-1} \right] = \mathbb{E}\left[ \mathbb{E}\left[ \left( g_{\omega_k^i}^+(z_k^{i-1}) \right)^2 \mid \widetilde{\mathcal{F}}_{k-1} \cup \{z_k^{i-1}\} \right] \mid \widetilde{\mathcal{F}}_{k-1} \right] \geq \frac{1}{c} \mathbb{E}\left[ \text{dist}^2(z_k^{i-1}, X \cap Y) \mid \widetilde{\mathcal{F}}_{k-1} \right]. \tag{23}$$

Substituting the lower estimate from the preceding relation back to relation (22), we obtain almost surely,

$$\mathbb{E}\left[ \text{dist}^2(z_k^i, X \cap Y) \mid \widetilde{\mathcal{F}}_{k-1} \right] \leq \left( 1 - \frac{\beta(2 - \beta)}{cM_g^2} \right) \mathbb{E}\left[ \text{dist}^2(z_k^{i-1}, X \cap Y) \mid \widetilde{\mathcal{F}}_{k-1} \right]. \tag{24}$$

Let us denote the constant $q$ as

$$q = \frac{\beta(2 - \beta)}{cM_g^2}. \tag{25}$$

The constant $q$ is always positive since $\beta \in (0, 2)$. We would want $q \leq 1$. Note that when $q = 1$, then the quantity $\mathbb{E}\left[\text{dist}^2(z_k^i, X \cap Y) \mid \widetilde{\mathcal{F}}_{k-1}\right] = 0$ for all $i = 1, \ldots, N_k$. This condition would imply we are already on the set which might be unlikely. Hence, let us assume that $q < 1$. Using the fact that $x_k = z_k^{N_k}$, we can recursively write expression (24) as

$$\mathbb{E}\left[\text{dist}^2(x_k, X \cap Y) \mid \widetilde{\mathcal{F}}_{k-1}\right] \leq (1-q)^{N_k - i + 1} \mathbb{E}\left[\text{dist}^2(z_k^{i-1}, X \cap Y) \mid \widetilde{\mathcal{F}}_{k-1}\right] \quad \text{a.s.}$$

Using equation (23) to upper estimate the right hand side of the preceding relation, we obtain a.s.

$$\mathbb{E}\left[\text{dist}^2(x_k, X \cap Y) \mid \widetilde{\mathcal{F}}_{k-1}\right] \leq c\,(1-q)^{N_k - i + 1} \mathbb{E}\left[\left(g_{\omega_k^i}^+(z_k^{i-1})\right)^2 \mid \widetilde{\mathcal{F}}_{k-1}\right].$$

The above expression can be re-written yielding almost surely

$$\mathbb{E}\left[\left(g_{\omega_k^i}^+(z_k^{i-1})\right)^2 \mid \widetilde{\mathcal{F}}_{k-1}\right] \geq \frac{1}{c}\frac{1}{(1-q)^{N_k - i + 1}}\mathbb{E}\left[\text{dist}^2(x_k, X \cap Y) \mid \widetilde{\mathcal{F}}_{k-1}\right].$$

Summing both sides of the above expression for $i = 1, \ldots, N_k$, and noting that $\sum_{i=1}^{N_k} \frac{1}{(1-q)^{N_k - i + 1}} = \frac{1 - (1-q)^{N_k}}{q(1-q)^{N_k}}$, we obtain almost surely

$$\sum_{i=1}^{N_k}\mathbb{E}\left[\left(g_{\omega_k^i}^+(z_k^{i-1})\right)^2 \mid \widetilde{\mathcal{F}}_{k-1}\right] \geq \frac{1 - (1-q)^{N_k}}{cq(1-q)^{N_k}}\mathbb{E}\left[\text{dist}^2(x_k, X \cap Y) \mid \widetilde{\mathcal{F}}_{k-1}\right].$$

Using the definition of $q$ from relation (25), the preceding relation can be simplified almost surely as

$$\frac{\beta(2-\beta)}{M_g^2}\sum_{i=1}^{N_k}\mathbb{E}\left[\left(g_{\omega_k^i}^+(z_k^{i-1})\right)^2 \mid \widetilde{\mathcal{F}}_{k-1}\right] \geq ((1-q)^{-N_k} - 1)\mathbb{E}\left[\text{dist}^2(x_k, X \cap Y) \mid \widetilde{\mathcal{F}}_{k-1}\right]. \tag{26}$$

Next, we take conditional expectation on both sides of relation (21) given the sigma algebra $\widetilde{\mathcal{F}}_{k-1}$, sum the expression over $i = 1, \ldots, N_k$, and then apply relation (26) to upper estimate the last quantity on the right hand side to obtain almost surely the desired relation of the lemma in part (b). **Q.E.D.**

# D. Missing Proofs of Section 4

## D.1. Proof of Lemma 4.3

*Proof.* We consider Lemma B.1 and substitute $Z = Y$, $w = v_{k+1}$, $x = x_k$, $z = y$, and $\alpha = \alpha_k$ to obtain the desired result of the lemma. **Q.E.D.**

## D.2. Lemma on boundedness of iterates for Algorithm 2

**Lemma D.1.** *Let Assumptions 2.1, 4.1, and 4.2 hold, and assume that the step size satisfies $0 < \alpha_k \leq 1/L$ for all $k \geq 0$ and $\beta \in (0, 2)$. Moreover, assume that the random initial point is bounded by a constant, i.e., $\|x_0\| \leq M_0$ surely for some scalar $M_0 > 0$. Then, the iterates $\{x_k\}$, $\{v_k\}$ and $\{z_k^i, i = 1, \ldots, N_k, k \geq 1\}$ are surely bounded by a constant, i.e., for all $k \geq 1$,*

$$\|x_k - x^*\| \leq B_0 + \|x^*\|, \qquad \|v_k - x^*\| \leq B_0 + \|x^*\|,$$

$$\|z_k^i - x^*\| \leq B_0 + \|x^*\| \quad \text{for all } i = 1, \ldots, N_k - 1,$$

*where $x^*$ is the solution to problem (P), and $B_0 \geq M_0$ is such that $\|x\| \leq B_0$ for all $x$ that lie in the level set $\in \{u \in \mathbb{R}^n \mid f(u) \leq f(x^*)\}$.*

*Proof.* From the analysis of the first relation of Lemma 3.1, for any $\beta \in (0, 2)$, we obtain surely for all $x \in X$ and all $k \geq 1$,

$$\|z_k^i - x\| \leq \|z_k^{i-1} - x\| \quad \text{for } i = 1, \ldots, N_k, \tag{27}$$

$$\|x_k - x\| \leq \|v_k - x\|. \tag{28}$$

By the definition of the method, we have that $x_k = z_k^{N_k}$ and $v_k = z_k^0$ (see Algorithm 1). In view of these relations and the inequalities in relation (27), we can see that it suffices to show that the sequences $\{\|x_k - x\|\}$ and $\{\|v_k - x\|\}$ are surely bounded for some $x \in X$. Let $x^*$ be the unique solution to problem (P), which exists due to the strong convexity of $f(\cdot)$. We will prove that $\{\|x_k - x^*\|\}$ and $\{\|v_k - x^*\|\}$ are surely bounded using the mathematical induction on the iterate index $k$. To facilitate the mathematical induction, we will use Lemma 4.3 and relation (28).

Since $f(\cdot)$ is strongly convex, the (lower) level sets of $f(\cdot)$ are bounded. Specifically, the set

$$\bar{\mathcal{L}}(x^*) = \{x \in \mathbb{R}^n \mid f(x) \leq f(x^*)\}$$

is bounded. Let $B_0 > 0$ be such that

$$\|x\| \leq B_0 \qquad \text{for all } x \in \bar{\mathcal{L}}(x^*). \tag{29}$$

Also, let $B_0 \geq M_0$. Then, $\|x_0 - x^*\| \leq M_0 + \|x^*\| \leq B_0 + \|x^*\|$. Hence, for $k = 0$ we surely have $\|x_0 - x^*\| \leq B_0 + \|x^*\|$.

Assume now that for some $k \geq 1$, the iterates $x_k$ and $v_k$ are surely within the ball centered at $x^*$ with the radius $B_0 + \|x^*\|$, i.e.,

$$\|x_k - x^*\| \leq B_0 + \|x^*\|, \qquad \|v_k - x^*\| \leq B_0 + \|x^*\|.$$

We consider the iterates $v_{k+1}$ and $x_{k+1}$. By Lemma B.1, where we use $Z = Y$, $w = v_{k+1}$, $x = x_k$, $\alpha = \alpha_k$, and $z = x^* \in X \cap Y$ and the fact that $1 - \alpha_k L \geq 0$, we surely have that

$$\|v_{k+1} - x^*\|^2 \leq (1 - \alpha_k \mu)\|x_k - x^*\|^2 - 2\alpha_k(f(v_{k+1}) - f(x^*)). \tag{30}$$

The vector $v_{k+1}$ is random and we either have $f(v_{k+1}) > f(x^*)$ or $f(v_{k+1}) \leq f(x^*)$ (with corresponding probabilities). If $f(v_{k+1}) > f(x^*)$, then from relation (30) we see that

$$\|v_{k+1} - x^*\|^2 < (1 - \alpha_k \mu)\|x_k - x^*\|^2 \leq \|x_k - x^*\|^2.$$

Thus, by the inductive hypothesis, it follows that $\|v_{k+1} - x^*\| < B_0 + \|x^*\|$. On the other hand, if $f(v_{k+1}) \leq f(x^*)$, then $v_{k+1} \in \bar{\mathcal{L}}(x^*)$, implying by the boundedness of the level set $\bar{\mathcal{L}}(x^*)$ (see relation (29)) that $\|v_{k+1}\| \leq B_0$. Thus, $\|v_{k+1} - x^*\| \leq B_0 + \|x^*\|$. Hence, in any case, we surely have that

$$\|v_{k+1} - x^*\| \leq B_0 + \|x^*\|. \tag{31}$$

For the iterate $x_{k+1}$, by relation (28), where $x = x^* \in X$, we have that

$$\|x_{k+1} - x^*\| \leq \|v_{k+1} - x^*\|,$$

thus implying by relation (31) that surely $\|x_{k+1} - x^*\| \leq B_0 + \|x^*\|$. Therefore, we surely have $\|v_k - x^*\| \leq B_0 + \|x^*\|$ for all $k \geq 1$ and $\|x_k - x^*\| \leq B_0 + \|x^*\|$ for all $k \geq 0$. By relation (27), it follows that for all $k \geq 1$, we also surely have $\|z_k^i - x^*\| \leq B_0 + \|x^*\|$ for all $i = 1, \ldots, N_k$. **Q.E.D.**

As a consequence of Lemma D.1, we can state the next corollary.

**Corollary D.2.** *Under the assumptions of Lemma D.1, the subgradients $d_k^i$ of Algorithm 1 are surely bounded. Also, the gradients $\nabla f(x_k)$ along the iterates $x_k$ generated by Algorithm 2 are surely bounded, i.e., there exist (deterministic) scalars $M_g > 0$ and $M_f > 0$ such that $\|d_k^i\| \leq M_g$ surely for all $i = 1, \ldots, N_k$ and $k \geq 1$, and $\|\nabla f(x_k)\| \leq M_f$ surely for all $k \geq 0$.*

This subgradient boundedness follows from by (Bertsekas et al., 2003)[Proposition 4.2.3] and the fact that the sequence $\{z_k^i, i = 1 \ldots, N_k, k \geq 1\}$ is surely bounded. Lemma D.1 still holds when $\mu = 0$, i.e., $f$ is merely convex. But for this case, we need to assume that $f$ has bounded sublevel set, and the solution set is nonempty and apply Lemma D.1 with an arbitrary solution $x^*$. This intuition for bounding the sublevel set will be used later when analyzing the T-DoWS algorithm (cf. Algorithm 4) and proving the boundedness of its iterates.

### D.3. Auxiliary results on the function decay and the infeasibility gap

**Theorem D.3.** *Under Assumptions 2.1, 2.2, 4.1, and 4.2, and using a constant step size $\alpha_k = \alpha \in (0, \frac{1}{L}]$ in Algorithm 2, the following exponentially decaying upper bound holds for the weighted iterate $\bar{v}_k = \frac{\alpha\mu}{1-(1-\alpha\mu)^k} \sum_{t=2}^{k+1}(1-\alpha\mu)^{k+1-t}v_t$,*

$$f(\bar{v}_k) - f(x^*) \leq \mu(1-\alpha\mu)^k \frac{\|v_1 - x^*\|^2}{2(1-(1-\alpha\mu)^k)} \quad \text{surely for all } k \geq 1.$$

*The iterates $x_k$ obtained by Algorithm 1 to reduce the functional infeasibility gap satisfy*

$$\mathbb{E}[f(x_k) - f(x^*) \mid \mathcal{F}_{k-1} \cup \{N_k\}] \geq -M_f(1-q)^{\frac{N_k}{2}}(B_0 + \|x^*\|) \quad \text{almost surely for all } k \geq 1,$$

*while the weighted iterate $\bar{x}_k = \frac{\alpha\mu}{1-(1-\alpha\mu)^{k+1}} \sum_{t=1}^{k+1}(1-\alpha\mu)^{k+1-t}x_t$ satisfies*

$$\mathbb{E}[f(\bar{x}_k) - f(x^*)] \geq -M_f \max_{1 \leq t \leq k+1} \mathbb{E}\left[(1-q)^{\frac{N_t}{2}}\right](B_0 + \|x^*\|) \quad \text{for all } k \geq 1.$$

*Proof.* We use $y = x^*$ and a fixed step size $\alpha_k = \alpha > 0$ for all $k \geq 0$ in Lemma 4.3 and obtain

$$\|v_{k+1} - x^*\|^2 \leq (1-\alpha\mu)\|x_k - x^*\|^2 - (1-\alpha L)\|v_{k+1} - x_k\|^2 - 2\alpha(f(v_{k+1}) - f(x^*)).$$

Re-arranging the terms in the preceding relation, we obtain surely for all $k \geq 0$,

$$\|v_{k+1} - x^*\|^2 + 2\alpha(f(v_{k+1}) - f(x^*)) + (1-\alpha L)\|v_{k+1} - x_k\|^2 \leq (1-\alpha\mu)\|x_k - x^*\|^2.$$

By Lemma 3.1, with $x = x^*$, we surely have for all $k \geq 1$,

$$\|x_k - x^*\|^2 \leq \|v_k - x^*\|^2.$$

By combining the preceding two relations, we find that surely for all $k \geq 1$,

$$\|v_{k+1} - x^*\|^2 + 2\alpha(f(v_{k+1}) - f(x^*)) + (1-\alpha L)\|v_{k+1} - x_k\|^2 \leq (1-\alpha\mu)\|v_k - x^*\|^2.$$

We note that Lemma B.2 holds with $p = 1 - \alpha\mu \geq 0$, since $\alpha L \leq 1$ and $\mu \leq L$, and

$$r_k = \|v_k - x^*\|^2, \quad s_k = 2\alpha(f(v_k) - f(x^*)) + (1-\alpha L)\|v_k - x_{k-1}\|^2 \quad \text{for all } k \geq 1.$$

Thus, by Lemma B.2 we have

$$\|v_{k+1} - x^*\|^2 + 2\alpha \sum_{t=2}^{k+1} p^{k+1-t}(f(v_t) - f(x^*)) + (1-\alpha L)\sum_{t=2}^{k+1} p^{k+1-t}\|v_t - x_{t-1}\|^2 \leq p^k\|v_1 - x^*\|^2.$$

Defining

$$\bar{v}_k = \frac{1}{S_k}\sum_{t=2}^{k+1} p^{k+1-t}v_t \quad \text{and} \quad \bar{x}_{k-1} = \frac{1}{S_k}\sum_{t=2}^{k+1} p^{k+1-t}x_{t-1} \quad \text{with} \quad S_k = \sum_{t=2}^{k+1} p^{k+1-t} = \frac{1-p^k}{\alpha\mu}$$

for all $k \geq 1$ and using the convexity of $f(\cdot)$ and the norm, we find that

$$\|v_{k+1} - x^*\|^2 + 2\alpha S_k(f(\bar{v}_k) - f(x^*)) + (1-\alpha L)S_k\|\bar{v}_k - \bar{x}_{k-1}\|^2 \leq p^k\|v_1 - x^*\|^2.$$

Dropping the non-negative terms on the left hand side of the preceding relation, we obtain the first relation of the theorem. In order to derive the second relation of the theorem, we note that

$$f(x_k) - f(x^*) = [f(x_k) - f(\Pi_{X \cap Y}[x_k])] + [f(\Pi_{X \cap Y}[x_k]) - f(x^*)].$$

The quantity $f(\Pi_{X \cap Y}[x_k]) - f(x^*)$ is always non-negative and can be dropped while estimating the lower bound of the preceding relation. Hence, we obtain

$$f(x_k) - f(x^*) \geq f(x_k) - f(\Pi_{X \cap Y}[x_k]).$$

Application of convexity, Cauchy-Schwarz inequality, and then boundedness of the gradients by the constant $M_f$, we obtain

$$\begin{aligned}
f(x_k) - f(x^*) &\geq \langle \nabla f(\Pi_{X \cap Y}[x_k]), x_k - \Pi_{X \cap Y}[x_k] \rangle \\
&\geq -\|\nabla f(\Pi_{X \cap Y}[x_k])\| \operatorname{dist}(x_k, X \cap Y) \\
&\geq -M_f \operatorname{dist}(x_k, X \cap Y).
\end{aligned}$$

Taking conditional expectation on both sides of the preceding relation given the sigma algebra $\mathcal{F}_{k-1}$ and the sample size $N_k$, we obtain almost surely for all $k \geq 1$

$$\begin{aligned}
\mathbb{E}[f(x_k) - f(x^*) \mid \mathcal{F}_{k-1} \cup \{N_k\}] &\geq -M_f \mathbb{E}[\operatorname{dist}(x_k, X \cap Y) \mid \mathcal{F}_{k-1} \cup \{N_k\}] \\
&\geq -M_f (1-q)^{\frac{N_k}{2}} (B_0 + \|x^*\|),
\end{aligned}$$

where the final inequality is obtained using the last relation of Lemma 3.1 for $x = x^*$ and Lemma D.1.

In order to obtain the third relation of the theorem, we take the averaged iterate $\bar{x}_k = \frac{1}{S_{k+1}} \sum_{t=1}^{k+1} p^{k+1-t} x_t$ and take total expectation on the function values yielding

$$\mathbb{E}[f(\bar{x}_k) - f(x^*)] \geq -M_f \mathbb{E}[\operatorname{dist}(\bar{x}_k, X \cap Y)]. \tag{32}$$

Taking total expectation on the last relation of Lemma 3.1 and applying boundedness of the iterates from Lemma D.1, we obtain

$$\mathbb{E}[\operatorname{dist}(x_k, X \cap Y)] \leq \mathbb{E}\left[(1-q)^{\frac{N_k}{2}}\right] (B_0 + \|x^*\|) \qquad \text{for all } k \geq 1. \tag{33}$$

Next, we have

$$\begin{aligned}
S_{k+1} \mathbb{E}[\operatorname{dist}(\bar{x}_k, X \cap Y)] &= S_{k+1} \mathbb{E}[\|\bar{x}_k - \Pi_{X \cap Y}[\bar{x}_k]\|] \\
&\leq S_{k+1} \mathbb{E}\left[\left\| \frac{\sum_{t=1}^{k+1} p^{k+1-t} x_t - \sum_{t=1}^{k+1} p^{k+1-t} \Pi_{X \cap Y}[x_t]}{S_{k+1}} \right\|\right] \\
&\leq \sum_{t=1}^{k+1} p^{k+1-t} \mathbb{E}[\|x_t - \Pi_{X \cap Y}[x_t]\|] \\
&\leq \sum_{t=1}^{k+1} p^{k+1-t} \mathbb{E}\left[(1-q)^{\frac{N_t}{2}}\right] (B_0 + \|x^*\|) \\
&= \max_{1 \leq t \leq k+1} \mathbb{E}\left[(1-q)^{\frac{N_t}{2}}\right] (B_0 + \|x^*\|) S_{k+1}.
\end{aligned}$$

The first inequality of the preceding relation comes from the definition of $\bar{x}_k$ and using the fact that the projection of a point has the least distance compared to average of other points. The second inequality follows from Jensen's inequality, whereas the third inequality comes from substituting relation (33). Simplifying the preceding relation, we obtain

$$\mathbb{E}[\operatorname{dist}(\bar{x}_k, X \cap Y)] \leq \max_{1 \leq t \leq k+1} \mathbb{E}\left[(1-q)^{\frac{N_t}{2}}\right] (B_0 + \|x^*\|).$$

Using the preceding relation back in equation (32) yields the third relation of the theorem. **Q.E.D.**

### D.4. Proof of Theorem 4.4

*Proof.* We start our analysis from the relation of Lemma 4.3 and add and subtract $2\alpha_k f(x_k)$ to obtain surely

$$\|v_{k+1} - y\|^2 \leq (1 - \alpha_k \mu)\|x_k - y\|^2 - (1 - \alpha_k L)\|v_{k+1} - x_k\|^2 - 2\alpha_k (f(x_k) - f(y)) - 2\alpha_k (f(v_{k+1}) - f(x_k)).$$

The last quantity on the right hand side of the preceding inequality can be upper estimated using strong convexity, and then Cauchy-Schwarz inequality followed by Young's inequality as follows

$$2\alpha_k(f(x_k) - f(v_{k+1})) \leq \langle 2\alpha_k \nabla f(x_k), x_k - v_{k+1} \rangle - \alpha_k \mu \|v_{k+1} - x_k\|^2$$

$$\leq 2\alpha_k^2 \|\nabla f(x_k)\|^2 + \frac{1}{2}\|v_{k+1} - x_k\|^2 - \alpha_k \mu \|v_{k+1} - x_k\|^2.$$

Using the preceding relation back to the main first expression, we obtain

$$\|v_{k+1} - y\|^2 \leq (1 - \alpha_k \mu)\|x_k - y\|^2 - \left(\frac{1}{2} - \alpha_k(L - \mu)\right)\|v_{k+1} - x_k\|^2 - 2\alpha_k(f(x_k) - f(y)) + 2\alpha_k^2 \|\nabla f(x_k)\|^2.$$

With the selection of step size $\alpha_k \leq \min\left\{\frac{1}{2(L-\mu)}, \frac{1}{\mu}, \frac{1}{L}\right\} = \min\left\{\frac{1}{2(L-\mu)}, \frac{1}{L}\right\}$, the second quantity on the right hand side of the preceding relation can be dropped. Hence, we obtain

$$\|v_{k+1} - y\|^2 \leq (1 - \alpha_k \mu)\|x_k - y\|^2 - 2\alpha_k(f(x_k) - f(y)) + 2\alpha_k^2 \|\nabla f(x_k)\|^2. \tag{34}$$

We want to select $\alpha_k$ such that $2\alpha_k\|\nabla f(x_k)\|^2 = \epsilon$, which yields $\alpha_k = \frac{\epsilon}{2\|\nabla f(x_k)\|^2}$. By the consequence of Lemma D.1, we surely have $\|\nabla f(x_k)\| \leq M_f$. Hence, with $\alpha_k = \min\left\{\frac{1}{2(L-\mu)}, \frac{1}{L}, \frac{\epsilon}{2\|\nabla f(x_k)\|^2}\right\}$ and $\bar{\alpha} = \min\left\{\frac{1}{2(L-\mu)}, \frac{1}{L}, \frac{\epsilon}{2M_f^2}\right\}$, we can obtain $\alpha_k \geq \bar{\alpha}$ for all $k \geq 0$. Using the fact that $1 - \alpha_k \mu \leq 1 - \bar{\alpha}\mu$, relation (34) simplifies to

$$\|v_{k+1} - y\|^2 \leq (1 - \bar{\alpha}\mu)\|x_k - y\|^2 - 2\alpha_k(f(x_k) - f(y)) + \epsilon\alpha_k.$$

Next, we use $y = x^* \in X \cap Y$ which is the optimal point of the problem. By Lemma 3.1, we have surely $\|x_k - x^*\|^2 \leq \|v_k - x^*\|^2$. Using the equation in the preceding relation, we obtain

$$\|v_{k+1} - x^*\|^2 + 2\alpha_k(f(x_k) - f(x^*)) - \epsilon\alpha_k \leq (1 - \bar{\alpha}\mu)\|v_k - x^*\|^2.$$

Next, the proof follows along the similar lines of Theorem D.3. Denote

$$p = 1 - \bar{\alpha}\mu, \quad r_k = \|v_k - x^*\|^2, \quad s_k = 2\alpha_{k-1}(f(x_{k-1}) - f(x^*)) - \epsilon\alpha_{k-1} \quad \text{for all } k \geq 1.$$

Now defining

$$\bar{x}_k = \frac{1}{S_k}\sum_{t=2}^{k+1} p^{k+1-t}\alpha_{t-1}x_{t-1} = \frac{1}{S_k}\sum_{t=1}^{k} p^{k-t}\alpha_t x_t \quad \text{with} \quad S_k = \sum_{t=2}^{k+1}\alpha_{t-1}p^{k+1-t} = \sum_{t=1}^{k}\alpha_t p^{k-t},$$

and following Lemma B.2 and using convexity of the function $f$, we finally obtain surely

$$\|v_{k+1} - x^*\|^2 + 2S_k\left(f(\bar{x}_k) - f(x^*)\right) - \epsilon S_k \leq p^k\|v_1 - x^*\|^2.$$

Dropping the first non-negative term on the left hand side of the preceding relation, and simplifying things, we obtain surely

$$f(\bar{x}_k) - f(x^*) \leq \frac{p^k}{2S_k}\|v_1 - x^*\|^2 + \frac{\epsilon}{2}. \tag{35}$$

Since $\alpha_k \geq \bar{\alpha}$ for all $k \geq 1$, then

$$S_k = \sum_{t=1}^{k}\alpha_t p^{k-t} \geq \bar{\alpha}\sum_{t=1}^{k} p^{k-t} = \bar{\alpha}\frac{1 - p^k}{\bar{\alpha}\mu} = \frac{1 - p^k}{\mu}.$$

Hence using the preceding relation in relation (35) and using the bound of Lemma D.1 for the first term on the right hand side of relation (35), we obtain surely

$$f(\bar{x}_k) - f(x^*) \leq \frac{\mu p^k}{2(1 - p^k)}(B_0 + \|x^*\|)^2 + \frac{\epsilon}{2}.$$

Now to stay within the $\epsilon$ radius tolerance for the upper bound, i.e., $f(\bar{x}_k) - f(x^*) \leq \epsilon$ surely, we want

$$\frac{\mu p^k}{2(1 - p^k)}(B_0 + \|x^*\|)^2 \leq \frac{\epsilon}{2}.$$

Using $p = 1 - \bar{\alpha}\mu$, the preceding relation upon simplification yields the minimum number of iterations as

$$k \geq \frac{\ln\left(\frac{\mu(B_0 + \|x^*\|)^2}{\epsilon} + 1\right)}{\ln\left(\frac{1}{1 - \bar{\alpha}\mu}\right)}.$$

Next, we need to find the lower bound for $\mathbb{E}[f(\bar{x}_k) - f(x^*)]$ and the associated number of samples $N_k$ in order to achieve $\epsilon$ accuracy. Following the similar lines of proof as Theorem D.3, we see

$$\mathbb{E}[f(\bar{x}_k) - f(x^*)] \geq -M_f \mathbb{E}[\text{dist}(\bar{x}_k, X \cap Y)] \geq -(B_0 + \|x^*\|)M_f \max_{1 \leq t \leq k} \mathbb{E}\left[(1 - q)^{\frac{N_t}{2}}\right].$$

We would require

$$-(B_0 + \|x^*\|) \max_{1 \leq t \leq k} \mathbb{E}\left[(1 - q)^{\frac{N_t}{2}}\right] \geq -\epsilon,$$

which finally yields a relation that depends on the number of samples as follows

$$\mathbb{E}\left[(1 - q)^{\frac{N_t}{2}}\right] \leq \frac{\epsilon}{(B_0 + \|x^*\|)M_f} \qquad \text{for all } 1 \leq t \leq k.$$

This completes the proof. **Q.E.D.**

**Comments on Theorem 4.4:** Note that the number of samples $N_k$ used in Theorem 4.4 can be deterministic (constant or growing with a specific schedule) or random depending on the support $\mathcal{I}_k$ and distribution $\mathcal{D}_k$. Both $\mathcal{D}_k$ and $\mathcal{I}_k$ may vary with $k$, for example by sampling from a uniform or Poisson distribution, or by fixing the distribution across all $k \geq 0$. The computation of $\mathbb{E}[(1 - q)^{N_k/2}]$ for specific cases is provided in Appendix D.5. Theorem 4.4 establishes geometric convergence of the expected objective function values to an $\epsilon$ neighborhood, with an adaptive step size and dependence on the number of samples $N_k$. If $N_k$ is deterministic with $N_k \geq N$, then Theorem 4.4 implies that the minimum number $N$ of samples needed to reach an $\epsilon$-tolerance is bounded below by $N \geq \frac{2\ln\left(\frac{(B_0 + \|x^*\|)M_f}{\epsilon}\right)}{\ln\left(\frac{1}{1-q}\right)}$. On a bounded set, these constants can be estimated, and the constant $c$ appearing in $q$ can be estimated using arguments similar to (Bertsekas, 1999), which we presented in Appendix A, provided a Slater point is known. If $N_k$ is stochastic, for example $N_k \sim \text{Poisson}(\lambda_k)$, a corresponding lower bound on the parameter $\lambda_k$ can be derived similarly for all $k \geq 1$.

## D.5. Evaluation of $\mathbb{E}\left[(1 - q)^{\frac{N_k}{2}}\right]$ for different distributions

The number of samples $N_k \in \mathcal{I}_k \subset \mathbb{N}$ is considered a random variable, which is sampled from the distribution $\mathcal{D}_k$ that changes over time. For example, a simple case can be choosing a deterministic $N_k$ in the Randomized Feasibility updates (Algorithm 1). Then, the quantity

$$\mathbb{E}\left[(1 - q)^{\frac{N_k}{2}}\right] = (1 - q)^{\frac{N_k}{2}},$$

which converges geometrically fast with respect to the number of samples $N_k$, which we can grow with time $k$ (say in the order of $N + \log(k)$ or keep it constant to $N$). Choosing a large number $N$ might be computationally expensive. Alternatively, for example, we can consider $N_k$ with a uniform sampling from the set $\mathcal{I}_k = [0, \lceil \log(k) \rceil]$, which can be efficient in terms of maintaining a low complexity for Algorithm 1).

Here, we show how to compute the quantity $\mathbb{E}\left[(1 - q)^{\frac{N_k}{2}}\right]$ for several distributions, such as, Poisson, Uniform, and Binomial. Other distributions can also be analyzed similarly.

### D.5.1. POISSON DISTRIBUTION

Consider the number of samples $N_k \sim \text{Poisson}(\lambda_k)$, where $\lambda_k > 0$ is its mean. Hence, the Probability Mass Function (PMF) of the distribution can be presented as

$$\mathbb{P}(N_k = j) = \frac{e^{-\lambda_k} \lambda_k^j}{j!}, \quad j = 0, 1, 2, \ldots.$$

Then, we obtain

$$\begin{aligned}
\mathbb{E}\left[(1-q)^{N_k/2}\right] &= \sum_{j=0}^{\infty} (1-q)^{j/2} \cdot \frac{e^{-\lambda_k} \lambda_k^j}{j!} \\
&= e^{-\lambda_k} \sum_{j=0}^{\infty} \frac{\left(\lambda_k (1-q)^{1/2}\right)^j}{j!} \\
&= e^{-\lambda_k} \cdot e^{\lambda_k (1-q)^{1/2}} \\
&= e^{\left(-\lambda_k \left(1 - (1-q)^{1/2}\right)\right)}.
\end{aligned}$$

Hence, it can be seen that for a Poisson distribution with parameter $\lambda_k$, the expected value decays exponentially fast, with the rate of decay depending on $\lambda_k$. It is intuitive since a larger $\lambda_k$ would imply a larger number of samples $N_k$. We can choose to grow $\lambda_k$ in the order of $\log(k)$ or $\log(\log(k))$.

### D.5.2. UNIFORM DISTRIBUTION

Consider the sample size $N_k \sim \text{Uniform}(a_k, b_k)$, where $a_k$ and $b_k$ are integers with $a_k \geq 1$ and $b_k > a_k$ and with $b_k$ growing with $k$, for example, $b_k = \lceil a_k + \log(k) \rceil$. The PMF of such a distribution is

$$\mathbb{P}(N_k = j) = \frac{1}{b_k - a_k + 1}, \quad j = a_k, \ldots, b_k.$$

Then, using this PMF, we obtain

$$\begin{aligned}
\mathbb{E}\left[(1-q)^{N_k/2}\right] &= \frac{1}{b_k - a_k + 1} \sum_{j=a_k}^{b_k} (1-q)^{j/2} \\
&\leq \frac{1}{b_k - a_k + 1} \int_{a_k - 1}^{b_k} \left(\sqrt{1-q}\right)^j \, dj \\
&= \frac{1}{b_k - a_k + 1} \frac{\left(\sqrt{1-q}\right)^{b_k} - \left(\sqrt{1-q}\right)^{a_k - 1}}{\ln\left(\sqrt{1-q}\right)} \\
&= \frac{2}{b_k - a_k + 1} \frac{\left(\sqrt{1-q}\right)^{a_k - 1} - \left(\sqrt{1-q}\right)^{b_k}}{\ln\left(\frac{1}{1-q}\right)}.
\end{aligned}$$

Now, if $a_k = 1$ and $b_k = \lceil h(k) \rceil$, where $h(\cdot)$ is some monotonically increasing function, then we see $\mathbb{E}\left[(1-q)^{N_k/2}\right] \leq O\left(\frac{1}{h(k)}\right)$. The function $h(\cdot)$ is user specified and can be $\log(\cdot)$ or $\sqrt{(\cdot)}$, etc. This example shows that uniform sampling might destroy the geometric convergence of $\mathbb{E}\left[(1-q)^{\frac{N_k}{2}}\right]$ in terms of the sample size $N_k$.

### D.5.3. BINOMIAL DISTRIBUTION

Here we consider the random variable $N_k \sim \text{Binomial}(n_k, p)$, with the PMF given by

$$P(N_k = j) = \binom{n_k}{j} p^j (1-p)^{n-j}, \quad j = 0, \ldots, n_k.$$

Thus, we can compute the expectation of $(1-q)^{N_k/2}$ as follows

$$\mathbb{E}\left[(1-q)^{N_k/2}\right] = \sum_{j=0}^{n_k}(1-q)^{j/2}\binom{n_k}{j}p^j(1-p)^{n_k-j}$$

$$= \sum_{j=0}^{n_k}\binom{n_k}{j}\left(p\sqrt{1-q}\right)^j(1-p)^{n_k-j}.$$

Using the Binomial theorem of expansion in the preceding relation, we obtain

$$\mathbb{E}\left[(1-q)^{N_k/2}\right] = \left(p\sqrt{1-q}+1-p\right)^{n_k} = \left(1-p\left(1-\sqrt{1-q}\right)\right)^{n_k}.$$

Hence, the expectation decays geometrically fast when the number $n_k$ increases with $k$. The number $n_k$ can be a slowly increasing function of $k$ as discussed for the uniform case.

Note that at each iteration $k \geq 0$, the number $N_k$ of samples can either be fixed or drawn from any chosen probability distribution. This allows for sampling from different distributions at different iterations. Since the results in Theorems D.3, 4.4 and 5.3 involve the term $\max_k \mathbb{E}[(1-q)^{N_k/2}]$, if one wants to obtain a geometric rate for this case, it is advisable to choose distributions that ensure a geometric rate for all $k$ with respect to its own parameter, for example, Poisson and Binomial distributions. Also, one can enforce sample size of $\max\{N_k, N\}$ for some constant integer $N$, so that the convergence rate remains geometric with respect to $N$ up to an error tolerance even in the worst case, for any choice of distribution.

## E. Missing proofs of DoWS with Randomized Feasibility

### E.1. Proof of Theorem 5.3

*Proof.* First, we derive an upper bound on the gap between the function value evaluated at the averaged iterate and the optimal function value, following arguments similar to (Khaled et al., 2023). The distance norm square of the iterate $v_{k+1}$ from the optimal point $x^*$ can be expressed as

$$\|v_{k+1}-x^*\|^2 = \|\Pi_Y[x_k - \alpha_k s_f(x_k)] - x^*\|^2$$
$$\leq \|x_k-x^*\|^2 - 2\alpha_k\langle s_f(x_k), x_k - x^*\rangle + \alpha_k^2\|s_f(x_k)\|^2.$$

The preceding relation upon simplification yields

$$\langle s_f(x_k), x_k - x^*\rangle \leq \frac{\|x_k-x^*\|^2 - \|v_{k+1}-x^*\|^2}{2\alpha_k} + \frac{\alpha_k}{2}\|s_f(x_k)\|^2. \tag{36}$$

From Lemma 3.1(a) with $x = x^*$, we surely have $\|x_{k+1}-x^*\|^2 \leq \|v_{k+1}-x^*\|^2$. Moreover, the left hand side of relation (36) can be lower estimated using the convexity of the function $f$ as $\langle s_f(x_k), x_k - x^*\rangle \geq f(x_k) - f(x^*)$. Using these relations back to relation (36), we obtain surely

$$f(x_k)-f(x^*) \leq \frac{\|x_k-x^*\|^2 - \|x_{k+1}-x^*\|^2}{2\alpha_k} + \frac{\alpha_k}{2}\|s_f(x_k)\|^2.$$

Multiplying both sides of the preceding relation with $\bar{r}_k^2$ yields surely

$$\bar{r}_k^2\left(f(x_k)-f(x^*)\right) \leq \frac{\bar{r}_k^2}{2\alpha_k}\left[\|x_k-x^*\|^2 - \|x_{k+1}-x^*\|^2\right] + \frac{\alpha_k\bar{r}_k^2}{2}\|s_f(x_k)\|^2.$$

Denote $\hat{d}_k = \|x_k - x^*\|$. We take summation on both sides of the preceding relation for $k = 1$ to $T$ yielding

$$\sum_{k=1}^{T}\bar{r}_k^2\left(f(x_k)-f(x^*)\right) \leq \frac{1}{2}\sum_{k=1}^{T}\frac{\bar{r}_k^2}{\alpha_k}\left(\hat{d}_k^2 - \hat{d}_{k+1}^2\right) + \frac{1}{2}\sum_{k=1}^{T}\alpha_k\bar{r}_k^2\|s_f(x_k)\|^2. \tag{37}$$

We will analyze individual summation terms on the right hand side of relation (37) one after the other. For the first summation term, we see that

$$\sum_{k=1}^{T} \frac{\bar{r}_k^2}{\alpha_k} \left( \hat{d}_k^2 - \hat{d}_{k+1}^2 \right) = \sum_{k=1}^{T} \sqrt{p_k} \left( \hat{d}_k^2 - \hat{d}_{k+1}^2 \right)$$

$$= \sqrt{p_1} \hat{d}_1^2 - \sqrt{p_T} \hat{d}_{T+1}^2 + \sum_{k=2}^{T-1} \hat{d}_k^2 (\sqrt{p_k} - \sqrt{p_{k-1}}).$$

Let us define $\bar{d}_{T+1} = \max_{1 \le k \le T+1} \hat{d}_k$. Then, the preceding relation can be upper estimated as follows:

$$\sum_{k=1}^{T} \frac{\bar{r}_k^2}{\alpha_k} \left( \hat{d}_k^2 - \hat{d}_{k+1}^2 \right) \le \sqrt{p_1} \bar{d}_{T+1}^2 - \sqrt{p_T} \hat{d}_{T+1}^2 + \bar{d}_{T+1}^2 (\sqrt{p_{T-1}} - \sqrt{p_1})$$

$$\le \sqrt{p_T} (\bar{d}_{T+1}^2 - \hat{d}_{T+1}^2)$$

$$= \sqrt{p_T} (\bar{d}_{T+1} - \hat{d}_{T+1})(\bar{d}_{T+1} + \hat{d}_{T+1})$$

$$\le 2\sqrt{p_T} \bar{d}_{T+1} (\bar{d}_{T+1} - \hat{d}_{T+1}). \tag{38}$$

Next, we upper estimate the term $(\bar{d}_{T+1} - \hat{d}_{T+1})$. Let us consider $i^* = \operatorname{argmax}_{1 \le k \le T+1} \hat{d}_k$. Then $\bar{d}_{T+1} = \hat{d}_{i^*}$. Hence,

$$\hat{d}_{i^*} - \hat{d}_{T+1} = \|x_{i^*} - x^*\| - \|x_{T+1} - x^*\| \le \|x_{i^*} - x_{T+1}\| \le \|x_{i^*} - x_0\| + \|x_{T+1} - x_0\| \le 2\bar{r}_{T+1},$$

where the last inequality follows from the definition of $\bar{r}_k$ (which also implies that the sequence $\{\bar{r}_k\}$ is monotonically non-decreasing.) Applying the preceding upper estimate in relation (38), we obtain the following upper estimate to the first term on the right hand side of relation (37):

$$\frac{1}{2} \sum_{k=1}^{T} \frac{\bar{r}_k^2}{\alpha_k} \left( \hat{d}_k^2 - \hat{d}_{k+1}^2 \right) \le 2\sqrt{p_T} \bar{d}_{T+1} \bar{r}_{T+1}. \tag{39}$$

The second quantity on the right hand side of relation (37) can be analyzed as follows:

$$\sum_{k=1}^{T} \bar{r}_k^2 \alpha_k \|s_f(x_k)\|^2 = \sum_{k=1}^{T} \frac{\bar{r}_k^4}{\sqrt{p_k}} \|s_f(x_k)\|^2$$

$$\le \bar{r}_{T+1}^2 \sum_{k=1}^{T} \frac{\bar{r}_k^2}{\sqrt{p_k}} \|s_f(x_k)\|^2$$

$$\le \bar{r}_{T+1}^2 \sum_{k=1}^{T} \frac{p_k - p_{k-1}}{\sqrt{p_k}}$$

$$= \bar{r}_{T+1}^2 \sum_{k=1}^{T} \left( \sqrt{p_k} - \sqrt{p_{k-1}} \right) \underbrace{\left( \frac{\sqrt{p_k} + \sqrt{p_{k-1}}}{\sqrt{p_k}} \right)}_{\le 2}$$

$$\le 2\bar{r}_{T+1}^2 \left( \sqrt{p_T} - \sqrt{p_0} \right)$$

$$= 2\bar{r}_{T+1}^2 \sqrt{p_T}, \tag{40}$$

since $p_0 = 0$ based on the initialization. Substituting the estimates from relations (39) and (40) back in relation (37) yields

$$\sum_{k=1}^{T} \bar{r}_k^2 \left( f(x_k) - f(x^*) \right) \le \bar{r}_{T+1} (2\bar{d}_{T+1} + \bar{r}_{T+1}) \sqrt{p_T}. \tag{41}$$

Since $p_0 = 0$, the quantity $p_T$ can be upper estimated as

$$p_T = p_0 + \sum_{k=1}^{T} \bar{r}_k^2 \|s_f(x_k)\|^2 \le \bar{r}_{T+1}^2 \sum_{k=1}^{T} \|s_f(x_k)\|^2 \le \bar{r}_{T+1}^2 M_f^2 T.$$

Hence, the following holds

$$\sqrt{p_T} = \bar{r}_{T+1} M_f \sqrt{T}.$$

Combining the preceding relation with equation (41), we obtain surely for all $k \geq 0$,

$$\sum_{k=1}^{T} \bar{r}_k^2 \left( f(x_k) - f(x^*) \right) \leq \bar{r}_{T+1}^2 (2\bar{d}_{T+1} + \bar{r}_{T+1}) M_f \sqrt{T}. \tag{42}$$

Let

$$\bar{x}_T = \frac{\sum_{k=1}^{T} \bar{r}_k^2 x_k}{\sum_{k=1}^{T} \bar{r}_k^2}.$$

By the convexity of the function $f$, from relation (42) we obtain surely

$$f(\bar{x}_T) - f(x^*) \leq (2\bar{d}_{T+1} + \bar{r}_{T+1}) M_f \sqrt{T} \left( \frac{\bar{r}_{T+1}^2}{\sum_{i=1}^{T} \bar{r}_i^2} \right).$$

We apply Assumption 5.2 to upper estimate $2\bar{d}_{T+1} + \bar{r}_{T+1} \leq 3D$. We change the subscript index from $T$ to $k$ yielding surely for any $k \geq 1$,

$$f(\bar{x}_k) - f(x^*) \leq 3DM_f \sqrt{k} \left( \frac{\bar{r}_{k+1}^2}{\sum_{i=1}^{k} \bar{r}_i^2} \right).$$

Let us denote the index

$$\tau = \operatorname*{argmin}_{k \in \{1, \ldots, T\}} \frac{\bar{r}_{k+1}^2}{\sum_{i=1}^{k} \bar{r}_i^2}. \tag{43}$$

Then, the preceding relation yields surely for all $T \geq 1$,

$$f(\bar{x}_\tau) - f(x^*) \leq 3DM_f \sqrt{T} \left( \min_{1 \leq k \leq T} \frac{\bar{r}_{k+1}^2}{\sum_{i=1}^{k} \bar{r}_i^2} \right).$$

Using Lemma B.3 with $s_k = \bar{r}_k^2$ for all $k \geq 1$, and $s_0 = \bar{r}_0^2 = r^2$ to upper estimate the right hand side of the preceding relation, we obtain surely for all $T \geq 1$,

$$f(\bar{x}_\tau) - f(x^*) \leq 3DM_f \sqrt{T} \left( \frac{\left( \frac{\bar{r}_{T+1}^2}{r^2} \right)^{\frac{1}{T}} \ln \left( \frac{e\bar{r}_{T+1}^2}{r^2} \right)}{T} \right) = \left( \frac{3DM_f \left( \frac{\bar{r}_{T+1}}{r} \right)^{\frac{2}{T}} \ln \left( \frac{e\bar{r}_{T+1}^2}{r^2} \right)}{\sqrt{T}} \right).$$

Applying Assumption 5.2 to upper bound $\bar{r}_{T+1} \leq D$, we can obtain surely for all $T \geq 1$,

$$f(\bar{x}_\tau) - f(x^*) \leq \left( \frac{3DM_f \left( \frac{D}{r} \right)^{\frac{2}{T}} \ln \left( \frac{eD^2}{r^2} \right)}{\sqrt{T}} \right). \tag{44}$$

Now, the quantity $f(\bar{x}_\tau) - f(x^*)$ can be negative since the point $\bar{x}_\tau$ can be infeasible. Hence, we will next obtain the lower bound on the expected value of this quantity using the feasibility update relations.

We proceed along the similar lines of analysis as done towards the end of the proof of Theorems D.3 and 4.4. We note relation (32) and we apply Assumption 5.2 on the last expression of Lemma 3.1 yielding

$$\mathbb{E}[f(\bar{x}_\tau) - f(x^*)] \geq -M_f \mathbb{E}[\operatorname{dist}(\bar{x}_\tau, X \cap Y)] \quad \text{and} \quad \mathbb{E}[\operatorname{dist}(x_k, X \cap Y)] \leq \mathbb{E}\left[ (1-q)^{\frac{N_k}{2}} \right] D. \tag{45}$$

Hence,

$$
\begin{aligned}
\mathbb{E}[\mathrm{dist}(\bar{x}_\tau, X \cap Y)] &= \mathbb{E}[\|\bar{x}_\tau - \Pi_{X \cap Y}[\bar{x}_\tau]\|] \\
&= \mathbb{E}\left[\left\|\frac{\sum_{k=1}^{\tau} \bar{r}_k^2 (x_k - \Pi_{X \cap Y}[x_k])}{\sum_{k=1}^{\tau} \bar{r}_k^2}\right\|\right] \\
&\leq \mathbb{E}\left[\frac{\sum_{k=1}^{\tau} \bar{r}_k^2 \mathbb{E}[\|x_k - \Pi_{X \cap Y}[x_k]\| \mid \tau]}{\sum_{k=1}^{\tau} \bar{r}_k^2}\right] \\
&\leq \mathbb{E}\left[\frac{\sum_{k=1}^{\tau} \bar{r}_k^2 \mathbb{E}\left[(1-q)^{\frac{N_k}{2}}\right] D}{\sum_{k=1}^{\tau} \bar{r}_k^2}\right],
\end{aligned}
\tag{46}
$$

where the last relation follows from equation (45). From here, there are two ways to proceed with the proof.

*Case 1:* We can upper estimate the right hand side of relation (46) as

$$
\mathbb{E}[\mathrm{dist}(\bar{x}_\tau, X \cap Y)] \leq D\mathbb{E}\left[\max_{1 \leq k \leq \tau} (1-q)^{\frac{N_k}{2}}\right] \frac{\sum_{k=1}^{\tau} \bar{r}_k^2}{\sum_{k=1}^{\tau} \bar{r}_k^2} \leq D \max_{1 \leq k \leq T} \mathbb{E}\left[(1-q)^{\frac{N_k}{2}}\right].
$$

Hence, the preceding relation when used in relation (45) implies

$$
\mathbb{E}[f(\bar{x}_\tau) - f(x^*)] \geq -DM_f \max_{1 \leq k \leq T} \mathbb{E}\left[(1-q)^{\frac{N_k}{2}}\right].
\tag{47}
$$

*Case 2:* Another way to upper estimate the right hand side of relation (46) is as follows:

$$
\mathbb{E}[\mathrm{dist}(\bar{x}_\tau, X \cap Y)] \leq D\left(\frac{\max_{1 \leq k \leq \tau} \bar{r}_k^2}{\sum_{k=1}^{\tau} \bar{r}_k^2}\right) \sum_{k=1}^{\tau} \mathbb{E}\left[(1-q)^{\frac{N_k}{2}}\right].
$$

Now since $\bar{r}_{k-1} \leq \bar{r}_k$ for all $k \geq 1$, we obtain $\max_{1 \leq k \leq \tau} \bar{r}_k^2 = \bar{r}_\tau^2$. Hence, the preceding relation becomes

$$
\mathbb{E}[\mathrm{dist}(\bar{x}_\tau, X \cap Y)] \leq D\mathbb{E}\left[\left(\frac{\bar{r}_\tau^2}{\sum_{k=1}^{\tau} \bar{r}_k^2}\right) \sum_{k=1}^{\tau} (1-q)^{\frac{N_k}{2}}\right].
\tag{48}
$$

Note that from relation (43), we obtain

$$
\min_{k \in \{1,\ldots,T\}} \frac{\bar{r}_{k+1}^2}{\sum_{i=1}^{k} \bar{r}_i^2} = \frac{\bar{r}_{\tau+1}^2}{\sum_{i=1}^{\tau} \bar{r}_i^2}.
$$

Upper estimating relation (48) with $\bar{r}_\tau^2 \leq \bar{r}_{\tau+1}^2$ and then using the preceding relation to it, we obtain

$$
\mathbb{E}[\mathrm{dist}(\bar{x}_\tau, X \cap Y)] \leq D\left(\min_{1 \leq k \leq T} \frac{\bar{r}_{k+1}^2}{\sum_{i=1}^{k} \bar{r}_i^2}\right) \mathbb{E}\left[\sum_{k=1}^{\tau} (1-q)^{\frac{N_k}{2}}\right] \leq D\left(\min_{1 \leq k \leq T} \frac{\bar{r}_{k+1}^2}{\sum_{i=1}^{k} \bar{r}_i^2}\right) \sum_{k=1}^{T} \mathbb{E}\left[(1-q)^{\frac{N_k}{2}}\right].
\tag{49}
$$

Using Lemma B.3 with $s_k = \bar{r}_k^2$ for all $k \geq 1$ and the initial point $s_0 = r^2$ and then applying Assumption 5.2, we obtain the upper estimate of the preceding relation as

$$
\mathbb{E}[\mathrm{dist}(\bar{x}_\tau, X \cap Y)] \leq \frac{D}{T}\left(\frac{\bar{r}_{T+1}^2}{r^2}\right)^{\frac{1}{T}} \ln\left(\frac{e\bar{r}_{T+1}^2}{r^2}\right) \sum_{k=1}^{T} \mathbb{E}\left[(1-q)^{\frac{N_k}{2}}\right] \leq \frac{D}{T}\left(\frac{D}{r}\right)^{\frac{2}{T}} \ln\left(\frac{eD^2}{r^2}\right) \sum_{k=1}^{T} \mathbb{E}\left[(1-q)^{\frac{N_k}{2}}\right].
$$

Hence, using the preceding relation back to equation (45), we obtain

$$
\mathbb{E}[f(\bar{x}_\tau) - f(x^*)] \geq -M_f \frac{D}{T}\left(\frac{D}{r}\right)^{\frac{2}{T}} \ln\left(\frac{eD^2}{r^2}\right) \sum_{k=1}^{T} \mathbb{E}\left[(1-q)^{\frac{N_k}{2}}\right].
$$

Combining the preceding relation with equation (47), we obtain the lower bound

$$\mathbb{E}[f(\bar{x}_\tau) - f(x^*)] \geq -DM_f \min \left\{ \max_{1 \leq k \leq T} \mathbb{E}\left[(1-q)^{\frac{N_k}{2}}\right], \frac{\left(\frac{D}{r}\right)^{\frac{2}{T}} \ln\left(\frac{eD^2}{r^2}\right) \sum_{k=1}^T \mathbb{E}\left[(1-q)^{\frac{N_k}{2}}\right]}{T} \right\}.$$

Combining the preceding lower bound with the upper bound in relation (44), we can obtain

$$\mathbb{E}[|f(\bar{x}_\tau) - f(x^*)|] \leq \max\{A_1(T), \min\{A_2(T), A_3(T)\}\},$$

where

$$A_1(T) = \frac{3DM_f}{\sqrt{T}} \left(\frac{D}{r}\right)^{\frac{2}{T}} \ln\left(\frac{eD^2}{r^2}\right),$$

$$A_2(T) = DM_f \max_{1 \leq k \leq T} \mathbb{E}\left[(1-q)^{\frac{N_k}{2}}\right],$$

$$A_3(T) = \frac{DM_f}{T} \left(\frac{D}{r}\right)^{\frac{2}{T}} \ln\left(\frac{eD^2}{r^2}\right) \sum_{k=1}^T \mathbb{E}\left[(1-q)^{\frac{N_k}{2}}\right].$$

This concludes the proof of the theorem. **Q.E.D.**

**Discussion on some of the bounds in Theorem 5.3:** The bounds $A_1(T)$ and $A_3(T)$ involve the quantity $\left(\frac{D}{r}\right)^{\frac{2}{T}}$, which tends to 1 as $T$ increases. Moreover, for all $T \geq 1$, the quantity $\left(\frac{D}{r}\right)^{\frac{2}{T}}$ in $A_1(T)$ and $A_3(T)$ can be replaced by a modest constant. To see this, we follow similar proof lines as in (Moshtaghifar et al., 2025) [Theorem 2.1]. If $T \geq \ln\left(\frac{D^2}{r^2}\right)$, then the quantity

$$\left(\frac{D^2}{r^2}\right)^{\frac{1}{T}} = \exp\left(\frac{1}{T} \ln\left(\frac{D^2}{r^2}\right)\right) \leq e. \tag{50}$$

On the other hand, if $T < \ln\left(\frac{D^2}{r^2}\right)$, then

$$e \ln\left(\frac{eD^2}{r^2}\right) \geq \ln\left(\frac{eD^2}{r^2}\right) \geq \ln\left(\frac{D^2}{r^2}\right) > T. \tag{51}$$

Moreover for this case, the upper bound on $f(\bar{x}_\tau) - f^*$ can be estimated using Assumptions 5.1 and 5.2 and the preceding relation, yielding

$$f(\bar{x}_\tau) - f^* \leq M_f D \leq \frac{M_f D}{\sqrt{T}} T \leq \frac{eM_f D}{\sqrt{T}} \ln\left(\frac{eD^2}{r^2}\right).$$

Hence for all $T \geq 1$, $A_1(T)$ can be upper bounded as

$$A_1(T) \leq \frac{2eDM_f}{\sqrt{T}} \ln\left(\frac{eD^2}{r^2}\right).$$

Similarly, we can also upper estimate $A_3(T)$. When $T < \ln\left(\frac{D^2}{r^2}\right)$, then from the analysis of the infeasibility gap as done in Appendix E.1, we see

$$\mathbb{E}[f(\bar{x}_\tau) - f^*] \geq -M_f \mathbb{E}[\text{dist}(\bar{x}_\tau, X \cap Y)].$$

Applying Assumption 5.2 to the preceding relation and finally relation (51), we can obtain

$$\mathbb{E}[|f(\bar{x}_\tau) - f^*|] \leq M_f D = \frac{M_f D}{T} T \leq \frac{eM_f D}{T} \ln\left(\frac{eD^2}{r^2}\right) \leq \frac{eM_f D}{T} \ln\left(\frac{eD^2}{r^2}\right) \max\left\{1, \sum_{k=1}^T \mathbb{E}\left[(1-q)^{\frac{N_k}{2}}\right]\right\}.$$

When $T \geq \ln\left(\frac{D^2}{r^2}\right)$, relation (50) holds true. Hence, we finally obtain for all $T \geq 1$,

$$A_3(T) \leq \frac{eDM_f}{T} \ln\left(\frac{eD^2}{r^2}\right) \max\left\{1, \sum_{k=1}^{T} \mathbb{E}\left[(1-q)^{\frac{N_k}{2}}\right]\right\}.$$

The bound on $A_2(T)$ decays geometrically fast with respect to $\max_{1 \leq k \leq T} \mathbb{E}\left[(1-q)^{\frac{N_k}{2}}\right]$. If the number $N_k$ of samples is deterministic, with $N_k \geq N$ for all $k \geq 1$, and the number $T$ of iterations is chosen in advance, then the minimum number $N$ of samples required to make the bounds $A_1(T)$ and $A_2(T)$ to be equal is bounded below as follows:

$$N \geq \max\left\{1, \frac{2\ln\left(\frac{\sqrt{T}}{3\left(\frac{D}{r}\right)^{\frac{2}{T}}\ln\left(\frac{eD^2}{r^2}\right)}\right)}{\ln\left(\frac{1}{1-q}\right)}\right\} = \max\left\{1, \frac{\ln T - \frac{4}{T}\ln\left(\frac{D}{r}\right) - 2\ln\left(3\ln\frac{eD^2}{r^2}\right)}{\ln\left(\frac{1}{1-q}\right)}\right\}.$$

Under Assumption 5.2, all constants can be estimated as discussed in Appendix D.4. If the number $N_k$ of samples is stochastic, then a selection range can be established on the parameter of the underlying sampling distribution.

## E.2. Upper bound on the quantity $\sum_{k=1}^{T} \mathbb{E}\left[(1-q)^{\frac{N_k}{2}}\right]$

In this subsection, we will derive the upper bound on the quantity $\sum_{k=1}^{T} \mathbb{E}\left[(1-q)^{\frac{N_k}{2}}\right]$. We will consider the cases when the sample selection $N_k$ is deterministic and stochastic, and when a constant upper bound for $\sum_{k=1}^{T} \mathbb{E}\left[(1-q)^{\frac{N_k}{2}}\right]$ exists.

### E.2.1. DETERMINISTIC $N_k$ WITH $N_k = \lceil k^{\frac{1}{p}} \rceil$ AND $p > 0$

For this case, the analysis follows as (Chakraborty & Nedić, 2025)[Lemma 5.4]. We obtain

$$\sum_{k=1}^{T} \mathbb{E}\left[(1-q)^{\frac{N_k}{2}}\right] = \sum_{k=1}^{T} \left(\sqrt{1-q}\right)^{\lceil k^{\frac{1}{p}} \rceil} \leq \sum_{k=1}^{T} \left(\sqrt{1-q}\right)^{k^{\frac{1}{p}}} \leq \int_{k=0}^{T} \left(\sqrt{1-q}\right)^{k^{\frac{1}{p}}} dk$$

$$= \int_{k=0}^{T} \exp\left(\ln\left(\sqrt{1-q}\right)^{k^{\frac{1}{p}}}\right) dk = \int_{k=0}^{T} \exp\left(-ak^{\frac{1}{p}}\right) dk,$$

where $a = \frac{1}{2}\ln\left(\frac{1}{1-q}\right)$. With change of variables as $\widetilde{p} = ak^{\frac{1}{p}}$, we obtain

$$\sum_{k=1}^{T} \mathbb{E}\left[(1-q)^{\frac{N_k}{2}}\right] \leq \frac{p}{a^p} \int_{0}^{aT^{\frac{1}{p}}} \widetilde{p}^{p-1} \exp(-\widetilde{p})d\widetilde{p} \leq \frac{p}{a^p} \int_{0}^{\infty} \widetilde{p}^{p-1} \exp(-\widetilde{p})d\widetilde{p} = \frac{p\Gamma(p)}{a^p} = \frac{2^p\Gamma(p+1)}{\left(\ln\left(\frac{1}{1-q}\right)\right)^p},$$

where $\Gamma(p)$ is the gamma function. Therefore, the preceding constant upper bound is quite general and holds for any $p > 0$ and any $1 \leq T \leq \infty$. For specific values of $p$, a tighter upper bound can be derived. We will not discuss these special cases here, as the analysis proceeds similarly without altering the upper limit of integration in the preceding relation and doing an integration by parts.

### E.2.2. SAMPLING BASED ON POISSON DISTRIBUTION WITH PARAMETER $\lambda_k = \lceil k^{\frac{1}{p}} \rceil$ AND $p > 0$

We consider $N_k \sim \text{Poisson}(\lambda_k)$ with $\lambda_k = \lceil k^{\frac{1}{p}} \rceil$. Using the expression of $\mathbb{E}\left[(1-q)^{N_k/2}\right]$ derived in Appendix D.5.1, we see

$$\sum_{k=1}^{T} \mathbb{E}\left[(1-q)^{\frac{N_k}{2}}\right] = \sum_{k=1}^{T} \exp\left(-\lceil k^{\frac{1}{p}} \rceil \left(1 - (1-q)^{1/2}\right)\right) \leq \sum_{k=1}^{T} \exp\left(-k^{\frac{1}{p}}\left(1 - \sqrt{1-q}\right)\right)$$

Let us consider $a = 1 - \sqrt{1-q}$. Then the rest of the analysis follows similar steps as Appendix E.2.1 and we can obtain the following constant upper bound for any $1 \leq T \leq \infty$,

$$\sum_{k=1}^{T} \mathbb{E}\left[(1-q)^{\frac{N_k}{2}}\right] \leq \frac{\Gamma(r+1)}{\left(1-\sqrt{1-q}\right)^p}.$$

### E.2.3. SAMPLING BASED ON BINOMIAL DISTRIBUTION WITH PARAMETER $n_k = \lceil k^{\frac{1}{p}} \rceil$ AND $p > 0$

We choose $N_k \sim \text{Binomial}(n_k, \bar{p})$ with the parameter $n_k = \lceil k^{\frac{1}{p}} \rceil$ and $p > 0$. Following the analysis as in Appendix D.5.3, we can obtain

$$\sum_{k=1}^{T} \mathbb{E}\left[(1-q)^{\frac{N_k}{2}}\right] = \sum_{k=1}^{T} \left(1 - \bar{p}\left(1 - \sqrt{1-q}\right)\right)^{\lceil k^{\frac{1}{p}} \rceil} \leq \sum_{k=1}^{T} \left(1 - \bar{p}\left(1 - \sqrt{1-q}\right)\right)^{k^{\frac{1}{p}}}$$

$$= \sum_{k=1}^{T} \exp\left(\ln\left(\left(1 - \bar{p}\left(1 - \sqrt{1-q}\right)\right)^{k^{\frac{1}{p}}}\right)\right) = \sum_{k=1}^{T} \exp\left(-ak^{\frac{1}{p}}\right),$$

where $a = \ln\left(\frac{1}{1-\bar{p}(1-\sqrt{1-q})}\right)$. The rest of the analysis follows the same lines as done in Appendix E.2.1 and we can finally obtain for all $1 \leq T \leq \infty$,

$$\sum_{k=1}^{T} \mathbb{E}\left[(1-q)^{\frac{N_k}{2}}\right] \leq \frac{\Gamma(r+1)}{\left(\ln\left(\frac{1}{1-\bar{p}(1-\sqrt{1-q})}\right)\right)^p}.$$

## F. Missing Details of T-DoWS with Randomized Feasibility

### F.1. Tamed Distance over Weighted Gradients (T-DoWS) with Randomized Feasibility Algorithm

We present the complete T-DoWS algorithm, combined with Randomized Feasibility (cf. Algorithm 1), in Algorithm 4 which is provided below.

---

**Algorithm 4** T-DoWS with Randomized Feasibility

---

1: **Input:** $v_0 = v_1 \in Y$, estimates $p_0 \geq 0$, $\bar{r}_0 = r > 0$
2: **Pass:** $v_1$ and $N_1 \in \mathcal{I}_1$ to Algorithm 1 to obtain $x_1 \in Y$
3: **Equate:** $x_0 = x_1 \in Y$
4: **for** $k = 1, 2, \ldots, T$ **do**
5:     $\bar{r}_k = \max\{\|x_k - x_0\|, \bar{r}_{k-1}\}$
6:     $p_k = p_{k-1} + \bar{r}_k^2 \|s_f(x_k)\|^2$
7:     $v_{k+1} = \Pi_Y[x_k - \alpha_k s_f(x_k)]$ with $\alpha_k = \begin{cases} \frac{\bar{r}_k^2}{2\sqrt{p_k}\ln\left(\frac{ep_k}{p_1}\right)} & \text{if } p_0 = 0 \\ \frac{\bar{r}_k^2}{\sqrt{2p_k}\ln\left(\frac{ep_k}{p_0}\right)} & \text{if } p_0 > 0 \end{cases}$
8:     $x_{k+1} = \text{Algorithm } 1(v_{k+1}, N_{k+1})$
9: **end for**

---

For Algorithm 4, it can be shown that, the iterates and the distance estimates are surely bounded, following arguments similar to those in (Khaled et al., 2023) [Lemma 4][2] and using Assumption 5.4. The next lemma in the next subsection formally states this fact.

---

[2]The T-DoWG version provided in (Khaled et al., 2023) [Lemma 4] is missing a scaling factor of $\frac{1}{2}$ in the step-size update on which their analysis is based – potentially a typographical oversight. In addition, their analysis uses the term $\ln\left(\frac{2p_k}{p_1}\right)$ in the step-size scaling, which would lead to a slightly different constant in the final bound compared to what is presented in (Khaled et al., 2023) [Lemma 4]. In contrast, we use the scaling factor $\ln\left(\frac{ep_k}{p_1}\right)$, which avoids such additional constants in the proof. If $p_0 = 0$ and one uses the scaling

## F.2. Lemma on boundedness of iterates, distance estimates and subgradients for Algorithm 4

**Lemma F.1.** *Let Assumptions 2.1, 5.1, and 5.4 hold. Let B be the bound defined in Assumption 5.4 and $x^*$ be a solution to problem (P) (which is assumed to exist as per Assumption 5.1).Then, if the initialization parameter $r$ satisfies $r \leq 4\|x_0 - x^*\|$, the sequences $\{x_k\}$ and $\{\bar{r}_k\}$ produced by Algorithm 4 are surely bounded for all $k \geq 1$, i.e.,*

$$\bar{r}_k \leq \widehat{B} := \max\{B + \|x_0\|, 4\|x_0 - x^*\|\}, \quad \|x_k - x^*\| \leq \widetilde{B} := \max\{B + \|x^*\|, 3\|x_0 - x^*\|\}.$$

*Moreover, the subgradients of the objective and the constraint functions are bounded, and the sequences $\{v_k\}$ and $\{z_k^i\}$ for $i = 1, \ldots, N_k$ are also bounded, i.e., there exist deterministic scalars $M_f > 0$ and $M_g > 0$ such that surely for all $i = 1, \ldots, N_k$ and $k \geq 1$, $\|s_f(x_k)\| \leq M_f$, $\|d_k^i\| \leq M_g$, and*

$$\|z_{k+1}^i - x^*\| \leq \|v_{k+1} - x^*\| \leq \bar{D}(p_0) \quad \text{where } \bar{D}(p_0) = \begin{cases} \dfrac{\widehat{B}^2 M_f}{2r\|s_f(x_0)\|} + \widetilde{B} & \text{if } p_0 = 0, \\ \dfrac{\widehat{B}^2 M_f}{\sqrt{2(p_0 + r^2\|s_f(x_0)\|^2)}} + \widetilde{B} & \text{if } p_0 > 0. \end{cases}$$

*Proof.* By using the definition of $v_{k+1}$ in Algorithm 4, the distance of the iterate $v_{k+1}$ from any arbitrary iterate $x \in Y$ can be given as

$$\|v_{k+1} - x\|^2 \leq \|x_k - x\|^2 - 2\alpha_k \langle s_f(x_k), x_k - x \rangle + \alpha_k^2 \|s_f(x_k)\|^2. \tag{52}$$

Now applying Assumption 5.1 and substituting the point $x = x^*$ to relation (52) which is a solution to problem (P), we obtain for all $k \geq 1$,

$$\|v_{k+1} - x^*\|^2 \leq \|x_k - x^*\|^2 - 2\alpha_k(f(x_k) - f(x^*)) + \alpha_k^2 \|s_f(x_k)\|^2.$$

Applying Lemma 3.1(a) to the left hand side of the preceding relation, we obtain

$$\|x_{k+1} - x^*\|^2 \leq \|x_k - x^*\|^2 - 2\alpha_k(f(x_k) - f(x^*)) + \alpha_k^2 \|s_f(x_k)\|^2. \tag{53}$$

Now, we consider two cases.

*Case (i):* If $f(x_k) < f(x^*)$ for any $k \geq 1$, then by Assumption 5.4, we already have $\|x_k\| \leq B$.

*Case (ii):* If $f(x_k) \geq f(x^*)$, then relation (53) can be upper estimated as

$$\|x_{k+1} - x^*\|^2 \leq \|x_k - x^*\|^2 + \alpha_k^2 \|s_f(x_k)\|^2.$$

For simplicity, let us denote $\hat{d}_k = \|x_k - x^*\|$. Then, we obtain

$$\hat{d}_{k+1}^2 - \hat{d}_k^2 \leq \alpha_k^2 \|s_f(x_k)\|^2 = \begin{cases} \dfrac{\bar{r}_k^4 \|s_f(x_k)\|^2}{4p_k \left(\ln\left(\frac{ep_k}{p_1}\right)\right)^2} & \text{if } p_0 = 0, \\ \dfrac{\bar{r}_k^4 \|s_f(x_k)\|^2}{2p_k \left(\ln\left(\frac{ep_k}{p_0}\right)\right)^2} & \text{if } p_0 > 0, \end{cases}$$

where in the preceding relation, we substituted the value of $\alpha_k$ used in Algorithm 4. Note that in the preceding relation, $\bar{r}_k^2 \|s_f(x_k)\|^2 = p_k - p_{k-1}$ for all $k \geq 1$. Next, we sum both sides of the preceding relation from $k = 1$ to $T$ for any $T \geq 1$, and use $\bar{r}_k^2 \leq \bar{r}_T^2$ to obtain

$$\hat{d}_{T+1}^2 - \hat{d}_1^2 \leq \begin{cases} \dfrac{\bar{r}_T^2}{4} \sum_{k=1}^T \dfrac{p_k - p_{k-1}}{p_k \left(\ln\left(\frac{ep_k}{p_1}\right)\right)^2} & \text{if } p_0 = 0, \\ \dfrac{\bar{r}_T^2}{2} \sum_{k=1}^T \dfrac{p_k - p_{k-1}}{p_k \left(\ln\left(\frac{ep_k}{p_0}\right)\right)^2} & \text{if } p_0 > 0. \end{cases}$$

---

factor $\ln(ep_k/p_0)$ in $\alpha_k$, then even though (Ivgi et al., 2023) [Lemma 6] results hold true, the step sizes $\{\alpha_k\}_{k \geq 1} = 0$, leading to no progress in the method. For this case, we used a different scaling on $\alpha_k$. On the other hand, if $p_0 > 0$, then (Ivgi et al., 2023) [Lemma 6] can be applied directly, and we can choose a different step size $\alpha_k$ with a reduced scaling factor $\frac{1}{\sqrt{2}}$ as provided in Algorithm 4, which still guarantees boundedness of the iterates.

Applying Lemma B.4 to the preceding relation, we obtain the following for any $p_0 \geq 0$,

$$\hat{d}_{T+1}^2 \leq \hat{d}_1^2 + \frac{\bar{r}_T^2}{2}. \tag{54}$$

The rest of the analysis follows along the similar lines as (Khaled et al., 2023)[Lemma 4]. In Algorithm 4, we initialize $x_1 = x_0$. Hence,

$$\hat{d}_1 = \|x_1 - x^*\| = \|x_0 - x^*\| = \hat{d}_0.$$

Using the preceding relation back to relation (54), we obtain

$$\hat{d}_{k+1}^2 \leq \hat{d}_0^2 + \frac{\bar{r}_k^2}{2} \quad \text{for all } 1 \leq k \leq T \text{ with any } T \geq 1. \tag{55}$$

In order to prove the boundedness of the iterates, we will take the help of recursion. Note that $\bar{r}_0 = \bar{r}_1 = r$ with the initialization $x_0 = x_1$ in Algorithm 4. We require

$$\bar{r}_0 = \bar{r}_1 = r \leq 4\hat{d}_0 = 4\|x_0 - x^*\|, \tag{56}$$

to hold in order to proceed further with the proof. Let us assume that we choose a small value of $r$ in the initialization such that the condition in relation (56) is satisfied. So let us consider that

$$\bar{r}_k \leq 4\hat{d}_0 \quad \text{for } k \geq 1. \tag{57}$$

Then by induction, we require to show that $\bar{r}_{k+1} \leq 4\hat{d}_0$ for $k \geq 1$. Using relation (55), we obtain

$$\hat{d}_{k+1}^2 \leq \hat{d}_0^2 + \frac{16}{2}\hat{d}_0^2 = 9\hat{d}_0^2. \tag{58}$$

Hence $\hat{d}_{k+1} \leq 3\hat{d}_0$ for any $k \geq 1$. Now,

$$\bar{r}_{k+1} = \max\{\|x_{k+1} - x_0\|, \bar{r}_k\}.$$

If $\bar{r}_k \geq \|x_{k+1} - x_0\|$, then $\bar{r}_{k+1} = \bar{r}_k \leq 4\hat{d}_0$. This ensures that $\hat{d}_{k+2} \leq 3\hat{d}_0$ following relations (55) and (58). On the other hand, if $\bar{r}_k < \|x_{k+1} - x_0\|$, then

$$\bar{r}_{k+1} = \|x_{k+1} - x_0\| \leq \|x_{k+1} - x^*\| + \|x_0 - x^*\| = \underbrace{\hat{d}_{k+1}}_{\leq 3\hat{d}_0} + \hat{d}_0 \leq 4\hat{d}_0.$$

Hence, in either case, $\bar{r}_{k+1} \leq 4\hat{d}_0$ and $\hat{d}_{k+2} = \|x_{k+2} - x^*\| \leq 3\hat{d}_0$ for any $k \geq 1$. This concludes the recursion. Hence, combining cases (i) and (ii), we conclude that if $r \leq 4\|x_0 - x^*\|$, then for all $k \geq 1$,

$$\bar{r}_k \leq \widehat{B} := \max\{B + \|x_0\|, 4\|x_0 - x^*\|\} \quad \text{and} \quad \hat{d}_k \leq \widetilde{B} := \max\{B + \|x^*\|, 3\|x_0 - x^*\|\} \quad \text{surely.} \tag{59}$$

Moreover, we also see that for all $k \geq 1$,

$$\|x_k\| \leq \max\{B, 3\|x_0 - x^*\| + \|x^*\|\}.$$

The preceding relation shows that the iterate sequences $\{x_k\}$ of Algorithm 4 are always surely bounded for all $k \geq 1$. This further implies that the subgradient of the objective function $\|s_f(x_k)\|$ and the subgradients of the constraints $\|d_k^i\|$ are surely bounded (Bertsekas et al., 2003)[Proposition 4.2.3], i.e., there exists constants $M_f > 0$ and $M_g > 0$ such that

$$\|s_f(x_k)\| \leq M_f \quad \text{and} \quad \|d_k^i\| \leq M_g \quad \text{for all } i = 1, \ldots, N_k \text{ and } k \geq 1. \tag{60}$$

Next, we show that the iterates $v_{k+1}$ are also surely bounded for all $k \geq 1$. We let $x = x_k \in Y$ in relation (52) and obtain

$$\|v_{k+1} - x_k\|^2 \leq \alpha_k^2 \|s_f(x_k)\|^2.$$

Using the subgradient boundedness (relation (60)), from the preceding relation we obtain

$$\|v_{k+1} - x_k\| \le M_f \alpha_k.$$

Now, the quantity $\|v_{k+1} - x^*\|$ can be upper estimated using triangle inequality as

$$\|v_{k+1} - x^*\| \le \|v_{k+1} - x_k\| + \|x_k - x^*\| \le M_f \alpha_k + \underbrace{\|x_k - x^*\|}_{\hat{d}_k}.$$

Applying relation (59) to the preceding relation yields

$$\|v_{k+1} - x^*\| \le M_f \alpha_k + \widetilde{B}. \tag{61}$$

Next, we establish a bound on $\alpha_k$ for all $k \ge 1$. By the definition of $\alpha_k$ used in Algorithm 4 and applying relation (59) along with the initialization $x_0 = x_1$, we see that for all $k \ge 1$,

$$\alpha_k = \begin{cases} \dfrac{\bar{r}_k^2}{2\sqrt{p_k}\ln\left(\frac{ep_k}{p_1}\right)} \le \dfrac{\widehat{B}^2}{2\sqrt{p_1}} = \dfrac{\widehat{B}^2}{2r\|s_f(x_0)\|} & \text{if } p_0 = 0, \\[4mm] \dfrac{\bar{r}_k^2}{\sqrt{2p_k}\ln\left(\frac{ep_k}{p_0}\right)} \le \dfrac{\widehat{B}^2}{\sqrt{2p_1}} = \dfrac{\widehat{B}^2}{\sqrt{2(p_0 + r^2\|s_f(x_0)\|^2)}} & \text{if } p_0 > 0. \end{cases}$$

Combining the bounds of the preceding relation with relation (61), we obtain the desired bound for $\|v_{k+1} - x^*\|$ stated in the lemma. Moreover, applying the properties of feasibility updates, we surely have for all $k \ge 1$ and all $i = 1, \ldots, N_k$,

$$\|z_{k+1}^i - x^*\| \le \|z_{k+1}^{i-1} - x^*\|.$$

Noting that $z_k^0 = v_k$, we obtain the bound $\|z_{k+1}^i - x^*\| \le \|v_{k+1} - x^*\|$. **Q.E.D.**

### F.3. Proof of Theorem 5.5

*Proof.* We follow a similar analysis to the proof of Theorem 5.3 and obtain relation (37), which is re-written below for ease of reference,

$$\sum_{k=1}^{T} \bar{r}_k^2 \left(f(x_k) - f(x^*)\right) \le \frac{1}{2} \sum_{k=1}^{T} \frac{\bar{r}_k^2}{\alpha_k}\left(\hat{d}_k^2 - \hat{d}_{k+1}^2\right) + \frac{1}{2}\sum_{k=1}^{T}\alpha_k\bar{r}_k^2\|s_f(x_k)\|^2, \tag{62}$$

where $\hat{d}_k = \|x_k - x^*\|$ for all $k \ge 1$. In order to proceed further with the analysis, we consider two cases when $p_0 = 0$ and when $p_0 > 0$.

*Case $p_0 = 0$:* Substituting the value of $\alpha_k$ from Algorithm 4 for the first term on the right hand side of relation (62), we see

$$\frac{1}{2}\sum_{k=1}^{T}\frac{\bar{r}_k^2}{\alpha_k}\left(\hat{d}_k^2 - \hat{d}_{k+1}^2\right) = \sum_{k=1}^{T}\sqrt{p_k}\ln\left(\frac{ep_k}{p_1}\right)\left(\hat{d}_k^2 - \hat{d}_{k+1}^2\right).$$

Following the same lines of analysis as done in Theorem 5.3, we can obtain

$$\frac{1}{2}\sum_{k=1}^{T}\frac{\bar{r}_k^2}{\alpha_k}\left(\hat{d}_k^2 - \hat{d}_{k+1}^2\right) \le 4\sqrt{p_T}\ln\left(\frac{ep_T}{p_1}\right)\bar{d}_{T+1}\bar{r}_{T+1}, \tag{63}$$

where $\bar{d}_{T+1} = \max_{1 \le k \le T+1}\hat{d}_k$. Now the second quantity on the right hand side of relation (62) can be simplified using the definition of $\alpha_k$ and using similar proof lines as Theorem 5.3 yielding

$$\frac{1}{2}\sum_{k=1}^{T}\alpha_k\bar{r}_k^2\|s_f(x_k)\|^2 = \frac{1}{4}\sum_{k=1}^{T}\frac{\bar{r}_k^4\|s_f(x_k)\|^2}{\sqrt{p_k}\ln\left(\frac{ep_k}{p_1}\right)} \le \frac{\bar{r}_{T+1}^2}{4\ln\left(\frac{ep_1}{p_1}\right)}\times 2(\sqrt{p_T} - \sqrt{p_0}) = \frac{\bar{r}_{T+1}^2\sqrt{p_T}}{2}. \tag{64}$$

Moreover, the quantity $p_T$ can also be upper estimated as

$$p_T \leq \bar{r}_{T+1}^2 M_f^2 T. \tag{65}$$

Hence, substituting relations (63), (64) and (65) back to relation (62), we can obtain

$$\sum_{k=1}^{T} \bar{r}_k^2 \left( f(x_k) - f(x^*) \right) \leq \bar{r}_{T+1}^2 M_f \sqrt{T} \left( 4 \ln \left( \frac{e \bar{r}_{T+1}^2 M_f^2 T}{p_1} \right) \bar{d}_{T+1} + \frac{\bar{r}_{T+1}}{2} \right).$$

Next, we apply Lemma F.1 to upper estimate $\bar{r}_{T+1}$ and $\bar{d}_{T+1}$ in the quantity $4 \ln \left( \frac{e \bar{r}_{T+1}^2 M_f^2 T}{p_1} \right) \bar{d}_{T+1} + \frac{\bar{r}_{T+1}}{2}$ to yield

$$\sum_{k=1}^{T} \bar{r}_k^2 \left( f(x_k) - f(x^*) \right) \leq \bar{r}_{T+1}^2 M_f \sqrt{T} \left( 4 \ln \left( \frac{e \widehat{B}^2 M_f^2 T}{p_1} \right) \widetilde{B} + \frac{\widehat{B}}{2} \right), \tag{66}$$

where $\widehat{B} := \max\{B + \|x_0\|, 4\|x_0 - x^*\|\}$ and $\widetilde{B} := \max\{B + \|x^*\|, 3\|x_0 - x^*\|\}$. Moreover, from Algorithm 4 $p_1 = p_0 + \bar{r}_1^2 \|s_f(x_1)\|^2$. By initialization, $p_0 = 0$ and $x_1 = x_0$. Also $\bar{r}_1 = \max\{\|x_1 - x_0\|, \bar{r}_0\} = r$. Hence, we can obtain

$$p_1 = r^2 \|s_f(x_0)\|^2.$$

Here, we assume that $\|s_f(x_0)\| \neq 0$, else we are already at a solution. So now, applying the preceding relation back to relation (66), we obtain

$$\sum_{k=1}^{T} \bar{r}_k^2 \left( f(x_k) - f(x^*) \right) \leq \bar{r}_{T+1}^2 M_f \sqrt{T} \left( 4 \ln \left( \frac{e \widehat{B}^2 M_f^2 T}{r^2 \|s_f(x_0)\|^2} \right) \widetilde{B} + \frac{\widehat{B}}{2} \right).$$

By following similar steps as in the proof of Theorem 5.3, we obtain the desired result.

*Case* $p_0 > 0$: The proof for this case follows the same lines as the case $p_0 = 0$, but it leads to different constants, as presented in the theorem. **Q.E.D.**

## G. Detailed Experimental Setup and Results for Section 6

### G.1. Quadratically Constrained Quadratic Programming (QCQP) Problem

The QCQP problem stated in Section 6 is

$$\min_{x \in [-10,10]^{10}} \langle x, Ax \rangle + \langle b, x \rangle$$
$$\text{s.t. } \langle x, C_i x \rangle + \langle u_i, x \rangle - e_i \leq 0 \quad \text{for } i = 1, \ldots, m.$$

Here, $g_i(x) = \langle x, C_i x \rangle + \langle u_i, x \rangle - e_i$. The matrices $A$ and $C_i$ for all $i = 1, \ldots, m$ are generated using the same technique as in (Chakraborty & Nedić, 2025; 2024a). First, a random matrix is generated and decomposed using QR decomposition. A diagonal matrix $\Lambda$ with the desired range of eigenvalues is then selected, and the final matrix is formed as $Q\Lambda Q^\top$. The matrices $\{C_i\}_{i=1}^m$ are constructed with eigenvalues in $[0, 2]$, while the matrix $A$ is constructed with eigenvalues in $[1, 10]$ for the strongly convex case ($\mu = 1$ and $L = 10$) and $[0, 10]$ for the convex objective function case ($L = 10$). The vectors $b$ and $\{u_i\}_{i=1}^m$ are generated by sampling from a standard normal distribution $\mathcal{N}(0, 1)$. The generation of $e_i$ will be specified later on a case-by-case basis. We simulate our Algorithms 2 (only for the strongly convex case), 3, and 4 with randomized feasibility (Algorithm 1) using a sample size schedule $N_k = \lceil k^{1/2} \rceil$, and compare them with other state-of-the-art algorithms. These include the method from (Nedić & Necoara, 2019), which corresponds to Algorithm 2 with $\alpha_k = \frac{4}{\mu(k+1)}$ and $N_k = \lceil k^{1/2} \rceil$ (only for the strongly convex case); the Arrow-Hurwicz primal-dual Lagrangian methods from (He et al., 2022) and (Zhang et al., 2022a) (referred to as Alt-GDA); and ADMM with log-barrier penalty functions (Yang et al., 2022) (referred to as ACVI). We also compare these methods with CVXPY (Diamond & Boyd, 2016; Agrawal et al., 2018) (https://www.cvxpy.org/), which uses interior-point solvers. For simplicity, we refer to the algorithm in (He et al., 2022) as Arrow-Hurwicz and the algorithm in (Zhang et al., 2022a) as Alt-GDA.

*Table 1.* Run time for a single experiment

| Method | Run time |
|---|---|
| (Nedić & Necoara, 2019) | $\sim 250$ ms |
| Arrow-Hurwicz (He et al., 2022) | $\sim 5$ s 580 ms |
| Alt-GDA (Zhang et al., 2022a) | $\sim 5$ s 580 ms |
| ACVI (Yang et al., 2022) | $\sim 3$ mins 50 s |
| CVXPY | $\sim 1$ s 500 ms |
| Algorithm 2 (Ours) | $\sim 250$ ms |
| Algorithm 3 (Ours) | $\sim 250$ ms |
| Algorithm 4 (Ours) | $\sim 250$ ms |

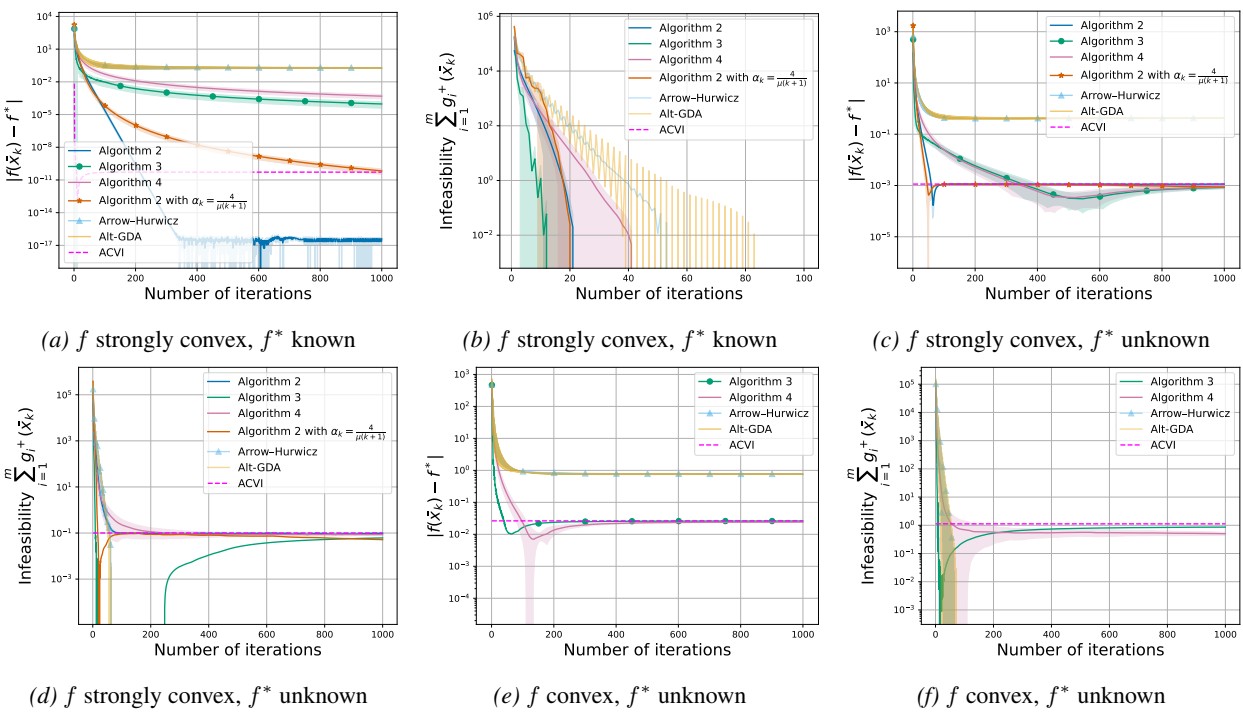

*(a) $f$ strongly convex, $f^*$ known*    *(b) $f$ strongly convex, $f^*$ known*    *(c) $f$ strongly convex, $f^*$ unknown*

*(d) $f$ strongly convex, $f^*$ unknown*    *(e) $f$ convex, $f^*$ unknown*    *(f) $f$ convex, $f^*$ unknown*

*Figure 4.* Simulation plots for the QCQP problem which is also presented in Section 6.

Since Algorithm 1 requires random selection of constraint indices, we run 5 independent experiments for Algorithms 2, 3, 4, and the method in (Nedić & Necoara, 2019) (which corresponds to Algorithm 2 with $\alpha_k = \frac{4}{\mu(k+1)}$). For our algorithms, we show the performance of the averaged iterates as per our theorems. For diminishing $\alpha_k$, we apply (Nedić & Necoara, 2019)[Theorem 3] and use the average $\bar{x}_k = \frac{\sum_{t=1}^{k}(t+1)^2 x_t}{\sum_{i=1}^{k}(i+1)^2}$. For CVXPY, Arrow-Hurwicz, Alt-GDA, and ACVI, running a single experiment is sufficient because they process all constraints in each iteration and do not involve random selection. We consider the normal averaging $\bar{x}_k = \frac{1}{T} \sum_{t=1}^{k} x_k$ for Arrow-Hurwicz and Alt-GDA. The ACVI algorithm, involves inner iteration over an index $K$ and an outer iteration over $T$, which is the total number of iterations. For ACVI, we consider averaged iterate over the inner iteration $K$ for each update in the outer iteration loop. For all the algorithms considered, we plot the values of $|f(\bar{x}_k) - f^*|$ and the infeasibility proxy $\sum_{i=1}^{m} g_i^+(\bar{x}_k)$, where $\bar{x}_k$ denotes the averaged iterate of the respective method. Error bars are shown for all methods that use Algorithm 1. All the algorithms are run for $T = 1000$ iterations and for $m = 1,000$ constraints. We provide a runtime comparison of all methods in Table 1. Note that since we use approximately the same setup with the same number of constraints for all the cases considered later, the runtime of the different methods follows a similar trend across those cases.

For Algorithm 1, we consider $\beta = 1$. For Algorithm 2, we consider $\epsilon = 10^6$ in the step size selection for all the strongly-convex cases. For Algorithms 3 and 4, we choose $p_0 = 0$ and $r = 10^{-1}$ for all the cases. For Arrow-Hurwicz method, we select the primal and dual step sizes to be $\frac{1}{\sqrt{T}}$ for all the experiments, where $T = 1000$ is the number of iterations as

per the details of (He et al., 2022). For Alt-GDA, we select both primal and dual step sizes to be $\frac{\mu}{2L^2}$ for strongly convex objective function (Zhang et al., 2022a)[Theorem 4] and $\frac{1}{2L}$ for convex objective function (Zhang et al., 2022a)[Corollary 3]. For ACVI algorithm, the parameters are very difficult to tune as a slightly wrong parameter leads to divergence of the algorithm. As per the notations of (Yang et al., 2022), we select the number of inner iterations $K = 5$ and $\delta = 0.5$. The choice of $\beta^{\text{ACVI}}$ and $\mu_{-1}^{\text{ACVI}}$ in the ACVI algorithm varies across cases and is therefore provided separately for each case below. Across all simulations, the Arrow–Hurwicz and Alt-GDA algorithms perform poorly in terms of function value convergence, although their feasibility performance remains reasonable.

**Case 1: Strongly Convex Objective Function ($f^*$ known)**   In order to understand the performance of all the methods, we want to know the optimal function value. Hence we choose $\{e_i\}_{i=1}^m$ in such a way so that the minimum of the unconstrained problem $x_{\text{opt}} = -(A + A^T)b$ is the minimum of the constrained problem, i.e., $x^* = x_{\text{opt}}$ and $f^* = f(x_{\text{opt}})$. We choose $e_i = \langle x_{\text{opt}}, C_i x_{\text{opt}} \rangle + \langle u_i, x_{\text{opt}} \rangle + l_i$, where the scalar $l_i$ is sampled from a uniform distribution in $[1, 2]$, which ensures the presence of interior points in the constrained set. With such a construction, Assumptions 2.1, 2.2, 4.1, 4.2, and 5.2 are satisfied. Note that while computing the average iterate $\bar{x}_k$ in Algorithm 2 for the strongly convex case (cf. Theorem 4.4), we require $\bar{\alpha}$, which depends on $M_f$ and estimating it in real settings can be computationally cumbersome. For the purpose of simulation, we considered a proxy estimate of $\bar{\alpha}$ as

$$\bar{\alpha}(T) = \min\left\{ \frac{1}{2(L - \mu)}, \frac{1}{L}, \frac{\epsilon}{2\max_{1 \le t \le T} \|\nabla f(x_t)\|^2} \right\}.$$

For this case, we initially considered $m = 10{,}000$ constraints for $x \in \mathbb{R}^{100}$. However, CVXPY failed to provide a solution and reported an error indicating that the matrix $A$ was not positive semidefinite, even though it was. In contrast, the algorithms that use randomized feasibility (Algorithm 1) produced solutions very quickly. The primal–dual and ADMM methods required significantly more time. Furthermore, when we increased the number of constraints to $m = 100{,}000$ for a decision of a lower dimension $x \in \mathbb{R}^{10}$, CVXPY continued to attempt to solve the problem without producing any output even after several hours. Additionally, we considered the state-of-the-art solver IPOPT (Wächter & Biegler, 2006). However, it also fails to produce results for $m = 10{,}000$ constraints with the decision vector $x \in \mathbb{R}^{100}$. Therefore, to ensure that all algorithms could be executed within a reasonable time, we considered a computationally inexpensive setting with $m = 1{,}000$ constraints for $x \in \mathbb{R}^{10}$.

We choose $\beta^{\text{ACVI}} = 0.08$ and $\mu_{-1}^{\text{ACVI}} = 10^{-5}$ for the ACVI algorithm. CVXPY solver provides the exact optimal solution $f^*$ that is known to us. The convergence of all the algorithms is shown in Figures 4a and 4b. The ACVI algorithm, which is based on an interior-point method, converges the fastest and reaches 0 infeasibility gap. Furthermore, we observe that Algorithm 2 converges linearly in function value and saturates at the lowest achievable value. Since, in this case, all algorithms converge quickly in terms of the infeasibility gap, Figure 4b is plotted for only 100 iterations to highlight the decay behavior of the different methods. We observe that the Alt-GDA algorithm exhibits significant oscillations.

**Case 2: Strongly Convex Objective Function ($f^*$ unknown)**   We use the preceding setup with the constants $\{e_i\}_{i=1}^m$ sampled from a uniform distribution on $[1, 2]$. In this case, the optimal solution is not known a priori, although the construction of the constraint set ensures that the solution lies on the boundary of the feasible region. We use CVXPY to compute the exact optimal function value $f^*$ and evaluate $|f(\bar{x}_k) - f^*|$ for all the methods. Since the solution lies on the boundary, the ACVI algorithm tends to diverge due to its use of log-barrier penalty functions. After several trials, we set $\beta^{\text{ACVI}} = 10^{-15}$ and $\mu_{-1}^{\text{ACVI}} = 10$. The results are provided in Figures 4c and 4d. Algorithms 2 (with both adaptive and diminishing step sizes), 3, 4, and ACVI converge to nearly the same objective function value, with Algorithms 3 and 4 performing slightly better in this scenario.

**Case 3: Convex Objective Function ($f^*$ unknown)**   The setup is the same as in the preceding case, except that the matrix $A$ has eigenvalues in $[0, 10]$. We compute $f^*$ using CVXPY and use it to evaluate $|f(\bar{x}_k) - f^*|$ for the other algorithms. We choose $\beta^{\text{ACVI}} = 10^{-15}$ and $\mu_{-1}^{\text{ACVI}} = 10$ after trial and error. The simulation plots are presented in Figures 4e and 4f. Since the objective function is only convex, we simulate only our Algorithms 3 and 4, both of which converge to the same threshold as ACVI. Algorithms 3 and 4 are easy to implement since no prior knowledge on the problem parameters are essential, whereas tuning the hyper-parameters in ACVI is sometimes extremely difficult.

We also illustrate through simulations that increasing the number of samples $N_k$ decreases the infeasibility gap for Case 3 of Algorithm 3; the results are shown in Figure 5. Similar behavior is observed for the other algorithms and cases as

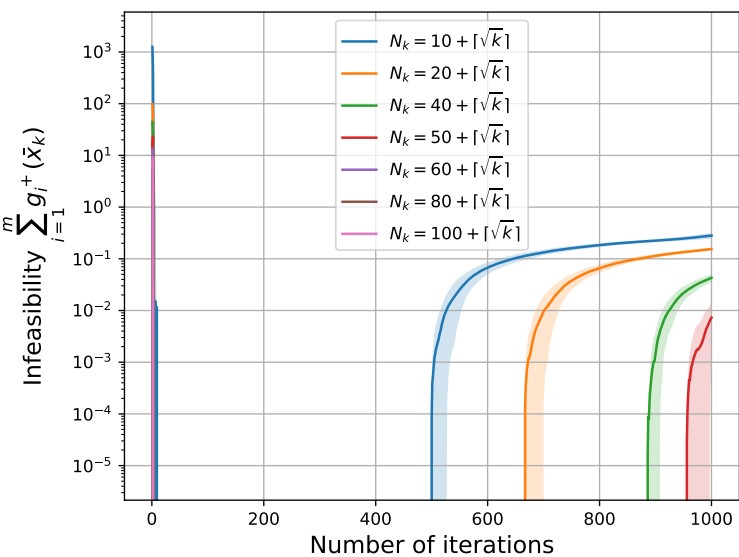

*Figure 5.* Infeasibility plot for Algorithm 3 (Case 3) with increasing sample size $N_k$.

well. Specifically, we choose $N_k = N + \lceil\sqrt{k}\rceil$ and perform multiple sweeps over $N \in \{10, 20, 40, 50, 60, 80, 100\}$. As $N$ increases, the saturation level decreases, consistent with our theoretical results for the randomized feasibility method. We observe that the infeasibility gap initially decreases to $0$ but later increases again for smaller values of $N$. We believe this occurs because the solution may lie close to the boundary, and sampling too few constraints can lead to violations. In contrast, for larger values of $N$, this behavior is not observed within $1000$ iterations.

### G.2. Support Vector Machine (SVM) Classification Problem

The problem considered here is the soft-margin SVM problem given by

$$\min_{w,b,\xi} \quad \frac{1}{2}\|w\|^2 + C\sum_{i=1}^{m}\xi_i$$

$$\text{subject to} \quad g_i(w, b, \xi_i) \le 0, \quad \xi_i \ge 0, \quad i = 1, \dots, m,$$

where $g_i(w, b, \xi_i) = 1 - \xi_i - y_i(w^\top z_i + b)$. The feature vectors are denoted as $z_i$ with $y_i \in \{-1, +1\}$ are its corresponding class labels. Hence, it is a binary classification problem. Let, $\xi = [\xi_1, \dots, \xi_m]^\top$ are the slack variables that measure margin violations, and $C > 0$ is a hyperparameter controlling the tradeoff between margin size and misclassification penalties. When all slack variables satisfy $\xi_i = 0$, the formulation reduces to the classic hard-margin SVM problem. Since the optimization involves the variables $w$, $b$, and $\xi$, we collect them into a single decision vector $x = [w, b, \xi]^\top$.

We use the regularizer constant $C = 10^{-6}$. Since the objective function is convex with respect to the variable $x$, we implement only Algorithms 3 and 4, using the initial diameter estimate $\bar{r}_0 = r = 10^{-2}$, and choose $\beta = 1$ for Algorithm 1. We compare our methods with primal–dual Lagrangian-based approaches. The Arrow-Hurwicz method (He et al., 2022) and the Alt-GDA method (Zhang et al., 2022a) do not converge with the theoretical step sizes proposed in those works, whereas our methods employ adaptive step sizes and converge without requiring knowledge of problem parameters. Since the Arrow–Hurwicz and Alt-GDA methods fail to converge with their theoretical step sizes, we performed 3-fold cross-validation on the training dataset and conducted a grid search over three candidate primal and dual step sizes for 200 iterations to identify step sizes that yield low misclassification error. This grid-search procedure introduces additional computational overhead for the primal-dual methods.

We consider the following datasets: (i) Banknote Authentication (4-dimensional features with 1097 training and 275 test data points) (Lohweg, 2012), used to classify genuine and forged banknotes; (ii) Breast Cancer Wisconsin (30-dimensional features with 455 training and 114 test data points) (Wolberg et al., 1993), used to classify benign and malignant tumors; and (iii) MNIST digits 3 and 5 classification (784-dimensional features with 11,552 training and 1,902 test data points) (LeCun

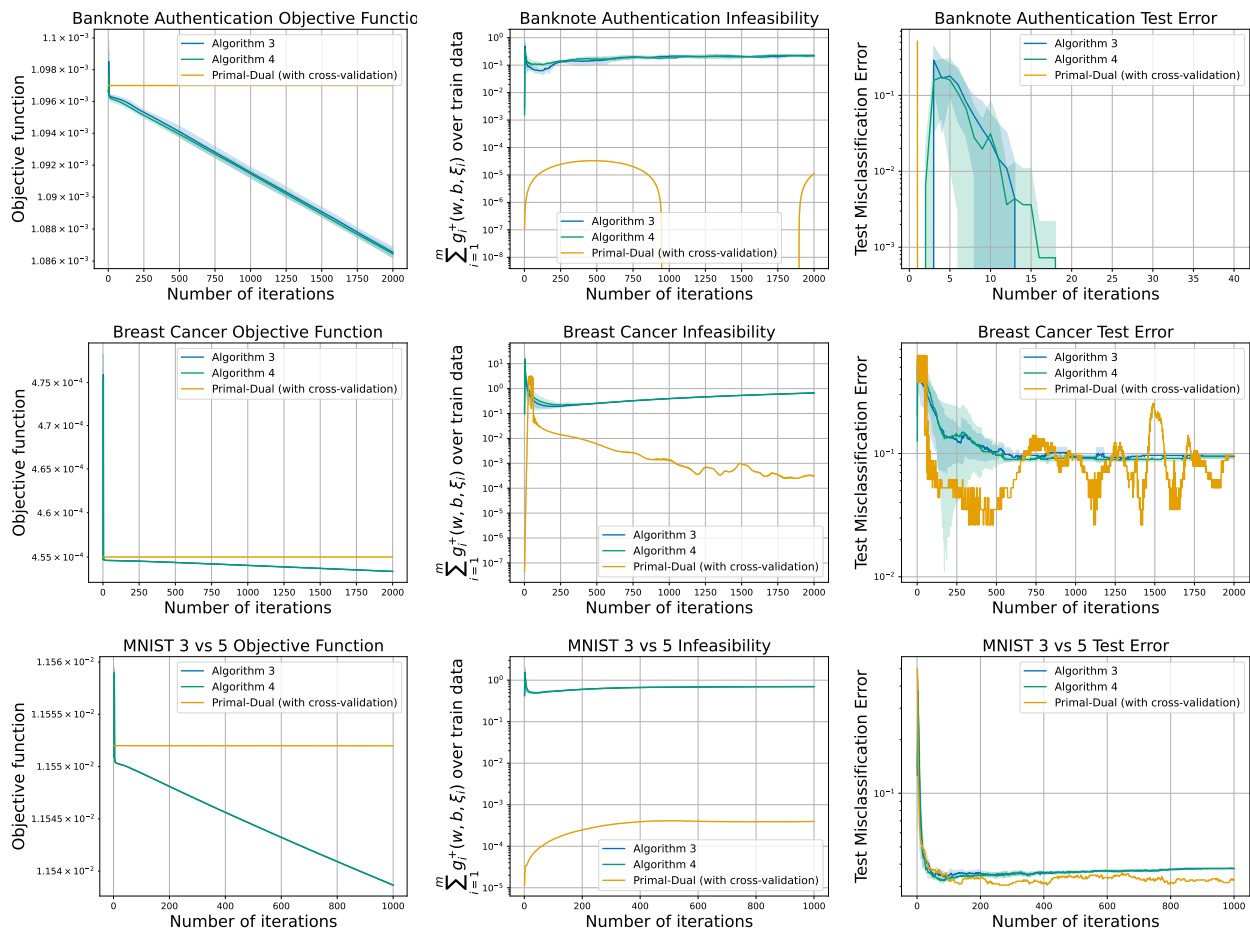

*Figure 6.* Objective function value, training feasibility violation, and test misclassification error for Algorithms 3 and 4, and the primal–dual method with step sizes chosen via 3-fold cross-validation.

et al., 2010). The Banknote Authentication dataset is linearly separable; therefore, a small sample size of $N_k = 50$ works well. On the other hand, to achieve good performance on the Breast Cancer Wisconsin and MNIST 3 vs. 5 datasets, we choose $N_k = 2000$ and $N_k = 10000$, respectively. In general, a larger sample size leads to lower test misclassification error.

Since primal-dual methods process all data points in a single update step, whereas Algorithm 1 processes randomly sampled data points sequentially, the computational time per iteration of the primal-dual methods is lower than that of Algorithms 3 and 4. However, primal-dual methods require cross-validation to select suitable step sizes, and performing multiple such comparisons incurs significant additional computational cost. Therefore, in the SVM experiments, we do not compare the computational time of all these methods, as such a comparison would not be fair.

Figure 6 indicates that, across all datasets, the objective function decreases most rapidly for Algorithms 3 and 4 compared to the primal–dual algorithm. However, in terms of the infeasibility gap on the training dataset and the test misclassification error, the primal–dual algorithm performs better, as its step sizes are selected via cross-validation. Algorithms 3 and 4 nevertheless achieve competitive performance in terms of test misclassification error on all three datasets.

### G.3. Constrained Logistic Regression with Group Fairness Constraints

For data generation, we considered $\widehat{N} = 10,000$ samples with feature dimension $n = 500$, and partitioned the data into $G = 10$ groups. We initially generated $m = 3000$ fairness constraints. Since each absolute-value fairness constraint is represented by two linear inequalities, the resulting feasibility problem contains $2m$ convex inequality constraints.

However, IPOPT failed to converge within a reasonable time and frequently crashed for this larger problem size. Since

the IPOPT solution is used to estimate the baseline value $f^*$, we reduced the number of fairness constraints to $m = 1000$, yielding a total of $2m = 2000$ convex inequality constraints, for which IPOPT could successfully return a solution.

For Algorithms 3 and 4, we chose

$$N_k = 100 + \lceil k^{1/1.1} \rceil.$$

The CPU runtime of all the methods are reported in Table 2.

*Table 2.* Run time for a single experiment

| Method | Run time |
|---|---|
| Arrow-Hurwicz | $\sim 8$ secs |
| IPOPT | $\sim 77$ secs |
| Algorithm 3 (Ours) | $\sim 17$ secs |
| Algorithm 4 (Ours) | $\sim 17$ secs |

