# OpenReview forum: "Randomized Feasibility Methods for Constrained Optimization with Adaptive Step Sizes"
_ICML.cc/2026/Conference — ICML 2026 regular_

### Official Review · Reviewer_qiom · 2026-02-18

**Soundness:** 3
**Presentation:** 3
**Significance:** 2
**Originality:** 3
**Overall Recommendation:** 4
**Confidence:** 3

**Summary:**

This paper focuses on the minimization of convex objective functions under the intersection of set constraints, and a "randomized feasibility update" is introduced to avoid projecting onto complicated sets. In the case of convex yet nonsmooth objective functions, the paper introduces a parameter-free adaptive step size scheme. Theoretic properties of the proposal (including rates of decrease of infeasibility violations as updates are performed) are provided, and an empirical study with two problem settings details the proposals competitive performance.

**Compliance With Llm Reviewing Policy:**

Affirmed.

**Final Justification:**

Limited significance but well-structured. I maintain my score

**Key Questions For Authors:**

See weaknesses section:

1. What does "easy to project on" mean?
2. Can public code be provided for reproducibility and transparency?
3. What are the runtimes for the grid search for the primal-dual method used in SVM experiments? Is it truly prohibitive in some settings?

**Limitations:**

Yes

**Strengths And Weaknesses:**

Strengths:

1. The presentation of the methodology section is quite clear: the assumptions are clearly listed for each case and algorithms are well-explained.
2. The methods in the paper address multiple cases, including convex/smooth objectives versus convex/potentially nonsmooth objective functions.
3. The analysis for the convergence rates of the proposed algorithms appears sound

Weaknesses:

1. The description of the set X in problem instance (P) is unclear: what does "easy to project on" mean? Providing an exact description would make it more clear as to when these methods are useful.

2. The motivation section omits details that would make the study of the problem instance more compelling. While constrained problems do arise in the applications listed, it is unclear whether the exact form enumerated in formulation (P) appears in these settings. For instance, do the state-action constraints in constrained reinforcement learning indeed often contain convex objective function with convex constraints? Listing applications where the form (P) emerges with a large number of constraints and a complicated set X would provide far more insight as to the importance of the specific study, and what applications cannot be addressed with naive or less complicated methods.

3. No code for reproducibility provided

4. Reporting the computational runtime of the grid search for the primal-dual method in SVM experiments would provide a clearer picture as to whether these methods are computationally prohibitive.

Minor Notes:

1. Line 414 references Figure 3, while it should be Figure 1

---

> ### Author Rebuttal · Authors · 2026-03-31
>
> We thank the reviewer for taking the time to read the manuscript and provide valuable feedback.
>
> >Q) What does ``easy to project on'' mean?
>
> "Easy to project on" means there exists a closed form expression for the projection operator. As for example:
>
> (1) Box constraints: If $X = \{x \in \mathbb{R}^n : \ell \le x \le u\}$, then the projection operator $\Pi_X[z]_i = \min[\max[z_i, \ell_i], u_i]$ for all $1 \leq i \leq n$ and for all $z\in \mathbb{R}^n$.
>
> (2) Euclidean norm ball: If $X = \{x \in \mathbb{R}^n: ||x||_2 \le r\}$, then
> for any $z\in \mathbb{R}^n$, the projection of $z$ on $X$ is given by $\Pi_X[z] = z$ if $||z||_2 \leq r$, else $\Pi_X[z] = r \dfrac{z}{||z||_2}$.
>
> (3) Hyperplane: If $X = \{x \in \mathbb{R}^n : a^\top x = b\}$ with $a\ne0$, then the projection of a point $z$ on $X$ is $\Pi_X[z] = z - \frac{a^\top z - b}{||a||_2^2} a$.
>
> >Q) "The motivation section omits details that would make the study of the problem instance more compelling. While constrained problems do arise in the applications listed, it is unclear whether the exact form enumerated in formulation (P) appears ...... $X$ would provide far more insight as to the importance of the specific study, and what applications cannot be addressed with naive or less complicated methods."
>
> We thank the reviewer for this insightful comment and agree that the motivation can be strengthened by more explicitly connecting formulation (P) to practical applications. Problem (P) represents a standard convex optimization framework with a potentially large number of constraints. Such formulations arise naturally in several application domains. For instance, in constrained reinforcement learning, many formulations involve convex policy optimization problems (e.g., policy evaluation or policy improvement in linear or tabular settings, or convex relaxations of policy optimization) with convex constraints such as safety, budget, or resource constraints. Similarly, large-scale constrained learning problems (e.g., empirical risk minimization with fairness, robustness, or resource constraints) can often be cast in the form of (P), where the number of constraints is large and potentially data-dependent. Moreover, problems with a large number of constraints also arise in areas such as distributed optimization, scenario-based optimization, and chance-constrained programming, where each sampled scenario induces a constraint. In such settings, handling all constraints simultaneously can be computationally prohibitive, which motivates randomized or incremental feasibility approaches such as the one studied in this work. Importantly, while general-purpose solvers or naive projection methods may be applicable in small- to medium-scale settings, they often become impractical when the number of constraints is very large or when projection onto the full feasible set is computationally expensive. Our framework is specifically designed for such regimes. We will refine the citations to relevant literature where we provide applications as motivation that align with our current assumptions.
>
> >Q) No code for reproducibility provided.
>
> Link to our codes: https://anonymous.4open.science/r/anony666-adaptive. Note that we have also added a new experiment on logistic regression under fairness constraints.
>
> >Q) Reporting the computational runtime of the grid search for the primal-dual method in SVM experiments would provide a clearer picture as to whether these methods are computationally prohibitive.
>
> We thank the reviewer for bringing it up. We select the SVM experiment on MNIST dataset classification and did 3-fold cross validation with 200 iterations for the primal-dual method and choose 3 sets of step sizes for the primal and also for the dual updates to perform the grid search. The cross validation step took 138.9797 seconds. Currently, since we have some idea about good range of step sizes, we could choose a few. But in real world settings, if one has no side information (i.e., good estimates of range) for the parameters of the problem, then one has to select many grid points that can account for large compute time for cross validation to find the best step size. Apart from the cross validation part, primal dual method is working faster than our feasibility method while running the experiment since we choose a large feasibility batch size $N_k$ to reach similar performance as fine tuned primal-dual. Currently, one single run of Algorithm 3 or 4 with 1000 iterations took around 1 min 29 secs, whereas for primal-dual method, it took around 37 secs without the cross validation part. There is a tradeoff as to whether one wants to spend time in grid search or one wants to just run a complete parameter-free method (Algorithm 3 and 4) to reach similar performance.
>
> >Q) Line 414 references Figure 3,...
>
> Thanks to the reviewer for pointing out the typo. We will fix it in the camera ready version.
>
> The answers to the Key Questions For Authors has also been provided above.

---

> > ### Author Rebuttal · Reviewer_qiom · 2026-03-31
> >
> > The authors have addressed questions/concerns regarding reproducibility, refinement of listed applications, and runtime reporting.
> >
> > (1) When refining the literature review of applicable settings, please consider the difference between easy-to-project convex settings versus the exact form enumerated in (P). For example, it is my understanding that the fairness optimization problem enumerated in Zafar et al. 2017 can already be solved quickly due to the low number of constraints (which are also linear). Therefore, despite being listed in the manuscript, I do not believe it is of the form (P).
> >
> > (2) Thank you for providing the link to the code repository. Please consider making the code repository prominent/public in standard GitHub location after anonymous review period, and list the link or repository name in the manuscript.
> >
> > No further questions remain for the authors.

---

> > > ### Author Response · Authors · 2026-04-04
> > >
> > > We are glad the concerns have been addressed, and we thank you again for the thoughtful feedback. We would appreciate it if this is reflected in your final evaluation. In any case, we sincerely appreciate your time and feedback.

---

### Official Review · Reviewer_yBfV · 2026-03-02

**Soundness:** 4
**Presentation:** 4
**Significance:** 3
**Originality:** 2
**Overall Recommendation:** 4
**Confidence:** 4

**Summary:**

The authors propose a randomized algorithm for solving convex constrained optimization problems. Specifically, the algorithm minimizes a convex objective function subject to the intersection of the feasibility set of convex inequality constraints and a convex set onto which projections are tractable. A main component of the algorithm is a randomized feasibility-seeking stage, where from a given point a subset of the constraints are selected randomly and a direction is chosen to move toward satisfaction of these randomly chosen constraints. The number of such steps that may be performed may be random. The overall algorithm mixes this stage with a descent step in the objective function. Theoretical guarantees are provided for cases when the step sizes are adaptive. The results of some numerical experiments are provided as well.

**Compliance With Llm Reviewing Policy:**

Affirmed.

**Final Justification:**

The paper is nice, albeit not very original. It would be reasonable to publish it.

**Key Questions For Authors:**

The literature review related to barrier methods is strange to me. It seems to suggest that the only place for an interior-point methodology to solve problem (P) is in the context of ADMM, which is very far from the truth. Why are interior-point methods discussed in this way, and not in terms of the state-of-the-art solvers like Ipopt and Knitro for solving problem (P)?

**Limitations:**

The main limitation of the proposed ideas, in my view, is that they only apply for convex problems, whereas many problems of interest these days are nonconvex.

**Strengths And Weaknesses:**

Strengths:

(1) The main ideas of the paper and the theoretical analyses appear to be sound.
(2) The theoretical guarantees are challenging to obtain, yet the presentation of the theory and its analyses are very good.
(3) The proposed algorithm is applicable for challenging and important problems, such as quadratically constrained quadratic optimization problems.

Weaknesses:

(1) The overall algorithm is not very original.  There is prior work, cited here, showing that the same algorithmic framework with diminishing step sizes has convergence guarantees.  Thus, the main contribution of this paper is convergence under adaptive step sizes.

---

> ### Author Rebuttal · Authors · 2026-03-31
>
> Thanks to the reviewer for taking the time to read our manuscript and for providing valuable feedback.
>
> >Q) The overall algorithm is not very original. There is prior work, cited here, showing that the same algorithmic framework with diminishing step sizes has convergence guarantees. Thus, the main contribution of this paper is convergence under adaptive step sizes.
>
> We thank the reviewer for bringing it up. We want to politely state that even though there is a prior work with the same algorithmic framework, we have improved the convergence rate of the method by choosing a different step size and a different averaging scheme of the iterates. Moreover, for the first time, we also have a complete parameter-free, line-search free, and projection free distance adaptive method in Algorithms 3 and 4 to solve constrained optimization problems, using the same algorithmic framework of descent method. Hence, a clever choice of step size is what matters here.
>
> >Q) The literature review related to barrier methods is strange to me. It seems to suggest that the only place for an interior-point methodology to solve problem (P) is in the context of ADMM, which is very far from the truth. Why are interior-point methods discussed in this way, and not in terms of the state-of-the-art solvers like Ipopt and Knitro for solving problem (P)?
>
> We thank the reviewer for this insightful comment. We agree that the current presentation of barrier and interior-point methods may give the unintended impression that such methods are primarily relevant in the context of ADMM, which was not our intention. Our goal was to highlight approaches most closely related to our proposed framework, particularly those that combine first-order methods with constraint handling. However, since the reviewer mentioned about the state-of-the-art solvers like Ipopt and Knitro, we looked at them. We found that Knitro needs a paid subscription, whereas Ipopt is free. Hence, for our QCQP problem, we simulated Ipopt along with CVXPY. We considered the cases where CVXPY failed to give outputs as described in the lines between 1994 and 2001 in Appendix G.1. We saw that Ipopt fails to provide output for 10,000 constraints and for the decision vector $x \in \mathbb{R}^{100}$. The system crashed in both local MacBook Pro 2021 (Apple M1 pro chip, 16GB RAM) and free version of Google colab. Whereas, our methods work in that case in local machine since we sample a batch of constraints to perform feasibility steps. Hence, depending on the batch size, the compute time in each iteration can be controlled. A new experiment on logistic regression under fairness constraints as also been done and code for Ipopt is also added in the current code repository for QCQP and this new experiment. Link: https://anonymous.4open.science/r/anony666-adaptive.
>
> >Q) The main limitation of the proposed ideas, in my view, is that they only apply for convex problems, whereas many problems of interest these days are nonconvex.
>
> We thank the reviewer for the comment. In our current scope of the paper, we focus only on convex functions. It is important to note that there exists no exponential convergence rate for strongly convex functions upto an error tolerance for this problem. Moreover, there exists no line search-free, parameter-free and projection free distance adaptive methods for solving constrained optimization problems with convex objective. The current work bridges that gap. An extension to non-convex objectives can be a future direction, as we cannot address it in the current paper.

---

> > ### Author Rebuttal · Reviewer_yBfV · 2026-04-02
> >
> > My concerns have been addressed.

---

> > > ### Author Response · Authors · 2026-04-03
> > >
> > > Thank you for your thoughtful review and for engaging with our rebuttal. We are glad that our responses addressed your concerns.
> > >
> > > We noticed that the acknowledgement indicates that all concerns have been fully resolved. Given this, we were wondering whether you might consider updating your overall score to better reflect your current assessment.
> > >
> > > In any case, we sincerely appreciate your time and feedback.

---

### Official Review · Reviewer_v8N2 · 2026-03-11

**Soundness:** 2
**Presentation:** 3
**Significance:** 2
**Originality:** 3
**Overall Recommendation:** 4
**Confidence:** 4

**Summary:**

**Summary**

This paper proposes adaptive stepsize strategies for the randomized feasibility method. The authors consider smooth, strongly convex, and nonsmooth convex settings and establish corresponding complexity guarantees. Some experiments validate the theoretical claims.

**Compliance With Llm Reviewing Policy:**

Affirmed.

**Final Justification:**

In the rebuttal, the authors addressed the theoretical gaps presented in the initial submission. I think this is a technically solid paper on random feasibility methods. Overall, I raise my score to 4.

**Key Questions For Authors:**

**Minor issues**

1. A possibly related literature

   Renegar, J., & Zhou, S. (2021). A different perspective on the stochastic convex feasibility problem. *arXiv preprint arXiv:2108.12029*.

2. Line 121

   The projection notation has a typo. $\mathbb{E}[\cdot]$ is stated but $\mathsf{E}[\cdot]$ is used.

3. Line 260

   It's better to mention the problem setting at the beginning of the section.

**Questions**

1. In **Algorithm 1**, could you do a line-search to choose $\alpha$?
2. When would you choose a random $N_k$?

**Limitations:**

Yes.

**Strengths And Weaknesses:**

**Strengths**

The paper is well-written and easy to follow. The motivation and presentations are clear.

**Weaknesses**

I have some concerns regarding some of the theoretical claims and the experiments.

1. Theoretical results

   I find some of the theoretical claims are not sufficiently justified. For example,

   - In **Theorem 4.4**, there is only a guarantee w.r.t. the optimality gap, but no guarantee is stated w.r.t. the infeasibility. I believe such a guarantee can be obtained using the convexity of the distance function and the same argument as equation (33) in the appendix, but it's currently missing.
   - On Line 239, it is claimed that $\\bar{\\alpha}$ can be computed based on the iteration trajectory. I find this claim unsupported. In particular, although the assumption $\\|\\nabla f(x_k)\\| \\leq M_g$ is w.r.t. the trajectory of the algrithm, the proof on Line 1107 uses the gradient of a point after projection $\\Pi_{X\\cap Y}[x_k]$. I don't think this quantity can be controlled by $\\max_k \\|\\nabla f(x_k)\\| $, and you might need to define it as $M_g = \\sup_{\\|x - x^\\star\\| \\leq B_0 + \\|x^\\star\\|} \\|\\nabla f(x)\\|$ instead. In this case, estimating $M_g$ may not be as trivial.
   - The statement of **Theorem 5.3** is unclear. The LHS takes total expectation, while $\\tau$ on the RHS is still a random variable (since it depends on $\\bar{r}_k$, which further inherits randomness from $x_k$). This point should be clarified.

   - The proof of **Theorem 5.3** is not fully justified. On line 1382, it is claimed that  $\sum_{k=1}^T \sqrt{p_k} (\hat d_k^2 - \hat d_{k+1}^2) \le \sqrt{p_K} \sum_{k=1}^T (\hat d_k^2 - \hat d_{k+1}^2)$. But this relation only holds if one assumes that $\hat{d}_k$ is monotonically decreasing. I do not see why **Algorithm 3** guarantees this deterministically.

2. Experiments

   The QCQP examples are too small, and I recommend increasing the problem size or removing them. Besides, the figures in the experiments seem to suggest the proposed algorithms only achieve $10^{-2}$ infeasibility tolerance. This performance looks unsatisfying.

Overall, I think the paper's contribution can be solid given that the above theoretical issues are clarified/addressed. I currently recommend weak rejection, and will raise my score if my concerns are properly resolved.

---

> ### Author Rebuttal · Authors · 2026-03-31
>
> We thank the reviewer for their time to review the draft and providing us with constructive feedback.
>
> Here, due to 5000 character limit, we replied to all the comments very briefly, but we have a detailed point by point reply in this pdf: https://anonymous.4open.science/r/rebuttal-ICML2026/ICML2026_Rebuttal.pdf . We encourage and request the reviewer to check it.
>
> >Q) In \textbf{Theorem 4.4}, there is only a guarantee w.r.t. the optimality gap, ..., but it is currently missing.
>
> We thank the reviewer for the comment. Definitely, the comment is valid. However, note that for strongly convex functions (Theorem 4.4), we provided the convergence up to $\epsilon$ tolerance and have provided a condition on the choice of the number of samples $N_k$ (lines 1220 to 1248 in Appendix D.4). This condition takes care of the convergence of the infeasibility gap up to $\epsilon$ tolerance.
>
> >Q) On Line 239, it is claimed that $\bar{\alpha}$ can be computed based on ...
>
> We thank the reviewer for pointing out the mistake. Estimating $M_g$ (which is $M_f$ in our paper) can be hard if we don't have an upper bound on the diameter $B_0 + ||x^*||$ (which bounds the iterates). We will remove the remark made in line 239.
>
> >Q) The statement of \textbf{Theorem 5.3} is unclear....
>
> Thanks to the reviewer for catching this omission. We can fix the issue by upper estimating $\sum_{k=1}^\tau (\cdot) \leq \sum_{k=1}^T (\cdot)$ before taking the expectation. Then, everything will work perfectly fine.
>
> >Q) The proof of \textbf{Theorem 5.3} is not fully justified....
>
> Thanks to the reviewer for this comment. The claim is incorrect since the sequence $\hat{d}_k$ is not monotonically decreasing. To correct this, we redo that part of the analysis using analogous steps as done in ``DoWG Unleashed: An Efficient Universal Parameter-Free Gradient Descent Method" by Ahmed Khaled, Konstantin Mishchenko, and Chi Jin (proof of Lemma 3). In the final bound of Theorem 5.3, The quantity $A_1(T)$ will become $$A_1(T) = \frac{3 D M_f}{\sqrt{T}} \left(\frac{D}{r}\right)^{\frac{2}{T}} \ln \left( \frac{e D^2}{r^2} \right).$$
>
> https://anonymous.4open.science/r/rebuttal-ICML2026/ICML2026_Rebuttal.pdf has step by step analysis and detailed response to all the comments.
>
> >Q) The QCQP examples are too small, .... Besides, the figures in the experiments seem to suggest the proposed algorithms only achieve $10^{-2}$ infeasibility tolerance. This performance looks unsatisfying.
>
> Large scale QCQP problems were also considered but CVXPY which provides benchmark to the optimal solution fails to work in those cases (Appendix G.1 between lines 1994 to 2001). Whereas our methods work as per theory. Moreover, the infeasibility tolerance decreases quickly as sample size is increased validating the theory. We simulated with $N_k = 100$ and infeasibility went down to 0.
>
> >Q) Some minor comments...
>
> Thanks to the reviewer for pointing them out. We will fix the typos and other minor comments and cite the related paper pointed out by the reviewer.
>
> >Q) In \textbf{Algorithm 1}, could you do a line-search to choose $\alpha$?
>
> This is a very interesting point. We believe that the reviewer wants to ask if line search to choose $\alpha$ is possible for Algorithm 2 because for Algorithm 1, Polyak step size is the optimal one. Yes, a line search to choose $\alpha_k$ is definitely possible and would be very beneficial if we have no knowledge on any of the problem parameters. New analysis will come up and it will add some extra steps to perform the search technique. But in our current setting for the strongly convex functions, we assume that we have knowledge of $L$ and $\mu$ and we choose step size $\alpha_k$ based on those parameters.
>
> >Q) When would you choose a random $N_k$?
>
> We thank the reviewer for bringing up this point. The random choice of batch size $N_k$ was primarily used for theoretical exploration and understanding how the methods behave under various scenarios. We were interested in understanding how convergence bounds behave under different distributions. Based on our analysis, when $N_k$ is deterministic, it does indeed provide the best scenario. However, when $N_k$ is random, we impose conditions on the distribution parameters. The use of random sampling was mainly for study purposes to see how the rates shift (Appendix D.5 and E.2). We also conducted simulations with $N_k$ drawn from a Poisson distribution, and the performance was similar to the deterministic case. While we did not include those simulation results in detail, we are happy to add them in a revised version if the reviewer finds that helpful.
>
> To view detailed replies for all the comments, please check https://anonymous.4open.science/r/rebuttal-ICML2026/ICML2026_Rebuttal.pdf
>
> Additionally, all our code is available at https://anonymous.4open.science/r/anony666-adaptive , where we have also included a new experiment on logistic loss with fairness constraints.

---

> > ### Author Rebuttal · Reviewer_v8N2 · 2026-04-02
> >
> > Thank you for the detailed response. I've read the updated proofs, and they look plausible. I'll keep the score for now and raise it to 4 once I finish checking the rest of the results.

---

### Official Review · Reviewer_WweD · 2026-03-12

**Soundness:** 3
**Presentation:** 3
**Significance:** 2
**Originality:** 2
**Overall Recommendation:** 4
**Confidence:** 3

**Summary:**

This paper studies constrained optimization problems in which the feasible set is defined by the
intersection of many constraints, making projection onto the full feasible region computationally
difficult. The authors consider settings where projection onto an auxiliary set is simple but enforcing
all constraints simultaneously is challenging.
To address this issue, the paper proposes randomized feasibility methods combined with adaptive
step-size strategies. The algorithm alternates between gradient (or subgradient) updates for optimizing
the objective function and randomized feasibility updates that enforce constraint satisfaction
by sampling subsets of constraints.
The authors analyze both strongly convex smooth objectives and general convex (possibly nonsmooth)
objectives. The proposed methods achieve linear convergence in expectation for the strongly
convex case and sublinear convergence guarantees for the general convex case with parameter-free
adaptive step sizes. Experimental results on QCQP and SVM problems demonstrate the computational
efficiency of the approach.

**Compliance With Llm Reviewing Policy:**

Affirmed.

**Final Justification:**

My problem is solved. I have raised my score.

**Key Questions For Authors:**

• The paper proposes randomized feasibility updates combined with adaptive step-size strategies
for constrained optimization. Could the authors further clarify the main conceptual differences
between the proposed approach and existing randomized projection or feasibility-based optimization
methods?

• The experiments are mainly conducted on QCQP and SVM problems. Could the authors provide
additional empirical results on more diverse constrained optimization tasks, particularly larger-scale
problems, to further demonstrate the practical effectiveness of the proposed method?

• The proposed framework involves randomized sampling of constraints during feasibility updates.
How sensitive is the performance of the algorithm to the sampling strategy and the number of
sampled constraints at each iteration?

**Limitations:**

While the paper presents a theoretically grounded framework for constrained optimization with
randomized feasibility updates and adaptive step-size strategies, several limitations remain.
First, the proposed methods rely on repeated randomized sampling of constraints during feasibility
updates. When the number of constraints is extremely large, the computational cost associated
with evaluating constraint violations may still become significant.
Second, the empirical evaluation is relatively limited in scope. Although the experiments on QCQP
and SVM problems provide useful evidence, additional evaluations on larger-scale or more diverse
constrained optimization tasks would help further assess the practical effectiveness of the approach.
Third, the performance of the algorithm may depend on the sampling strategy and the number
of sampled constraints at each iteration. A deeper analysis of how these design choices affect
convergence behavior and practical performance could provide further insights into the method.
Finally, the proposed framework is primarily evaluated in controlled experimental settings, and its
behavior in more complex real-world optimization scenarios remains to be further explored.

**Strengths And Weaknesses:**

Strengths:

• The paper addresses constrained optimization problems where projection onto the full feasible
region is computationally expensive, which is a practically relevant setting in many large-scale
optimization tasks.

• The proposed framework combines randomized feasibility updates with adaptive step-size strategies,
providing a principled way to handle complex constraint sets without requiring direct projection
onto their intersection.

• The paper provides theoretical guarantees for both strongly convex smooth objectives and general
convex (possibly nonsmooth) objectives, including linear convergence in the strongly convex case.
• The proposed methods are supported by experimental evaluations on benchmark problems such
as QCQP and SVM, demonstrating competitive computational performance.

Weaknesses:

• The level of novelty may be somewhat limited, as randomized feasibility updates and projection-based
methods have been studied extensively in the literature, and the paper could better clarify
its distinctions from existing approaches.

• The experimental evaluation is relatively limited in scope. In particular, the experiments are
conducted on a small number of benchmark problems (e.g., QCQP and SVM), and it would be
helpful to include additional large-scale or more diverse constrained optimization tasks to better
demonstrate the robustness and practical applicability of the proposed method.

• The algorithm introduces several design components (e.g., randomized constraint sampling and
adaptive step-size schemes), and a more detailed ablation study could help better understand
the contribution of each component.

---

> ### Author Rebuttal · Authors · 2026-03-31
>
> We thank the reviewer for their careful evaluation and helpful feedback.
>
> >Q) The level of novelty may be somewhat limited ...
>
> We thank the reviewer for bringing this point. The novelty is given in the contributions part of the paper; here, we would like to restate all the points in detail and draw a clear distinction about what is new here. (1) All the prior analysis for feasibility methods assumes boundedness of the subgradients of the constraint functions. In this work, we explicitly prove that the iterates and the gradients (as well as subgradients) of the functions stay bounded (lines 235 to 250 and Lemma D.1). The proof technique is not conventional and uses inductive hypothesis which is novel in itself. (2) We for the first time proved a linear convergence up to $\epsilon$-tolerance for Algorithm 2 by choosing adaptive step size and a different exponentially moving averaging scheme (Theorem 4.4) for constrained optimization problems that do not use a direct projection on the constraint set. (3) For constrained optimization, there exists no fully parameter-free, fully line search free, and projection-free method. Here, for the first time, we developed such a method with distance adaptive step sizes (Algorithms 3 and 4). (4) Compared to the prior research on DoWG method, by which Algorithm 3 is inspired, we improved the constants in the convergence rate analysis (Theorem 5.3). Moreover, compared to T-DoWG (tamed DoWG) (from which Algorithm 4 is inspired), we improved the step size selection constants (highlighted in detail in the footnotes of pages 31 and 32 of the Appendix), and also obtained better constants in the convergence analysis (Theorem 5.5). Moreover, we also showed that for Algorithm 4, the iterates of the method (and hence the subgradient of the objective and the constraint functions) are bounded. This result is new for constraint optimization that uses distance adaptive and feasibility methods and does not involve projections on the whole constraint set.
>
> Apart from these 4 major contributions, we also have two other minor contributions and exploration study. (5) In the prior literature on randomized feasibility methods, study was done only with deterministic constant batch size or with a deterministically growing batch size. Here, in addition to that study, we also studied batch sizes which are sampled from a random distribution. It is more of a theoretical exploration and the impact on the rates for such sampling strategies are presented in Appendices D.5 and E.2. (6) We connect the results from prior works [lewis1998error, bertsekas1999note, mangasarian1998error], and show that a \emph{uniform} upper bound on optimal dual multipliers exists under Strong Slater condition, without assuming compactness of the set $Y$. This result is not readily available in the existing literature. However, we do not claim any novelty here.
>
> >Q) The reviewer asked for extra simulations. Also asked large scale QCQP or SVM is not simulated.
>
> We thank the reviewer for this suggestion. We considered large-scale QCQP problems, but standard solvers such as CVXPY fail in these settings (Appendix G.1 lines 1994 to 2001), whereas our methods remain effective. Hence, we had to drop that case since no baseline for $f^*$ was available. The SVM experiment on MNIST is also high-dimensional. Additionally, we include a new experiment on constrained logistic regression with box and fairness constraints. Link to all of our codes and a pdf document for the fairness problem can be found here: https://anonymous.4open.science/r/anony666-adaptive .
>
> >Q) Could the authors further clarify the main conceptual differences between the proposed approach and existing randomized projection or feasibility-based optimization methods?
>
> If we have closed form expression for the projection on each individual set (i.e., the set is easy to project on), then we can perform randomized projection techniques by randomly sampling a set and projecting on it. However, if the set is not easy to project on, then we can replace the projection step by a feasibility step on the sampled constraint. In prior literature, gradient method with feasibility steps in the strongly convex case are analyzed with diminishing step sizes yielding $O(1/T)$ rate, but here, we have adaptive step sizes, that yields a linear convergence rate up to $\epsilon$ tolerance. Moreover, for convex functions (possibly non-smooth), we obtain the optimal $O(1/\sqrt{T})$ rate with a novel completely parameter-free and line search free step size selection.
>
> Unfortunately, due to character limitations we were not able to present point by point replies to all the comments in detail here. Hence, we wrote all the review comments in a pdf which is available in the link: https://anonymous.4open.science/r/rebuttal-ICML2026/ICML2026_Rebuttal.pdf . We kindly request the reviewer to refer to this document for detailed responses, including those addressing sensitivity to sampling strategies.

---

> > ### Author Rebuttal · Reviewer_WweD · 2026-04-03
> >
> > Thanks for your response.

---

### Decision · Program_Chairs · 2026-04-30

**Decision:**

Accept (regular)

**Comment:**

The paper proposes randomized feasibility algorithm with Polyak steps and adaptive stepsize strategies for minimizing an objective function over the intersection of lower-level sets of convex functions. the paper has uniformly weak-accept scores (4,4,4,4) and generally the reviewers liked the paper (although some minor concerns were raised).